# Oceanographic Processes Driving Low Oxygen Conditions Inside Patagonian Fjords

Pamela Linford[1,2], Iván Pérez-Santos[2,3,4,*], Paulina Montero[3,4], Patricio Díaz[2,5], Claudia Aracena[6,7], Elías Pinilla[8,14], Facundo Barrera[2,9,10], Manuel Castillo[11], Aida Alvera-Azcárate[12], Mónica Alvarado[13], Gabriel Soto[8], Cécile Pujol[12], Camila Schwerter[2], Sara Arenas-Uribe[2], Pilar Navarro[2], Guido Mancilla-Gutiérrez[2], Robinson Altamirano[2], Javiera San Martín[8], Camila Soto-Riquelme[8].

[1]Programa de Doctorado en Ciencias, Mención Conservación y Manejo de Recursos Naturales, Universidad de Los Lagos, Puerto Montt, Chile.

[2]Centro i-mar, Universidad de Los Lagos, Casilla 557, Puerto Montt, Chile.

[3]Center for Oceanographic Research COPAS Sur-Austral and COPAS COASTAL, Universidad de Concepción, Chile.

[4]Centro de Investigación en Ecosistemas de la Patagonia (CIEP), Coyhaique, Chile.

[5]CeBiB, Universidad de Los Lagos, Casilla 557, Puerto Montt, Chile.

[6]Centro de Investigación en Recursos Naturales y Sustentabilidad, Universidad Bernardo O'Higgins, Avenida Viel 1497, Santiago, Chile.

[7]Laboratorio Costero de Recursos Acuáticos de Calfuco, Universidad Austral de Chile, Valdivia, Chile.

[8]Instituto de Fomento Pesquero (IFOP), CTPA-Putemún, Castro, Chile.

[9]Fundación Bariloche and CONICET, San Carlos de Bariloche, Argentina .

[10]Centro Austral de Investigaciones Científicas (CADIC), CONICET, Bernardo Houssay 200, Ushuaia, Argentina.

[11]Centro de Observación Marino para estudios de riesgos del ambiente Costero, Universidad de Valparaíso, Chile.

[12]AGO-GHER, University of Liège, Belgium.

[13]Servicio Hidrográfico y Oceanográfico de la Armada de Chile.

[14]Department of Civil and Environmental Engineering, University of Maine, 5711 Boardman Hall, Orono, ME, USA.

*Correspondence to*: I. Pérez-Santos (ivan.perez@ulagos.cl), https://orcid.org/0000-0002-0184-1122.

## Abstract

The dissolved oxygen (DO) levels of oceanic-coastal waters has decreased over the last decades owing to the increase in surface and subsurface water temperature caused by climate change. In addition, biological and human activity in coastal zones, bays, and estuaries has contributed to the acceleration of current oxygen loss. The Patagonian fjord and channel system is one world region where low DO water (LDOW, 30%–60% oxygen

saturation) and hypoxia conditions (<30% oxygen saturation, 2 ml L$^{-1}$ or 89.2 µmol L$^{-1}$) is observed. An *in-situ* data set of hydrographic and biogeochemical variables (1507 stations), collected from sporadic oceanographic cruises between 1970 and 2021, was used to evaluate the mechanisms involved in the presence of LDOW and hypoxic conditions in northern Patagonian fjords. Results denoted two main areas with LDOW (e.g., Puyuhuapi Fjord-Jacaf

channel, Comau Fjord, and the Reloncaví estuarine system) extending from 25–400 m depth. Simultaneously, hypoxia was recorded in the Puyuhuapi Fjord, Jacaf Channel, and Quitralco Fjord. Quitralco registered the lowest values of DO (9.36 µmol L$^{-1}$ and 1.6% oxygen saturation) of the entire Patagonian fjord system. Areas of LDOW and hypoxia coincided with the accumulation of inorganic nutrients. Water mass analysis confirmed the contribution of equatorial subsurface water in the advection of the LDOW to only the Puyuhuapi Fjord and Jacaf Channel. In

addition, in Puyuhuapi Fjord, hypoxic conditions occurred when the community respiration rate (6.6 g C m$^{-2}$d$^{-1}$) exceeded the gross primary production estimate (1.9 g C m$^{-2}$d$^{-1}$) possibly due to the increased consumption of DO during the use of both autochthonous and allochthonous organic matter.  This study elucidates the physical and biogeochemical processes contributing to hypoxia and LDOW in the northern Patagonian fjords, highlighting the significance of performing multidisciplinary research and applying of numerical model.


**1 Introduction**

Hypoxic conditions and low dissolved oxygen water (LDOW)have expanded globally over the last decade along coastal waters and oceans (Schmidtko et al., 2017; Breitburg et al., 2018). While hypoxia is mostly attributed to anthropogenic processes, such as eutrophication (Díaz et al., 2001; Conley et al., 2009; Meire et al., 2013),

LDOW and deoxygenation occurs most prominently in the open ocean as a combination of natural and anthropogenic forcing (Oschlies et al., 2018; Garcon et al., 2019). Therefore, for decades, the scientific community has been paying close attention to this issue because of the expected impacts on the survival, abundance, development, growth, reproduction, and behavior of the most important taxonomic groups at different stages of their life cycles, such as mollusks, crustaceans, and fish (Sampaio et al., 2021, Ekau et al., 2010; Batiuk et al., 2009;

Vaquer-Sunyer y Duarte, 2008; Breitburg et al., 1997; Díaz and Rosenberg, 1995; Andrewartha and Birch, 1986; Davis, 1975), which could also affect all services provided to humans (Laffoley and Baxter, 2019).

Some examples of the impact of LDOW on biological behaviors are changes in the composition of benthic communities in prolonged periods of DO<4.2 ml L$^{-1}$ (Hoos, 1973); negative impact on the growth and abundance of cod (3.6 ml L$^{-1}$; 70%), the limit of the distribution of sardine larvae (2.6 ml L$^{-1}$; 50%), the distribution of jellyfish

(1.6 ml L$^{-1}$; 30%), the decrease of the abundance, swimming capacity and filtration of copepods (0.52 ml L$^{-1}$; 10%), described by Ekau et al., (2010). Additionally, oxygen deficiency affects the growth rate and feed conversion efficiency, and in some species, even increases the concentration of toxic substances (Davis, 1975). In a Patagonian fjord used for recreational fishing  of rainbow trout (*Salmo gairdneri*) , values under 50% saturation caused a reduction in swimming speed (Jones, 1971b) and altered respiration and metabolism (Kutty, 1968a). In the case of

coho salmon (*Oncorhynchus kisutch*), a commonly farmed species, a growth rate proportional to the oxygen level was observed for saturations between 40% and 80% (Herrmann, 1958), and the hypoxia modulated the transcriptional immunological response (Martinez et al., 2020). Finally, Pérez–Santos et al. (2018) reported habitat

reduction of microzooplankton in a Patagonian fjord (Puyuhuapi Fjord) due to the presence of hypoxic conditions at depths below 100 m.

Throughout the world's oceans, there are areas in which the dissolved oxygen (DO) is significantly lower than in well-oxygenated areas (such as <20 µM, ~ 0.4 mg L$^{-1}$, or 0.31 ml L$^{-1}$ as shown by Breitburg et al., (2018). These areas are known as oxygen minimum zones (OMZs). OMZs result from organic matter degradation, weak water circulation, long residence times, and weak ventilation (Fuenzalida et al., 2009). The major ocean OMZs are located in the Eastern South Pacific, the Arabian Sea, the Bay of Bengal (Indian Ocean), the West Bering Sea, and the Gulf of Alaska, covering approximately 8% of the total ocean or approximately 30 million km$^2$ (Paulmier and Ruiz-Pino, 2009; Fuenzalida et al., 2009).

Along the Perú-Chile coastline, the Eastern South Pacific (ESP) OMZ extends poleward, decreasing in strength and size to the south near the Patagonian fjord system (Silva et al., 2009). Recently, Linford et al. (2023) demonstrated poleward transport of hypoxic and LDOW of the Equatorial Subsurface water mass (ESSW) alongside the Patagonian region. As ESSW (originated in the Equatorial region) moves south, it passes throughout the OMZ, carrying oxygen-poor water (2−3 ml L$^{-1}$) with high nitrate concentration (20−30 µM) and elevated salinity (34.9) (Silva et al., 2009). Studies of water masses inside Patagonian fjords and channels have detected the presence of ESSW only in the northern region, between 41° S and 45° S. This water mass enters the northern Patagonia region via the Guafo mouth (Figure 1b), a deep channel with a depth of 150−200 m and width of 35 km (Sievers and Silva, 2008; Pérez-Santos et al., 2014; Schneider et al., 2014). ESSW is one of the main causes of hypoxia and LDOW inside Patagonian fjords (e.g., Puyuhuapi Fjord and Jacaf Channel). Nevertheless, these conditions are also found in other areas where sills block the pass of ESSW, e.g., the Reloncaví system (Reloncaví Fjord and Reloncaví Sound), Aysén, Comau, and Quitralco fjords (Figure 1) (Silva and Vargas, 2014; Linford et al., 2023; Díaz et al., 2023).

Regarding other processes favoring hypoxia and LDOW inside Patagonian fjords, Silva (2008) proposed those biological processes that include the consumption of DO, such as respiration and the remineralization of organic matter. A high load of organic matter (autochthonous and allochthonous) in the water column and sediments increases DO consumption during microbial community respiration, contributing to the LDOW content in most of the Patagonian fjord headwaters (Castillo et al., 2016) as well as higher community respiration rates than primary production (Montero et al., 2011; Montero et al., 2017a). Weak deep ventilation and long residence times in fjord waters are also assumed to promote a reduction in the DO concentration (Schneider et al., 2014; Silva and Vargas, 2014).

Finally, most published manuscripts hypothesize and discuss the processes favoring hypoxia and LDOW inside Patagonian fjords without showing any evidence, particularly that based on g in-situ data and fieldwork experiments. The Patagonian fjords ecosystem is under substantial continued economic pressure due to salmon aquaculture and other economic activities (Billi et al., 2022). The northern Patagonian fjords (Figure 1b, 1c) reported half of the national production and a significant number of salmon concessions (Billi et al., 2022). A risk analysis carried out in this region established that the Reloncaví estuarine system, and the Comau, Puyuhuapi, Quitralco, and Cupquelan fjords are regions with an especially elevated level of risk for the development of harmful algal blooms (HABs) and eutrophication events, owing to the nutrients input by the intense aquaculture of salmon

(Soto et al., 2021). Nevertheless, environmental studies on salmon farming have shown that this economic activity has a geographically limited impact, because most nutrients are quickly recycled by biological processes in the water column (Soto and Norambuena, 2004).

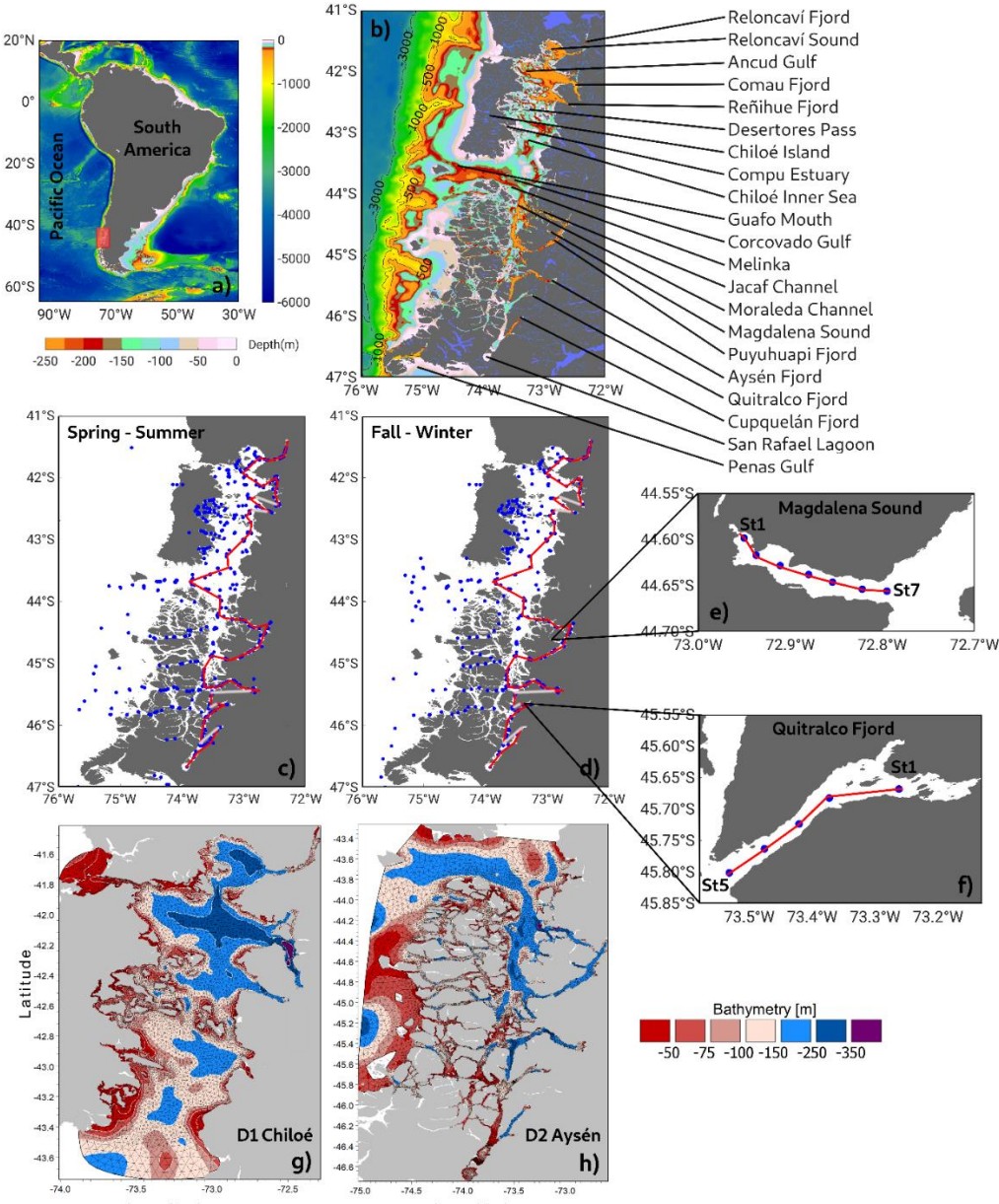

**Figure 1**. (a) Map showing the study area in the (b) northern region of Patagonian fjords. Color bars represent
bathymetric features from ETOPO2. (c, d) Sampling stations during the spring-summer and fall-winter seasons. Red lines in (c, d) represent along-fjord transects carried out to describe vertical features of hydrographic-chemical stations from Figure 2, 4, and 5. Data was collected on sporadic cruises, from 1970–2021 (see Table 1). (e, f) Details of Magdalena Sound and Quitralco Fjord sampling stations. (g, h) High-resolution model domain includes the northern Patagonia inner sea and is denoted as D1_Chiloé, while D2_Aysén covers the southern section. Bathymetry
used was based on an SHOA nautical chart from the Chilean Navy.

Another meaningful impact of salmon production is the increase in allochthonous organic matter load, which, when combined with the high production of autochthonous organic matter by phytoplankton, favors a decrease in DO in the water column and sediments (Quiñones et al., 2019). Salmon farming supplies allochthonous dissolved substrates through the dissolution of organic particles derived from feces and uneaten feed (Wang et al., 2012). This organic material is considered highly degradable (Nimptsch et al., 2015; Montero et al., 2022), promoting enhanced rates of heterotrophic bacterial activity (e.g., bacterial production (BP) and extracellular enzymatic activity (EEA) (Montero et al., 2022).

The main goal of this study was to analys the processes contributing to hypoxia and LDOW in northern Patagonia, such as ESSW advection, DO consumption during the use of organic matter (community respiration), biogeochemical processes, deep-water ventilation, and residence times of water inside fjords. An *in-situ* dataset of ~2000 profiles of hydrographic-chemical variables was used to describe the seasonal distribution of water masses and their relationship with biogeochemical processes. Primary production experiments and the output of a 3D hydrodynamic model were utilized to demonstrate the occurrence of other processes promoting low oxygen conditions in the northern Patagonian fjords.

## 2. Materials and Methods

2.1 Hydrographic and chemical data

A total of 1507 stations were used to describe northern Patagonia's hydrographical and chemical processes; 593 stations were sampled during the fall-winter seasons and 914 during the spring-summer seasons (Table 1). Most temperature and salinity records were obtained using a CTD (Conductivity, Temperature, and Depth profiler) instrument (e.g., SBE19 plus, SBE15, and AML Metrec XL). When SEB CTDs were used, data were processed according to the manufacturer's protocol and software (SBE Data Processing). When using AML CTD, the raw data underwent a quality control, eliminating records out of range according to historical CTD data (Pérez-Santos et al., 2014; Pérez-Santos et al., 2021; Linford et al., 2023), after which the data were averaged every 1 m. Additionally, membrane and optical sensors (e.g., SBE 43 and Optode 4831) together with the Winkler method have been used to obtain DO data (Strickland and Parsons, 1968). Experiments were conducted to validate the CTD-DO data with the Winkler method showing satisfactory results and high statistical correlation ($R^2$ ranging 0.92–0.99, Figure not included).

Figures 1b and 1c show the station positions during fall-winter and spring-summer. The absolute salinity ($S_A$ in $gkg^{-1}$) and conservative temperature ($\Theta$ in °C) were calculated using the Thermodynamic Equation of Seawater 2010 (TEOS-10) (IOC et al., 2010). In TEOS-10, absolute salinity represents the spatial variation in the composition of seawater, considering the different thermodynamic properties and gradients of horizontal density in the open ocean. Conservative temperature is similar to potential temperature but represents the heat content of seawater with greater precision.

A temperature/salinity (TS) diagram was constructed based on the conservative temperature and absolute salinity values, which was used to identify and quantify the water masses in the Patagonian fjords using only the salinity criteria proposed by Sievers (2008) and Sievers and Silva (2008). According to this classification, Estuarine

Water (EW, 0−1 g kg$^{-1}$) was detected in the surface layer owing to the water supply from rivers and summer time glacier melting. Below the EW, a Modified Subantarctic Water (MSAAW) layer was found, with salinities values

ranging 31−33 g kg$^{-1}$. The MSAAW originated from mixing of the EW and Subantarctic Water (SAAW, salinity 33−33.8 g kg$^{-1}$). Finally, an Equatorial Subsurface Water (ESSW) layer was identified, with salinity values higher than 33.8 g kg$^{-1}$. The ESSW was localized near the bottom of the Puyuhuapi Fjord and Jacaf Channel (Linford et al., 2023).

The long-term seasonal means were calculated using the Data Interpolation Variational Analysis (DIVA) gridding

software developed by the University of Liege (http://modb.oce.ulg.ac.be/mediawiki/index.php/DIVA). The DIVA software was used to analyze and interpolate datasets via an optimal interpolation method, which considers the coastline and bathymetry of the study area; the calculations were executed on a finite element mesh adapted to the gridding domains (Troupin et al., 2010). Additionally, the DIVA results were graphically displayed using the Ocean Data View software (https://odv.awi.de).


**Table 1**. Oceanographic campaigns carried out in Patagonian fjords and channels.

| Expeditions | Date | Season | Stations | Measurements |
|---|---|---|---|---|
| HUDSON | 06/03-01/04, 1970 | Summer | 112 | (Temperature, Salinity) [*] + O2(bottle-Winkler) + Nutrients (PO4, NO3, Si; bottle) |
| CIMAR-01 | 18/10-04/11, 1995 | Spring | 99 | CTD + O2(bottle-Winkler) + Nutrients (PO4, NO3, NO2, Si; bottle) + pH(bottle) |
| CIMAR-04-I | 28/09-09/10, 1998 | Spring | 31 | CTD + O2(bottle-Winkler) + Nutrients (PO4, NO3, Si; bottle) |
| CIMAR-07-I | 07-21/07/2001 | Winter | 49 | CTD + O2(bottle-Winkler) |
| CIMAR-07-II | 13-25/11/2001 | Spring | 51 | CTD + O2(bottle-Winkler) + Nutrients (PO4, NO3, Si; bottle) |
| CIMAR-08-I | 06-20/07/2002 | Winter | 51 | CTD + O2(bottle-Winkler) + Nutrients (PO4, NO3, Si; bottle) |
| CIMAR-08-II | 16-24/11/2002 | Spring | 39 | CTD + O2(bottle-Winkler) + Nutrients (PO4, NO3, Si; bottle) |
| CIMAR-09-I | 09-23/08/2003 | Winter | 58 | CTD + O2(bottle-Winkler) + Nutrients (PO4, NO3, Si; bottle) |
| CIMAR-09-II | 07-20/11/2003 | Spring | 55 | CTD + O2(bottle-Winkler) + Nutrients (PO4, NO3, Si; bottle) |
| CIMAR-10-I | 26/06-31/08, 2004 | Winter | 49 | CTD + O2(bottle-Winkler) + Nutrients (PO4, NO3, Si; bottle) |
| CIMAR-10-II | 12-23/11/2004 | Spring | 63 | CTD + O2(bottle-Winkler) + Nutrients (PO4, NO3, Si; bottle) |
| CIMAR-11-I | 18-25/07/2005 | Winter | 78 | CTD + O2(bottle-Winkler) + Nutrients (PO4, NO3, Si; bottle) |
| CIMAR-11-II | 11-21/11/2005 | Spring | 80 | CTD + O2(bottle-Winkler) + Nutrients (PO4, NO3, Si; bottle) |
| CIMAR-12-I | 10-19/07/2006 | Winter | 32 | CTD + O2(bottle-Winkler) +Nutrients (PO4, NO3, Si; bottle) |
| CIMAR-12-II | 04-12/11/2006 | Spring | 40 | CTD + O2(bottle-Winkler) + Nutrients (PO4, NO3, Si; bottle) |
| CIMAR-13-I | 27/07-07/08, 2007 | Winter | 42 | CTD + O2(bottle-Winkler) + Nutrients (PO4, NO3, Si; bottle) |
| CIMAR-13-II | 02-12/11/2007 | Spring | 46 | CTD + O2(bottle-Winkler) + Nutrients (PO4, NO3, Si; bottle) |
| CIMAR-17 | 17/10-14/11, 2011 | Spring | 195 | CTD + O2(bottle-Winkler) + Nutrients (PO4, NO3, Si; bottle) |
| CIMAR-18 | 01/06-04/07, 2012 | Fall-Winter | 98 | CTD + O2(bottle-Winkler) + Nutrients (PO4, NO3, Si; bottle) |
| CIMAR-19 | 04-16/07/2013 | Winter | 70 | CTD + O2(bottle-Winkler) + Nutrients (PO4, NO3, Si; bottle) |
| CHEPU-IFOP | 05-17/08/2017 | Winter | 22 | CTD + O2(optic) |
| CIMAR-24 | 26/09-16/10, 2018 | Spring | 37 | CTD + O2(bottle-Winkler) + Nutrients (NO3; bottle) |

| CHEPU-MR-I | 06-08/06/2018 | Fall | 12 | CTD + O2(optic) |
|---|---|---|---|---|
| CHEPU-MR-II | 11/10/2018 | Spring | 2 | CTD + O2(optic) |
| PN-I (IFOP) | 13-25/11/2020 | Spring | 32 | CTD + O2(optic) + Nutrients (PO4, NO3, NO2, Si; bottle) + pH, Tur, Fluor (optic) |
| PN-II (IFOP) | 24/02-04/03, 2021 | Summer | 32 | CTD + O2(optic) + Nutrients (PO4, NO3, NO2, Si; bottle) + pH, Chl-a, Tur, Fluor (optic) |
| PN-III (IFOP) | 28/07-10/08, 2021 | Winter | 32 | CTD + O2(optic) + Nutrients (PO4, NO3, NO2, Si; bottle) + pH, Chl-a, Tur, Fluor (optic) |
| | | | | 593 Fall-Winter + 914 Spring-Summer = 1507 stations |

*Temperature and salinity were measured with a reversing thermometer and inductive salinometer respectively.

2.2 Primary production, community respiration, bacterial production, and phytoplankton community.

During the spring-summer period of 2020−2021 and summer of 2022, nine *in situ* experiments were conducted to measure gross primary production (GPP) and community respiration (CR) in some fjords of northern Patagonia (Table 2). Water samples were obtained from three depths (2, 10, and 20 m depth) at each sampling station. GPP and CR rates were estimated from changes in dissolved oxygen concentrations observed during *in situ* incubation of light and dark bottles (Strickland, 1960). Water from the Niskin bottles was transferred into 125 mL

(nominal volume) borosilicate bottles (gravimetrically calibrated) using a silicone tube. Five time-zero bottles, five light bottles, and five dark bottles were used at each incubation depth. Water samples were collected at dawn and incubated throughout the entire light period (approximately ~8−9 h). Time-zero bottles were fixed at the beginning of each experiment, whereas the light and dark incubation bottles were attached to the surface-tethered mooring system. The samples were incubated at the depths from which they were collected.

Dissolved oxygen concentrations were determined according to the Winkler method (Strickland and Parsons, 1968), using an automatic Metrohom burette (Dosimat plus 865) and automatic end-point detection (AULOX Measurement System). Daily GPP and CR rates were calculated as follows: GPP = (mean $[O_2]$ light bottles – mean $[O_2]$ dark bottles); CR = (mean $[O_2]$ time zero bottles – mean $[O_2]$ dark bottles). The GPP and CR values were converted from oxygen to carbon units using a conservative photosynthetic quotient of 1.25 (Williams

and Robertson, 1991) and a respiratory quotient of 1.

    A total of five bacterial production (BP) experiments were performed during the same period as mentioned above (Table 2). Experiments were conducted using the same water samples collected for the *in situ* GPP and CR incubation experiments. The BP estimates were based on the incorporation of leucine into proteins using the microcentrifugation method (Smith and Azam, 1992). Briefly, a blank and three samples (1.5 mL) were taken from

each sampling depth and incubated with L-[3,4,5-$^3$H]-leucine (123.8 Ci mmol$^{-1}$, 40 nM final concentration) in the dark for 1 h. After incubation, samples were extracted with 100% trichloroacetic acid (TCA), rinsed with 5% TCA, and centrifuged at 13500 rpm twice for 15 min before removal of the supernatant. Liquid scintillation cocktail Ecoscint (1 mL) (National Diagnostic) was added to each sample. The samples were counted for dpm using a Packard (Mod. 1600 TR) liquid scintillation counter. Discrete depth estimates of the GPP, CR, and BS rates were

integrated to 20 m using the trapezoidal method.

    For analyses of the phytoplankton community, water samples were collected from four discrete depths (2, 4, 10, and 20 m) using a 5 L Niskin bottle. Samples were stored in 120 mL clear plastic bottles and preserved in 1% Lugol's iodine solution (alkaline). From each sample, a 10 mL subsample was placed in a sedimentation chamber

and allowed to settle for 12 h (Utermöhl, 1958) prior to identification at 40× and 100× under an inverted microscope (Carl Zeiss, Axio Observer A.1). Finally, taxonomic descriptions from Tomas (1997) were used to identify phytoplankton composition.

**Table 2**. *In situ* experiments were carried out in Patagonian fjords and channels. Gross primary production (GPP), community respiration (CR), and bacterial secondary production (BSP).

| Fjord region | Date (mm-dd-yyyy) | Season | Measurements (g C m$^{-2}$d$^{-1}$) | | | |
|---|---|---|---|---|---|---|
| | | | GPP | CR | BSP | GPP:CR |
| *Reloncaví | 02/27/2009 | Summer | 3.83 | 3.31 | - | 1.16 |
| Compu | 10/03/2020 | Spring | 1.22 | 0.92 | 0.06 | 1.32 |
| Quitralco | 11/17/2020 | Spring | 1.41 | 2.66 | 0.05 | 0.53 |
| Camou | 12/12/2020 | Spring | 0.12 | 1.67 | 0.15 | 0.07 |
| Puyuhuapi | 01/21/2020 | Summer | 1.89 | 6.64 | 0.25 | 0.28 |
| Reloncaví | 01/14/2021 | Summer | 2.60 | 1.90 | 0.58 | 1.37 |
| Compu | 03/19/2022 | Summer | 1.25 | 0.76 | - | 1.64 |
| Quitralco | 02/28/2022 | Summer | 0.73 | 1.71 | - | 0.42 |
| Camou | 03/16/2022 | Summer | 0.56 | 1.12 | - | 0.50 |
| Puyuhuapi | 01/20/2022 | Summer | 2.18 | 2.55 | - | 0.85 |

*Data were taken from Montero et al. (2011).

2.3 Biogeochemical variables and analysis

Biogeochemical variable datas were collected in November 2020 (Expedition PN-I IFOP, Table 1). The water samples were obtained by filtering seawater collected (1−2 L) using a 25-mm diameter, GF/F filter pre-combusted with a 0.7μm pore diameter. Suspended particulate matter (SPM, μg L$^{-1}$), was determined by gravimetry using the weight difference between the dried filter and the same filter before filtration (Grasshoff et al., 2009). Particulate organic carbon (POC, μmol L$^{-1}$), total nitrogen (TN, μmol L$^{-1}$), stables carbon ($\delta^{13}$C, ‰) and nitrogen ($\delta^{15}$N, ‰) isotopes were obtained following the method described by Verardo et al., (1990), with modifications by Barrera et al., (2017) and Díaz et al., (2023), and measured at the Stable Isotope Facility at the Pontifical Catholic University of Chile by using an elemental analyzer (EA Flash 2000 Thermo Finnigan), interfaced to a continuous flow isotope ratio mass spectrometer (IRMS Delta V Advantage).

The stable carbon isotope composition of organic carbon ($\delta^{13}$C) can vary depending on its origin, with values ranging from −23 to −19‰ for carbon sources such as marine phytoplankton (Fry and Sherr, 1989; Harmelin-Vivien et al., 2008) and values closer to those of terrestrial organic matter (−30 to −26‰) for other sources (Fry and Sherr, 1989).By analyzing the carbon isotopic composition of organic carbon, it is possible to differentiate carbon between different sources and estimate their contributions. We calculated the relative importance of allochthonous and autochthonous organic matter with two-source endmember mixing models (Bianchi, 2007). The importance of autochthonous marine and allochthonous terrestrial fractions were calculated based on the following equation:

$$\%POC_{allo} = \frac{(\delta^{13}C_s - \delta^{13}C_m)}{(\delta^{13}C_t - \delta^{13}C_m)} 100\%$$

where $\delta^{13}C_s$ is the isotopic composition of a sample, $\delta^{13}C_m$ is the marine endmember from more oceanic stations (-13.447 ‰), and $\delta^{13}C_t$ terrestrial is the riverine/lake endmember values for POC (-42.933 ‰) as proposed for this area by González et al., (2019). The resulting $\%POC_{allo}$, represents the relative contribution of allochthonous (terrestrial) organic carbon to the overall organic carbon pool. Higher values indicate a more significant influence of terrestrial organic carbon, while lower values suggest a higher proportion of autochthonous (marine) organic carbon (González et al., 2019). The carbon: nitrogen ratio was also calculated as a proxy for the organic matter pool (Barrera et al., 2017).

Dissolved inorganic nutrients ($NO_3^-$, $NO_2^-$, $PO_4^{3-}$, and $Si(OH)_4$) were analyzed from 15 mL seawater samples, stored at -20 ºC in HDPE bottles, using a Seal AA3 Autoanalyzer according to the methodology described by Grasshoff et al., (1983) and standard methods for seawater analysis (Kattner and Becker, 1991). Chromophoric Dissolved Organic Matter (CDOM) plays a crucial role in understanding the optical properties and biogeochemical processes in aquatic ecosystems (Stedmon and Nelson, 2015). It is an important component of dissolved organic matter (DOM) that absorbs light in the visible spectrum (ultraviolet and blue wavelengths) and fluoresces at longer wavelengths (Stedmon and Nelson, 2015). CDOM was determined by fluorometry using quinine sulfate dihydrate ($\mu gL^{-1}$ QSU) diluted in 0.1 N sulfuric acid at a specific wavelength (Ex/Em = 350/450 nm) as standard in a Trilogy Turner Design fluorometer and CDOM module (Kim et al., 2018).

## 2.4 Satellite images

Sentinel-2 level 1 images were downloaded from the Copernicus Open Access Hub (https://scihub.copernicus.eu/) for specific dates and regions, as shown in Table 3. The chosen dates correspond to periods of high discharge of freshwater from the continent. Using ACOLITE v 20220222.0 (https://odnature.naturalsciences.be/remsem/software-and-data/acolite, (Vanhellemont and Rudick, 2018), we calculated suspended particulate matter following Nechad et al. (2016). These data have a spatial resolution of 10 m and allow the resolution of the small-scale distribution of suspended particulate matter within the narrow channels and fjords of the region.

Table 3. Dates and regions of the Sentinel-2 datasets analyzed.

| Date | Region |
| --- | --- |
| 17 March 2017 | Puyuhuapi Fjord |
| 09 May 2017 | Reloncaví Sound and Fjord |
| 06 April 2018 | Comau Fjord |
| 09 June 2022 | Quitralco Fjord |

## 2.5 Marine current register with ADCP

In the study region, three mooring systems were deployed in the Corcovado Gulf and Puyuhuapi Fjord. In each mooring, a Teledyne RD Instruments (TRDI) and a WorkHorse−300 kHz Acoustic Doppler Current Profiler (ADCP) were installed with the transducers facing downward. The instruments were configured with 1-m cell size and 1 h ensembles. The moorings covered the period from January to December 2016, with the ADCP sensors moored between 40 and 100 m. The ADCP provides magnitude and direction (in Earth coordinates); thus, the

current vector is decomposed into zonal (u) and meridional (v) components. The basic and standard protocol of quality control was applied to identify outliers and low-quality data following the methodology suggested by TRDI. To obtain the mean pattern of the zonal and meridional components of the currents, they were filtered using a low-

pass cosine-Lanzcos filter of 121 weight and 40 h half-power.

2.6 Circulation model

To gain a better understanding of the physical forcing related to the DO dynamics, we used the hydrodynamic model MIKE 3 FM), which solves continuity, momentum, temperature, and salinity transport

equations using the finite volume method (DHI, 2019). In this study, two high-resolution model domains include the North Patagonian inner sea named D1_Chiloé (41.3° S– 43.7° S) and D2_Aysén, with an area from 43.6°S to 46.8° S (Figures 1d, 1e). The MIKE 3 model, maintaining the same configuration, has a track record of previous validation and application to investigate the physical aspects of HABs (Díaz et al., 2021, Mardones et al., 2021) as well as other oceanographic processes (Pérez-Santos et al., 2021, Soto-Riquelme et al. 2023). It has also been

utilized for studying sub-domains within this Patagonian region (Pinilla et al., 2020; Díaz et al., 2021).

The time frame for the modeling process extended over six years, from 2016 to 2021. The use of two overlapping domains, specifically D1_Chiloe and D2_Aysen was influenced by computational capacity, resulting in a staggered development over time, from 2016 onwards. Initially, the D1_Chiloe domain was developed, followed by the D2_Aysen domain. The model configuration for both domains assumes a hydrostatic variant of the MIKE 3

model, with vertical discretization executed via hybrid coordinates: surface-based sigma coordinates and z-level coordinates in the bottom layers. The model included 55 layers for D1_Chiloe and 40 layers for D2_Aysen, which were utilized to accurately depict stratification, focusing on the highest resolution in the upper water column (approximating ~ 1m near the surface). Horizontal eddy viscosity was characterized using the Smagorinsky model (Smagorinsky, 1963), while the vertical viscosity was incorporated through the $\kappa - \varepsilon$ turbulence scheme, which

solves the transport equations for both turbulent kinetic energy ($\kappa$) and turbulent dissipation rate ($\varepsilon$) (Rodi, 1984). Bathymetry was based on SHOA nautical chart soundings (horizontal resolution of 500 meters), and a digital elevation model was constructed using the natural neighbor method (Sibson, 1981). This domain was discretized using triangular elements of different sizes. The highest resolution was observed in coastal, narrow, and shallow areas, with an average element size of ~300 m, whereas the spatial resolution near the boundary was ~1000 m.

Atmospheric forcings, such as wind stress and heat fluxes over the sea surface, were introduced using spatially and temporally varying fields from the Weather Research and Forecasting (WRF) atmospheric model (Skamarock et al., 2008), development by Instituto Fomento Pesquero (IFOP), named in this manuscript as WRF-IFOP. The WRF-IFOP model, with its superior spatial resolution of 3 km compared to global models, enables a more accurate representation of the complex fjord topography.  The performance of the WRF-IFOP model was

evaluated as described by Pinilla et al. (2020). Soto-Riquelme et al. (2023) evaluated the performance of the WRF-IFOP model near the Guafo Mouth and confirmed its effective correlation with meteorological stations in at area. Notably, their results revealed a high correlation with atmospheric pressure (R=0.99), accompanied by significant

correlations for wind direction: R=0.81 for the east-west component and a more robust correlation of R=0.94 for the north-south component (Soto-Riquelme et al., 2023).

Freshwater sources were identified through the FLOW-IFOP hydrological model, which employs precipitation and temperature series data from the CR2MET (Climate Resilience 2 Meteorological) gridded product (http://www.cr2.cl/datos-productos-grillados/). Using this data, characterized by a spatial resolution of $5 \times 5$ km, enables the simulation of runoff and the calculation of daily discharge series. Based on FLOW-IFOP estimates, the average annual freshwater discharge entering the D1_Chiloé domain during the 2016–2021 period was

approximately 2545 $m^3$ $s^{-1}$, while for the D2_Aysén domain, it was approximately 3126 $m^3$ $s^{-1}$. The performance of the FLOW-IFOP model at the gauged river stations of the Chilean Water Authority can be accessed at http://chonos.ifop.cl/flow/. Furthermore, the correlations for the three main rivers in these regions—Puelo, Palena, and Aysén, Table S1, Fig. S1 were found to be 0.88, 0.76, and 0.87 respectively (see Fig. S2 in the supplementary material).

The water levels at the open boundary were set using harmonic analysis (Pawlowicz et al., 2002), based on a regional barotropic model data (Pinilla et al., 2012). The performance of water levels at seven locations (Table S1, Fig. S1) is presented as supplementary material (Table S2, Figure S3). The results demonstrate robust correlations between the sea level at different stations, with minimum correlations ranging from 0.93 to 0.97 (Table S2). Additionally, they exhibited an appropriate amplification of amplitude particularly in the northern area of this

domain (Puerto Montt and Comau fjord) (Figure S3).

The flow data at the boundaries were set to zero in our hydrodynamics model, leading to internal balance in the current dynamics within the model domain across both spatial and temporal scales. The performance of the current data within the models contrasted with the ADCP data from different mooring points (Figure S1, Table S1), as shown in the supplementary material (Figures S4 and S5). The hydrodynamic model current was evaluated

against the current velocities gathered by the ADCP. An empirical orthogonal function analysis (Thomson and Emery, 2014) was performed on the currents (both observed and modeled) to discern if the model could accurately depict the dominant modes of variability identified with the observations. Similar analyses have been previously performed to assess the suitability of hydrodynamic models in Patagonia (Pinilla et al., 2020, Soto-Riquelme et al., 2023).

Mode 1 plus Mode 2 accounted for over 95% of the variance in the model, whereas they accounted for approximately 86% of it in the observations, except in the Quitralco fjord, with a 77% variance observed in the model and 62% in the ADCP (Figure S4). The distribution of these percentages of variance between the model and the ADCP was correlated with an R-value of 0.96, as shown in Figure S5, which highlights the distinctiveness of the Quitralco fjord. On the other hand, the spatial vertical structures of Modes 1 and 2 between the model and the ADCP

data were identical, generally showing agreement in barotropic structures, two-layer baroclinic structures, and even three-layer structures, as observed in the Quitralco fjord (Figure S4). Though the model is not without flaws, our analysis suggests an agreement between the modes of variability, further implying a proficiency of the model in representing the processes vital to water transport in different locations.

The temperature and salinity boundary conditions were derived from CTD profiles obtained during CIMAR
FIORDOS oceanographic cruises (Guzman and Silva 2002; Silva and Guerra 2004; Valdenegro and Silva 2003;
Carrasco and Silva 2010). The performance of temperature time series and salinity fields within the model are
presented as Supplementary Material. The model demonstrated commendable alignment in replicating the annual
temperature cycle at various depths and locations (Table S1, Figure S1), generally exhibiting R-values above 0.78.
An exception was observed in the Moraleda Channel at a depth of 120 m, where a lower R-value (0.28) was
recorded (Figures S6 and S7).

To assess the water masses present within the fjords, salinity was employed as a key parameter. High
salinity waters (>33.5 g kg$^{-1}$), associated with SAAW and ESSW ocean masses, enter the deep layer of the Guafo
mouth, traverse the Corcovado Gulf, and end their journey in the deep layers of the Puyuhuapi Fjord and Jacaf
Channel. A reduction in salinity, indicative of Estuarine Water (EW), is also observed due to the melting of ice from
the San Rafael Lagoon. The model's task was to accurately depict the arrival of specific salinity categories inside the
fjords and channels (see Figure S8 in the Supplementary Material). Although the 34 g kg$^{-1}$ isohaline is not visually
present within the model domains, we believe the model successfully replicates the spatial structure, which suggests
that the processes controlling salinity transport within the fjords were successfully incorporated. While the model
may not perfectly replicate these processes, it does allow us to understand the fundamental physical transport
mechanisms at play.

The MIKE 3 FM hydrodynamic model data were used to calculate the flushing time via a conservative
tracer. According to Takeoka (1984) and Monsen et al. (2002), flushing time is defined as the time required for the
total mass of a material within a specific area such as a fjord, sound, or bay to be reduced by a factor of e$^{-1}$
(approximately 37%) (Prandle, 1984). The interior and exterior of the area were assigned initial concentration values
of 1 and 0, respectively. Over the course of the simulation, the original water mass of the basin was incrementally
replaced with inputs from open boundaries and rivers. This variable symbolizes the proportion of original water in
each element within the area of interest at a given time, thereby, facilitating the identification of less flushed areas
within the modeled basins (Andrejev et al., 2004). The final values were derived from a temporal average from 2016
to 2021, and a vertical average was calculated from 50 m to the bottom, which represents the area that typically
displays the lowest oxygen values. In the present study, the flushing time was implemented in the Ecolab module of
the MIKE 3 FM for every fjord within D1_Chiloé and D2_Aysén.

The MIKE 3 FM hydrodynamic model data was utilized in calculating the flushing time via a conservative
tracer. According to Takeoka (1984) and Monsen (2002), flushing time is defined as the time required for the total
mass of a material within a specified area - such as a fjord, sound, bay, etc. - to be reduced by a factor of e$^{-1}$
(approximately 37%) (Prandle, 1984). The interior and exterior of the area were assigned initial concentration values
of 1 and 0 respectively. Over the course of the simulation, the original basin's water mass was incrementally
replaced with inputs from open boundaries and rivers. This variable symbolizes the proportion of the original water
in each element within the area of interest at a given time, thereby facilitating the identification of less flushed areas
within the modeled basins (Andrejev et al., 2004). Furthermore, the final values are derived from a temporal average
from 2016 to 2021 and a vertical average was calculated from 50m to the seabed - the zone typically displaying the

lowest oxygen values. In the present study, the flushing time was implemented in the Ecolab module of the MIKE 3 FM for every fjord within D1_Chiloé and D2_Aysén.

### 3 Results

3.1. Long-term annual mean of hydrographic-chemical parameters

The long-term annual means include all data sets presented in Figures 2a and 2b. This section scrutinizes the behavior of conservative temperature, absolute salinity, DO, and inorganic nutrients. The long-term seasonal mean of conservative temperature (CT) denoted warmer water in the northern region (41.5°–43° S) during both seasons, between the Desertores Pass and the Reloncaví Fjord (Figure 2c-d). In contrast, cold waters were observed in the deep layer of the Puyuhuapi Fjord and Jacaf Channel, but colder water was registered in San Rafael Lagoon. This location also had the lowest absolute salinity ($S_A$=15–21 gkg$^{-1}$), indicating the presence of EW owing to the contribution of the ice melting from the San Rafael Lagoon and rivers discharges (Figure 2e-f). In this area, EW moved from south to north and mixed with the SAAW, contributing to the origin of the MSAAW, as was observed in the vertical-horizontal salinity distribution. Moreover, EW was also observed in the northern domain of the study area, especially at the surface layer in the Reloncaví system and the Comau and Reñihue fjords, contributing to the formation of the MSAAW. In this region, the origin of EW was mainly attributed to the freshwater supply from river discharge. We identified two different sources favoring the presence of EW and formation of the MSAAW: 1) The combination of ice melting from the San Rafael Lagoon with river discharge in the southern region and 2) the freshwater supply by river discharge in the northern region. Both sources contributed to the difference in conservative temperature observed in the MSAAW.

Finally, ESSW enters the deep layer of the Guafo mouth, crosses the Corcovado Gulf, and ends its travel at the deep layers of the Puyuhuapi Fjord and Jacaf Channel. During the fall-winter season, a slight reduction in ESSW distribution was observed (Figure 2e–2f). In the area contained by the ESSW, low DO (LDOW) and hypoxic waters were observed, but LDOW was registered at Reloncaví Fjord, where ESSW was not observed.

In general, the Chiloé Inner Sea showed a homogenized water column, mainly during the fall-winter seasons, in which high DO values (267–312 µmol L$^{-1}$) and oversaturated waters (< 100% DO Saturation) were registered. Additionally, more extensions of the hypoxic conditions and LDOW were registered during the spring-summer seasons (Figure 2g–2h). Apparent oxygen utilization showed higher values (~200 µmol kg$^{-1}$) in areas where LDOW and hypoxia were detected (Figure 2i-j).

Table 4 details the statistical characteristics of the water masses identified in this study. Furthermore, Figure 3 shows a TS diagram representing the quantification of water masses, which highlights the dominance of the MSAAW, with 60.96% and 54.67% proportions during the spring-summer and fall-winter seasons, respectively. The SAAW was the second dominant water mass, with 15.25% and 22.64% proportions, followed by the EW. Finally, the ESSW showed the smallest proportion, with 10.77% and 11.15% values. The ESSW was generally characterized by cold, salty, and poor DO water when compared to those of the EW, MSAAW, and SAAW (Table 4 and Figure 3).

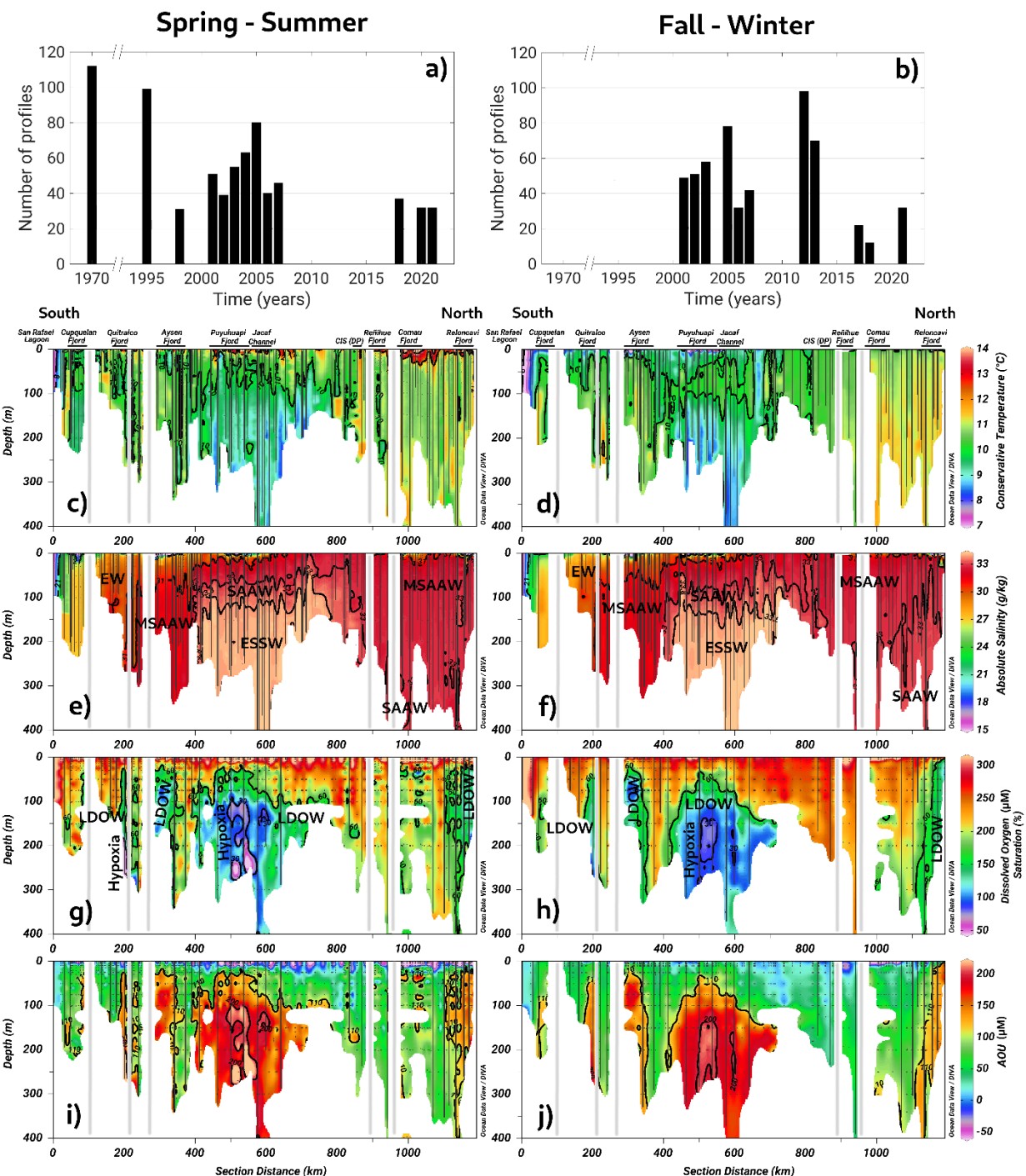

**Figure 2**. (a, b) Time series of CTD profiles used to compute seasonal averages. Long-term seasonal means of (a-b) conservative temperature, (c-d) absolute salinity, and (e-f) dissolved oxygen collected along a vertical section in the northern Patagonian fjord during the fall-winter and spring-summer seasons.

Table 4. Statistical characteristics of the water masses identified in the northern Patagonian fjords.

| Salinity criteria (Sievers and Silva, 2008) | | Spring-Summer Campaigns | | | | Fall-Winter Campaigns | | | |
|---|---|---|---|---|---|---|---|---|---|
| Water masses | Salinity range (g kg$^{-1}$) | Water masses (%) | CT Mean Std[a] | $S_A$ (g kg$^{-1}$) Mean Std | DO (µmol L$^{-1}$) Mean-Std | Water masses (%) | CT Mean Std | $S_A$ (g kg$^{-1}$) Mean Std | DO (µmol L$^{-1}$) Mean-Std |
| EW | 0–31 | 12.64 | 10.29 ±1.1 | 27.92 ±4.7 | 245.25 ±65.8 | 11.92 | 9.70 ±0.8 | 28.75 ±3.4 | 238.02 ±56.8 |
| MSAAW | 31–33 | 60.96 | 10.64 ±0.5 | 32.44 ±0.5 | 197.11 ±46.2 | 54.67 | 10.55 ±0.6 | 32.34 ±0.5 | 231.82 ±37.6 |
| SAAW | 33–33.8 | 15.25 | 10.49 ±0.6 | 33.25 ±0.2 | 196.03 ±41.9 | 22.64 | 10.85 ±0.6 | 33.19 ±0.2 | 208.34 ±46.1 |
| ESSW | 33.8> | 11.15 | 9.12 ±0.4 | 34.11 ±0.1 | 109.57 ±32.6 | 10.77 | 9.27 ±0.5 | 34.13 ±0.1 | 114.77 ±36.0 |

[a]Std: standard deviation.

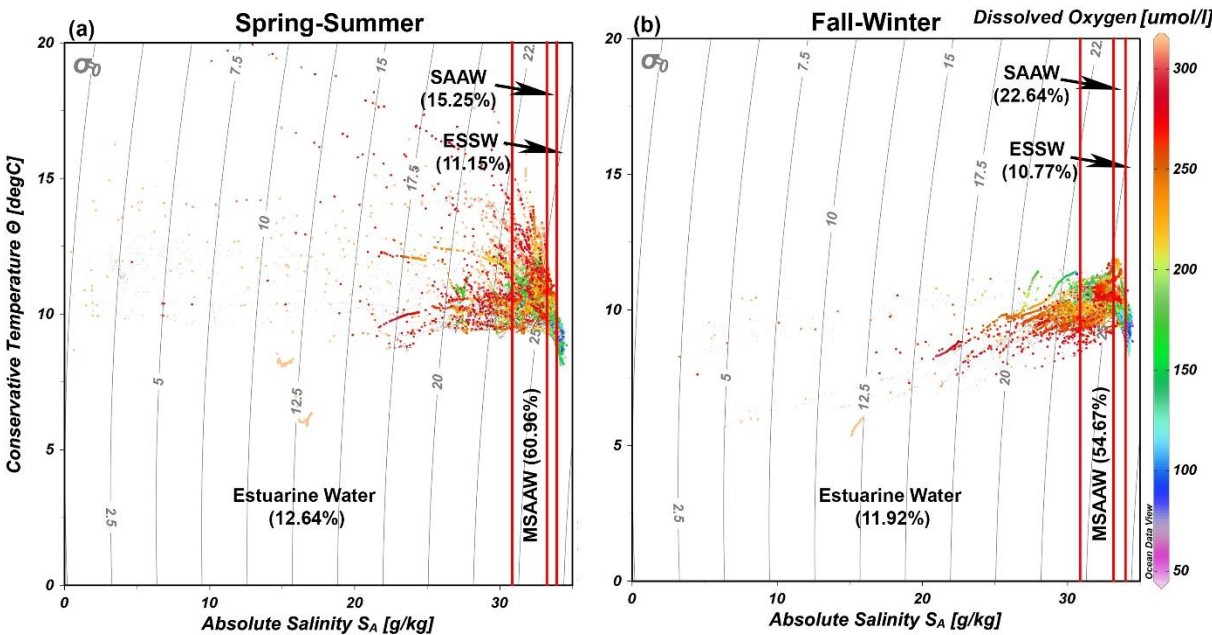

**Figure 3**. Temperature/salinity (TS) diagram with dissolved oxygen records showing the representation of water masses identified during (a) spring-summer and (b) fall-winter seasons in the northern Patagonian fjords. The dataset utilized in the TS diagram is described in Table 1.

As previously shown, the Puyuhuapi Fjord and Jacaf Channel were regions where high-salinity and hypoxic-LDOW waters were registered. In addition, a high concentration of inorganic nutrients was observed in the subsurface layer at a depth of 50 m (Figure 4). In the Puyuhuapi Fjord and the Jacaf Channel, between 100−300 m depth, nitrate (Figure 4a), phosphate (Figure 4c), and silicic acid (Figure 4e) range from 25–30 µM, 2–3 µM, and 30–50 µM, respectively. A second area with high inorganic nutrients was detected in the Reloncaví system, in which the highest absolute values of surface silicic acid were registered during the fall-winter season (e.g., 210 µM). Comparing seasonal concentrations, the fall-winter season showed the highest abundance in the water column.

However, the absolute maximum nitrate (Figure 4b), phosphate (Figure 4d), and silicic acid (Figure 4f) values were recorded in the subsurface layer during the spring and summer seasons.

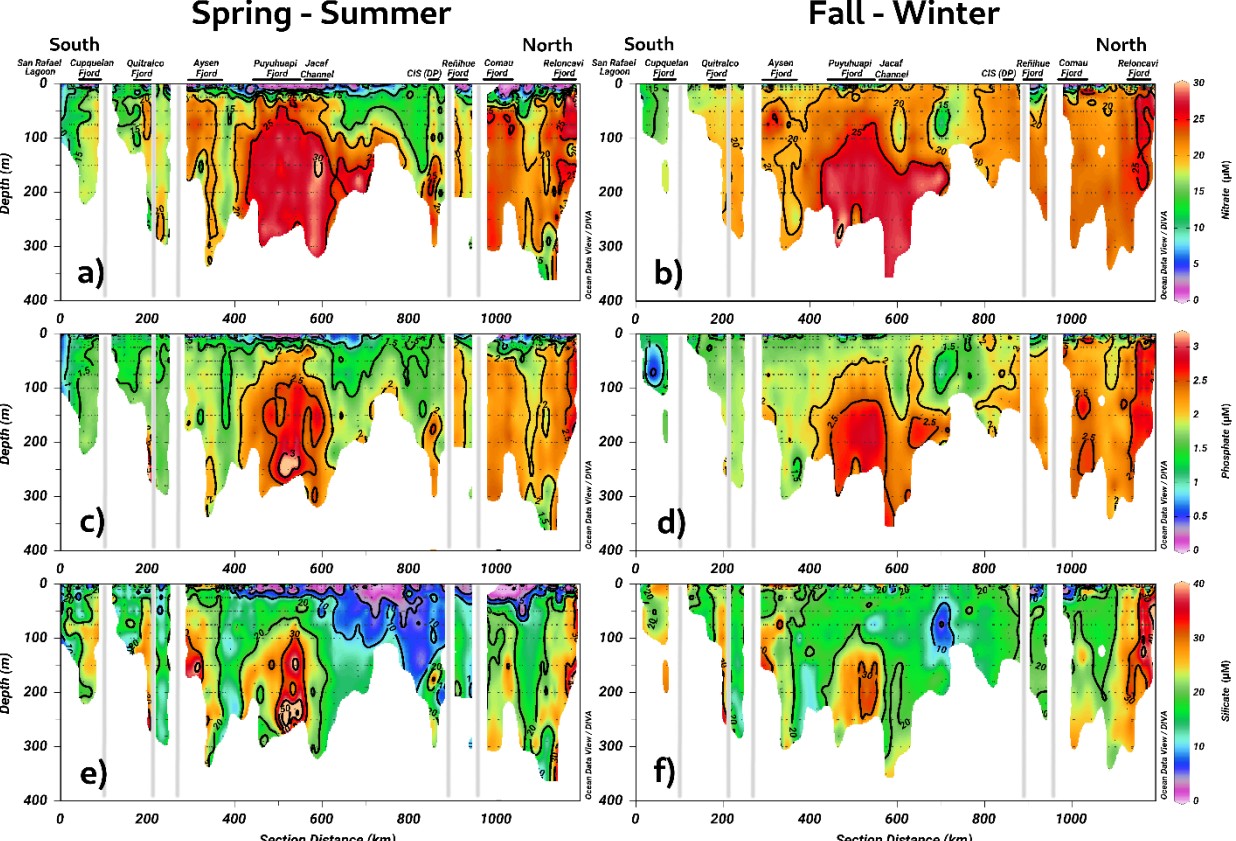

**Figure 4**. Long-term seasonal mean of inorganic nutrients. (a, b) Nitrate, (c, d) phosphate, and (e-f) silicate collected along a vertical section in the northern Patagonian fjord system during fall-winter and spring-summer seasons.

The calculation of the Brunt-Väisälä frequency (BVF), a variable used to identify mixed/stratified regions, evidenced a permanent mixing of the water column in the Chiloé inner sea (BVF<10 cycl h$^{-1}$) (Figure 5a–b). Furthermore, the homogenization of physical (e.g., conservative temperature and absolute salinity) and chemical (e.g., DO oxygen and inorganic nutrients) variables was observed in these regions. On another side, stratified water (between 2 and 10 m depth) was detected during spring-summer inside the fjords, e.g., Cupquelan, Quitralco, Aysén, Puyuhuapi, Comau, and Reloncaví, showing values over BVF>50 cycl h$^{-1}$ (Figure 5a).

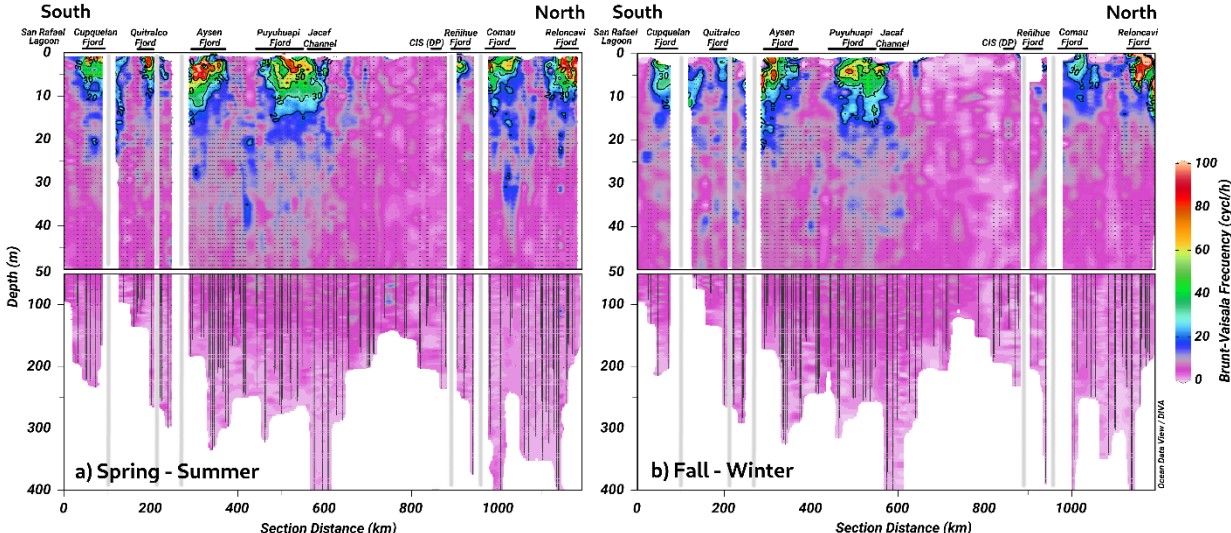

**Figure 5**. The long-term seasonal mean of Brut-Väisälä frequency along a vertical section in the northern Patagonian fjord during the (a) spring-summer, and (b) fall-winter seasons.

3.3. DO and chlorophyll-a features from sporadic oceanographic cruises

The analysis of hypoxia and LDOW conditions in the northern Patagonian fjord system highlighted the presence of two areas with water bodies with these characteristics: the Puyuhuapi-Jacaf and Reloncaví regions. Moreover, Magdalena Sound (44.6° S, 72.9° W), located between the Puyuhuapi Fjord and the Jacaf Channel (Figure 1), showed the shallowest hypoxia over the entire northern Patagonian fjords. At this location, a 30% DO saturation isoline was observed at a depth of 70 m, with the LDOW reaching a depth of approximately 20 m (Figure 6a–c). Additionally, the lowest DO values of the northern Patagonian fjord system (9.36 µmol L$^{-1}$ and 1.6% oxygen saturation) were recorded at the deepest layer (~200 m) of the Quitralco Fjord (45.7° S, 73.3° W) (Figure 6d–f).

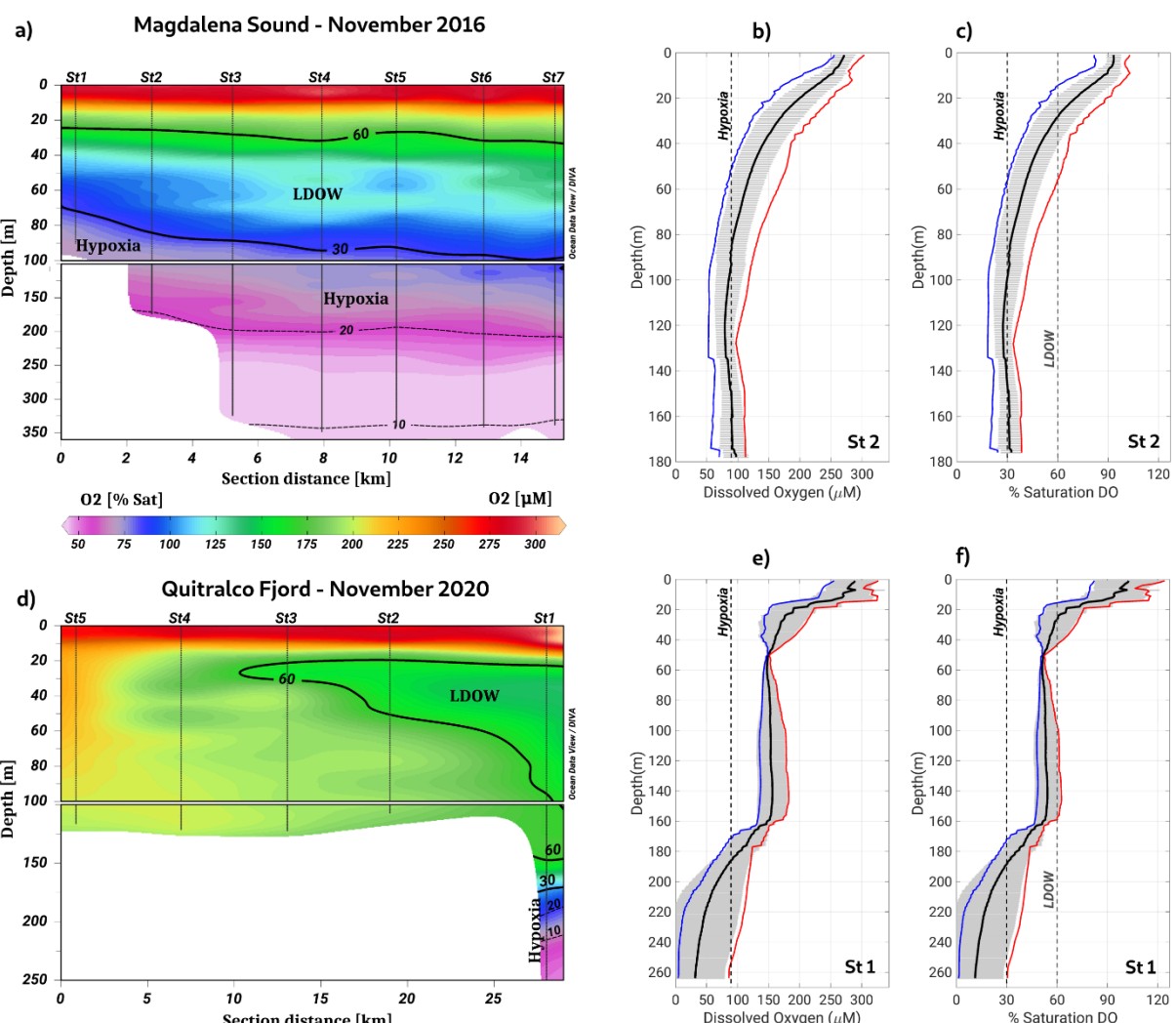

**Figure 6**. Vertical section of dissolved oxygen along (a-c) Magdalena Sound and (d-f) Quitralco Fjord during November 2016 and November 2020, respectively. In (b, c, e, f), red, black, and blue lines represent the absolute maximum, average, and minimum values of DO, respectively. Grey horizontal lines show the standard deviations calculated using 16 profiles for Magdalena Sound and 3 profiles for the Quitralco Fjord.

The chlorophyll-a (Chl-a) data collected during two oceanographic campaigns (Figure 7a) denoted high concentrations of Chl-a in the Reloncaví Fjord, the Puyuhuapi Fjord, and the Jacaf Channel (Figure 7b, 7c) where LDOW and hypoxia conditions were registered. The Chl-a values observed during the winter season in the Reloncaví Fjord were exceptionally high (32.8 mg m$^{-3}$). This observation is scrutinized in the discussion section. Regardless, in general, the winter season of August 2021 showed a deficiency of Chl-a in the southern part of the study area (Figure 7c). The vertical distribution of the Chl-a data described before was obtained through a statistical analysis between the fluorescence data from CTD (every 1-m depth) and the in-situ Chl-a samples from discrete levels (i.e., 0, 5, 10, 15, 25, and 50 m) (Figure 7d, 7e).

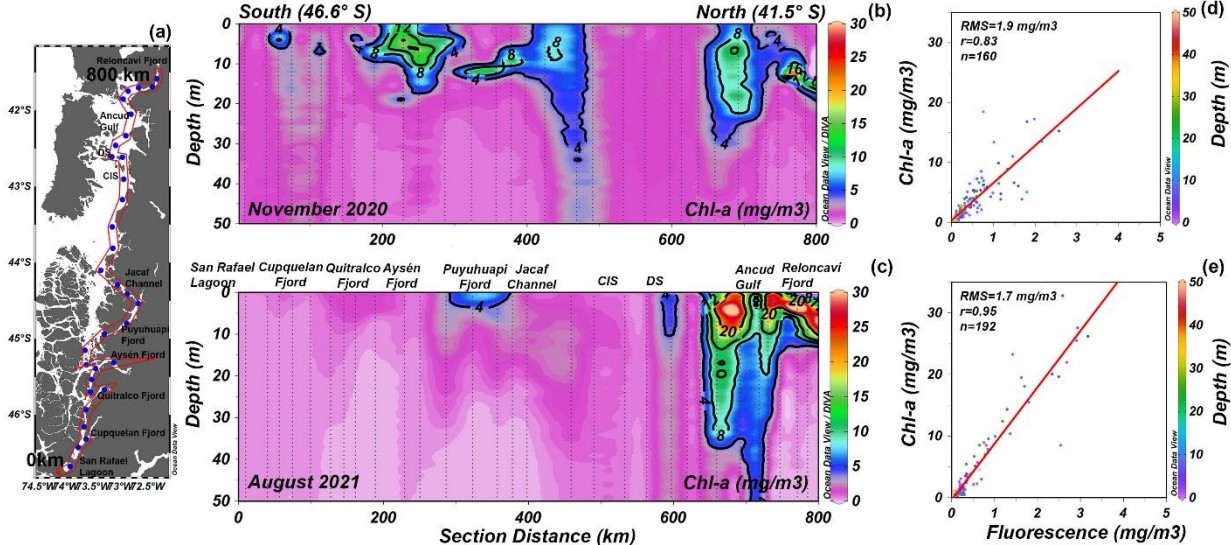

**Figure 7**. Distribution map of the stations. Vertical distribution of chlorophyll-a (Chl-a) obtained during (b) November 2020 and (c) August 2021 in the northern Patagonian fjords. (d and e) Statistical relationship between the fluorescence data from CTD and in-situ Chl-a samples. In (d and e) RMS is the root mean square and n is the number of data points used to obtain the R-values.

3.2. Rates of primary production, community respiration, and bacterial production

The GPP and CR estimates obtained in our sampling campaigns agreed well with the measurements reported in previous studies for the Patagonian fjords during the productive period (Iriarte et al., 2007; Montero et al., 2011; Montero et al., 2017a). The GPP and CR rates showed a weak coupling, with most experiments indicating that less oxygen is produced than consumed (Figure 8a). Thus, most fjords exhibited heterotrophic metabolism (GPP:CR <1) in the surface layer (0–20 m) during the study period (Figure 8b). The highest rates of GPP in terms of oxygen (18.6 to 41.02 mmol $O_2$ m$^{-3}$ d$^{-1}$) were mainly recorded in the upper 5 m of the water column, whereas the lowest values (0 to 17.6 mmol $O_2$ m$^{-3}$ d$^{-1}$) were observed between 10 and 20 m depth (Figure 8c, 8j). CR rates did not show a definite vertical pattern, registering highest (23.3 to 76.3 mmol $O_2$ m$^{-3}$ d$^{-1}$) and lowest (0 to 12.3 mmol $O_2$ m$^{-3}$ d$^{-1}$) values throughout the water column (0–20 m) (Figure 8c, 8j).

Throughout the study period in different fjords, integrated GPP and CR rates (down to 20 m) ranged from 0.1–2.6 g C m$^{-2}$ d$^{-1}$ and from 0.8–6.6 g C m$^{-2}$ d$^{-1}$, respectively, (Figure 9, Table 2). Neither rate showed significant differences ($p$ >0.05) between spring-summer 2020–2021 (Figure 9a) and summer of 2022 (Figure 9b). Although there were no significant differences ($p$ >0.05) between GPP values recorded from different fjords in both study periods (Figure 9a, 9b), Comau Fjord showed the lowest rates of GPP (0.1 to 0.6 g C m$^{-2}$ d$^{-1}$) whereas Puyuhuapi and Reloncaví fjords showed the highest values (1.9 to 2.6 g C m$^{-2}$ d$^{-1}$). In the case of CR, the highest rates were recorded in Puyuhuapi Fjord (2.2 to 6.6 g C m$^{-2}$ d$^{-1}$) and the lowest in Comau Fjord (0.8 to 0.9 g C m$^{-2}$ d$^{-1}$) (Figure 9a, 9b). The phytoplankton composition in the study area showed a similar pattern to the GPP values, highlighting maximum abundances on the surface and minimal abundances at a depth of 20 m (data not shown). In addition, Comau Fjord showed the lowest abundances (66 × 10$^3$ cell L$^{-1}$) whereas Reloncaví Fjord showed the highest concentrations of phytoplankton (1,227 × 10$^3$ cell L$^{-1}$).

Estimated rates of BSP ranged between 0.05 and 0.6 g C m$^{-2}$ d$^{-1}$ within the study area and were positively and significantly correlated with GPP values (Figure 9, Table 2). The percentage of GPP utilized by bacteria ranged from 3% to 56%, except in Comau Fjord, where a higher utilization percentage (151%) was recorded, suggesting that more organic carbon is consumed than is produced locally.

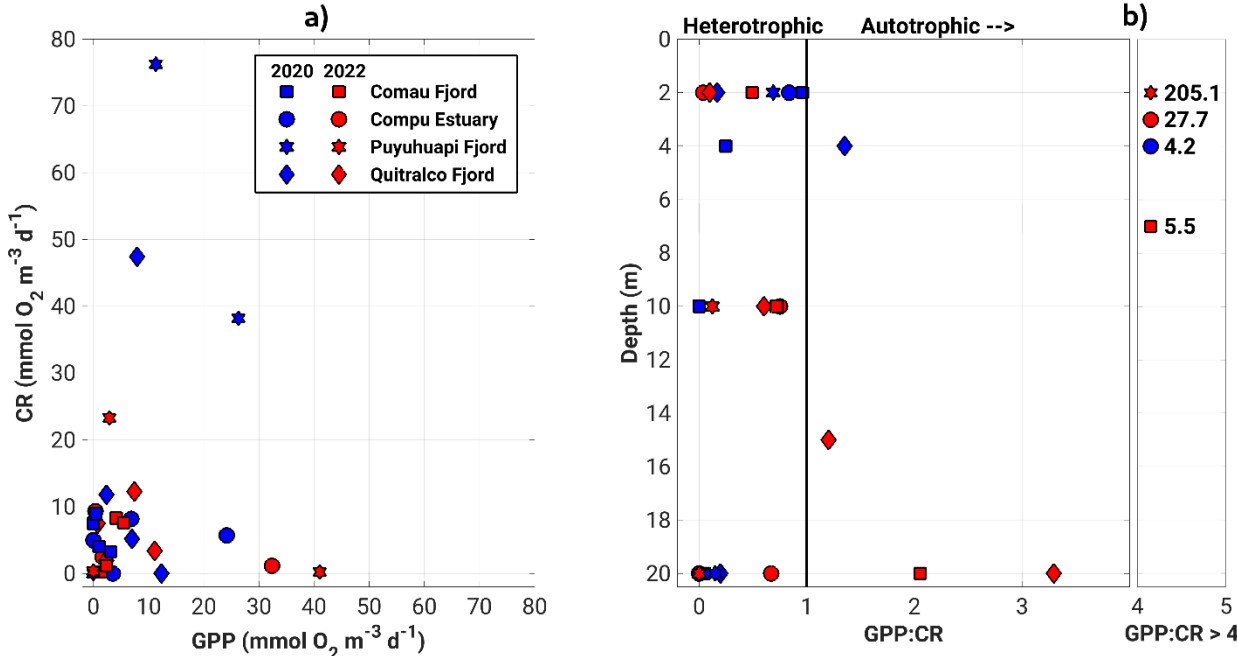

**Figure 8**. (a) Relationship and (b) vertical patterns of all datasets from the GPP and CR. (c−j) represents the vertical distribution of GPP and CR obtained during experiments of primary production on (c−f) spring-summer of 2020−2021, and (g−j) summer of 2022. GPP: Global primary production, and CR: community respiration.

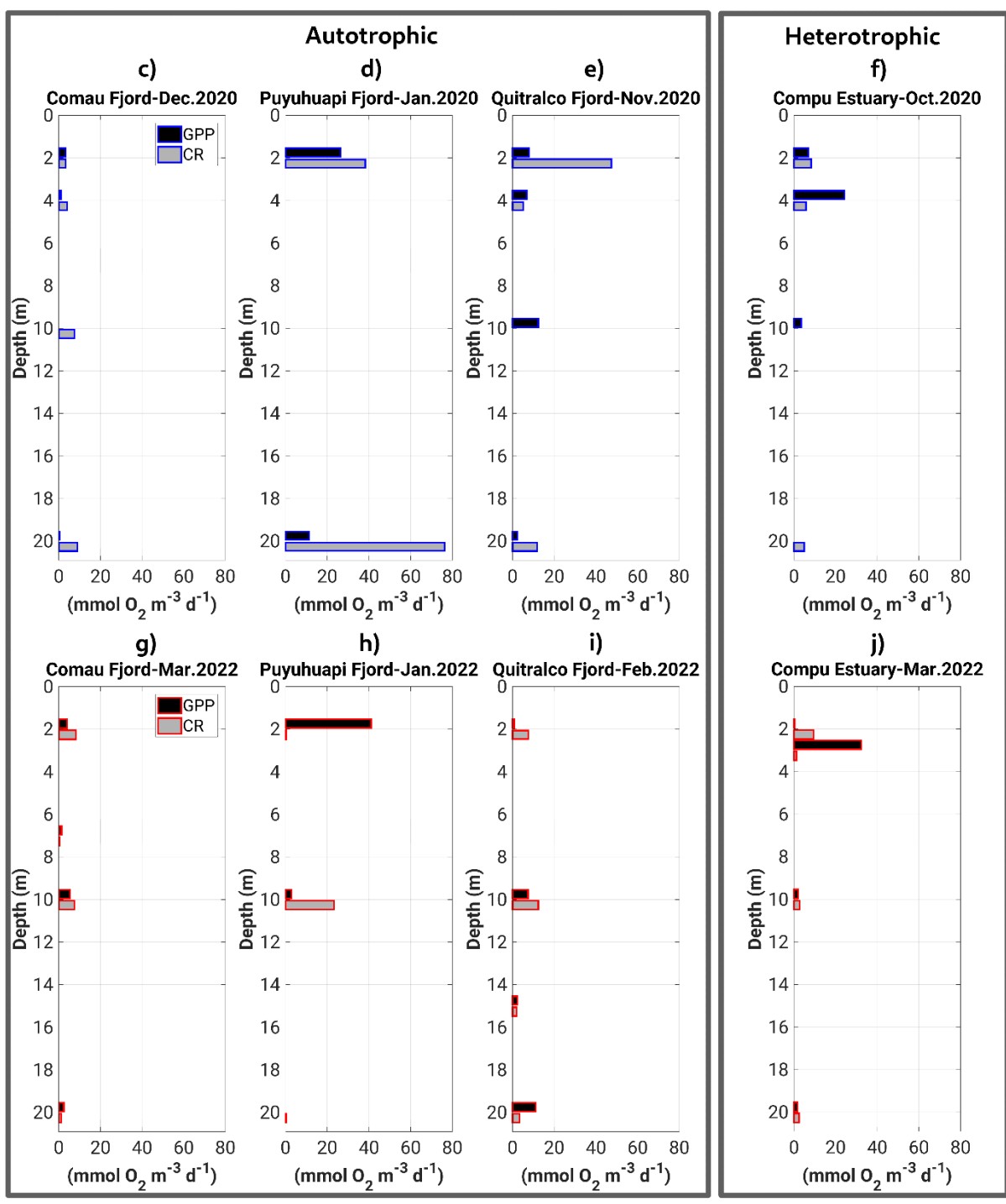

Figure 8. Continued.

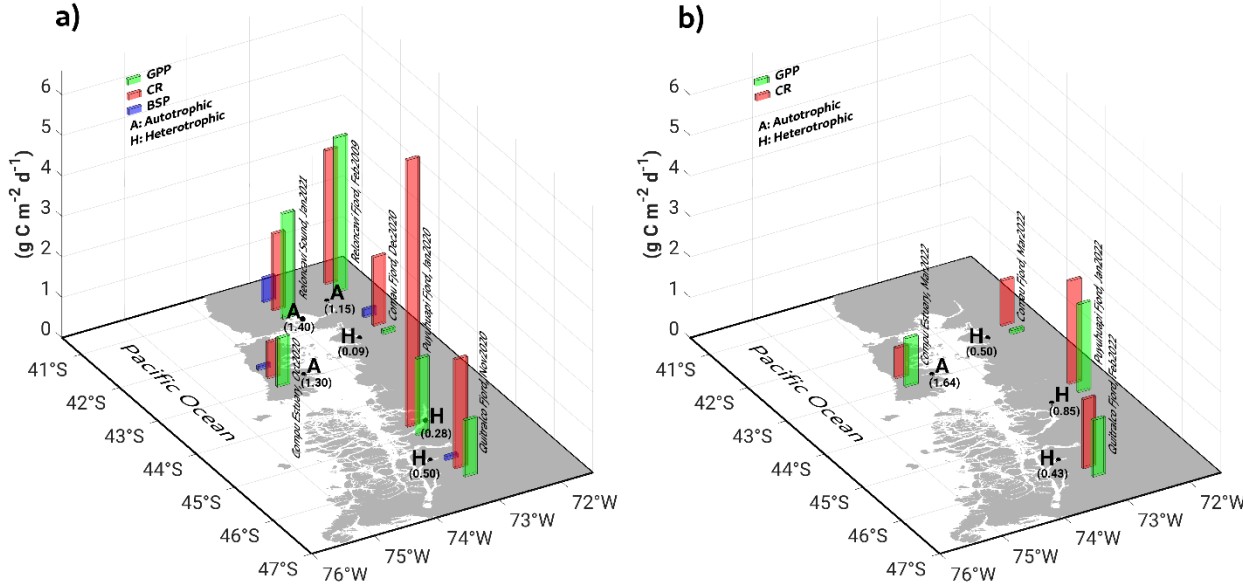

**Figure 9**. *In-situ* experiment of primary production during (a) spring-summer of 2020−2021, and (b) summer of 2022, covering some fjords in the northern Patagonian fjord system. GPP: Global primary production, CR: Community respiration, BSP: Bacterial secondary production. Results shown from Reloncaví Fjord in (a) were extracted from Montero et al., (2011).

### 3.3. Satellite image of organic matter supply to fjords

Satellite images show that freshwater inputs into the sea were accompanied by large amounts of suspended sediments (Figure 10), possibly acting as a source of allochthonous organic matter, favoring the heterotrophic metabolism reported in the previous section. On May 9, 2017, a large amount of suspended matter flowed into the Reloncaví Sound from its fjord, mainly dominated by three rivers: Petrohue, Cochamo, and Puelo (Figure 10b). In the southeast part of the Reloncaví Sound, the observed high concentration of sediments was due to the presence of a diatom bloom (evidenced by greenish waters in the southeast part of the Reloncaví Sound), mainly formed by *Skeletonema costatum*, which attained concentrations of more than 5 million cells L$^{-1}$ within the fjord (Figure not shown). This bloom affected both the eastern part of the Reloncaví Sound and the Reloncaví Fjord. While the amount of suspended matter visible in the Sentinel-2 images was generally homogeneously high along the fjords e.g., on June 9, 2022, in the Quitralco Fjord (Figure 10c), and May 9, 2017, in Reloncaví, (Figure 10b). Some examples of this phenomenon, such as the one on April 6, 2018, on Comau Fjord (Figure 9d), show a very heterogeneous distribution of suspended sediments within the fjord, suggesting the influence of the currents in the distribution of waters rich in sediment. On this date, waters flowing from the river Blanco in the northern part of the fjord's mouth were whitish in color, indicating carbonate-rich sediments. Water flowing into this fjord from the south, principally from the Lloncochaígua and Vodudahue rivers, has a more brownish color, indicating a higher quantity of organic matter.

At the Quitralco Fjord (Figure 10c), suspended sediments from Rio de los Huemules (at 73.30° E 45.54° S, the brown riverbed is clearly visible in the natural color insert) appeared to propagate within the fjord, affecting the

water clarity along the entire fjord. Inspection on other dates (not shown) revealed high variability in the direction of the river outflow, sometimes entering the fjord (and staying there for extended periods), and sometimes being directed towards the south. Finally, the Puyuhuapi Fjord also registered a high signal from suspended sediments due to river discharge; for example, the Vestiquero River was located at the fjord head (Figure 10e). Other rivers, such as the Cisnes, Marta, and Uspallante, could also influence the supply of allochthonous organic matter to the Puyuhuapi Fjord, contributing to a decrease in DO and hypoxia. However, the cloud cover does not allow for better satellite images.

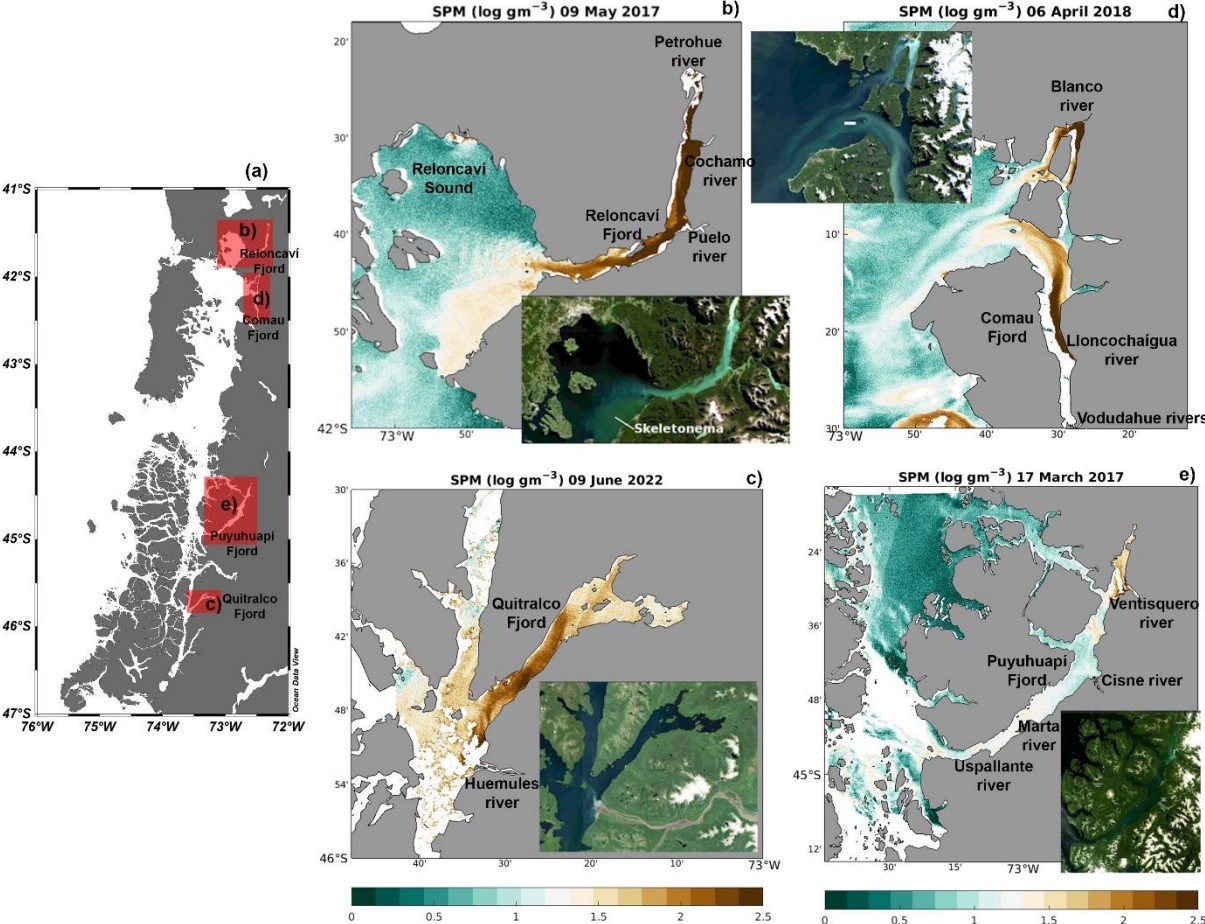

**Figure 10**. (a) Map showing the study sites. Suspended particulate matter as measured by Sentinel 2 on (b) Reloncaví Sound, (c) Quitralco Fjord, (d) Comau Fjord, and (e) Puyuhuapi Fjord.

3.4. Biogeochemical variables

We observed differential signals of SPM (Figure 11b) along a latitudinal transect conducted from north to south in the northern Patagonia fjords (Figure 11a). Specifically, significant increases in SPM were registered at the surface layer of the mouths of the Comau and Aysén fjords. Furthermore, in the southern region, the SPM levels were high at the subsurface layer (50−150 m) from San Rafael Lagoon to Jacaf Channel, with maximum values exceeding 4 mg L$^{-1}$, except in Quitralco Fjord, which showed values close to 2 mg L$^{-1}$. Similar conditions were

560 observed in the transition zone between the Puyuhuapi Fjord and Jacaf Channel. In the central region, within the Chiloé Inner Sea, SPM values were minimal. In the northern area, a small subsurface core near Comau Fjord and the Reloncaví system exhibited an increase in SPM. Along the water column, characteristic cores were found in the inner channels south of the study area, between 45º S and 47º S. Other deep but less intense cores (<3 mg L$^{-1}$) were located in the Inner Sea of Chiloé and Reloncaví system.

The percentage of allochthonous organic carbon (%POC$_{allo}$) exhibited an irregular distribution in the southern area, with surface minimums under 30% at the mouths of the Quitralco, Aysén, and Puyuhuapi fjords (Figure 11c). In the northern area, the Comau Fjord and Reloncaví Sound presented low %POC$_{allo}$ values. Along the water column, %POC$_{allo}$ appeared to increase with depth, reaching values of approximately 70%, except in the Inner Sea of Chiloé, where it was under 50%. The C:N ratio showed more homogeneous values at the surface layer 570 (C:N<8) (Figure 11d) but increased from a 50-m depth until the seabed (C:N>12), especially in the region between the San Rafael Lagoon and the Jacaf Channel, coinciding with the high SPM region presented in Figure 11b. In the Chiloé Inner Sea, the C:N ratio decreased along the water column (from surface to the seabed).

      The isotopic signal of the carbon ($\delta^{13}$C) increased in the same sites where %POC$_{allo}$ decreased, with values ranging from –25‰ to –15‰ (Figure 11e). At the subsurface layer, $\delta^{13}$C signal became lighter than that in the 575 surface. An exception was observed in the Chiloé Inner Sea and the transition zone between the Aysén Fjord and San Rafael Lagoon, where the C:N ratio coincidentally decreased. In the region extending from the southern Chiloé Inner Sea to Aysén Fjord, elevated levels of isotopic signal of nitrogen ($\delta^{15}$N) enrichment were observed, primarily at the surface and subsurface levels (Figure 11f). Notably, there was a significant increase in nitrogen values, with some exceeding 9‰; particularly, in the Chiloé Inner Sea, the enrichment extended to depths of 150−200 m.

Notable observations were made regarding the concentrations of CDOM in the Patagonian fjords region, with high concentrations throughout the water columns in the north area, spanning from the Chiloé Inner Sea to the Reloncaví system (Figure 11g). Conversely, in the southern area, CDOM concentrations declined, with minimum values as low as 20 µg L$^{-1}$ of specific UV absorbance between the Jacaf Channel and San Rafael Lagoon. Despite this overall decrease, localized increases in CDOM concentrations were identified in certain fjords, such as Aysén, Quitralco, 585 and the San Rafael Lagoon. These subsurface concentration spikes highlight the spatial variability of CDOM distribution within the fjord system. Moreover, these findings provide valuable insights into the spatial distribution patterns of CDOM in the Patagonian fjords, indicating higher concentrations in the northern region and a decline towards the southern end of the study area. The localized increases in CDOM concentrations in specific fjords suggest the influence of local processes or inputs contributing to the observed spatial heterogeneity.

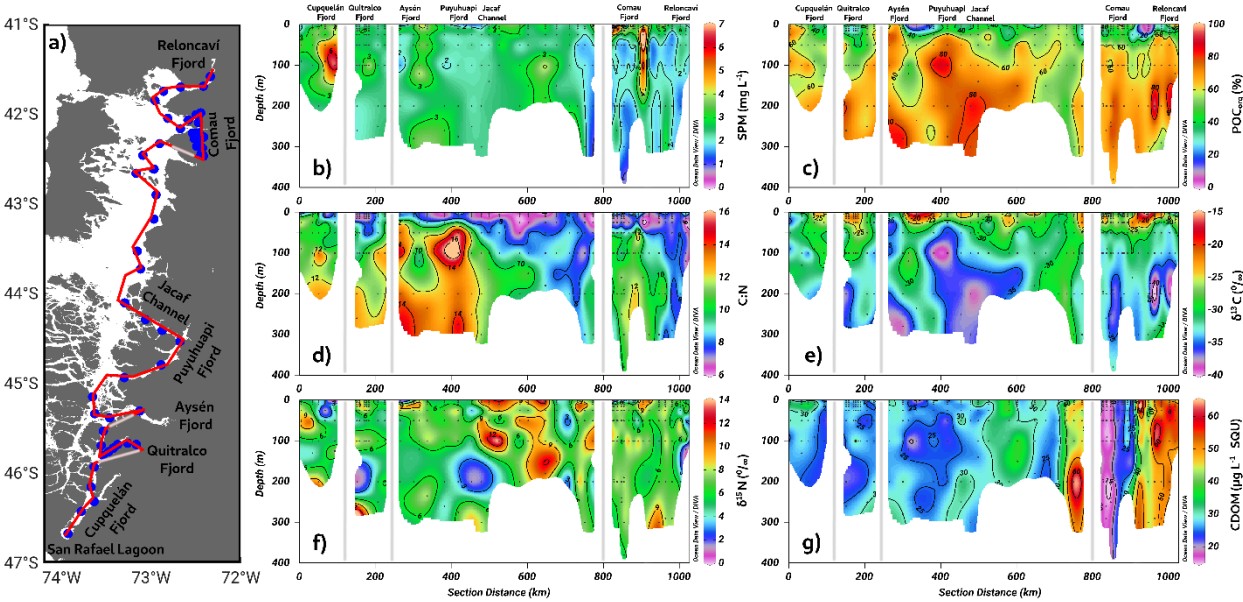

**Figure 11**. (a) Study area showing transect (red color) used to biogeochemical study case carried out in November 2020 during Expeditions PN-I IFOP. (b) Suspended particular material (SPM), (c) allochthonous carbon percentage (POC$_{allo}$), (d) Carbon and nitrogen ratio (C:N), Isotopic signal of (e) $\delta^{13}$C and (f) $\delta^{15}$N, and (g) Chromophoric Dissolved Organic Matter (CDOM).

### 3.5. Circulation regime from ADCP measurements

The low-frequency deep circulation (> 40 m depth) in the inner sea of the region showed a marked east–west regional gradient in the along-channel current amplitude (Figure 12). The Corcovado currents were dominated by an eastward flow with a maximum intensity of approximately 30 cm s$^{-1}$ (Figure 12a-b), whereas at the mouth of the Puyuhuapi Fjord (Figure 12c-d), the maximum current was approximately 10 cm s$^{-1}$.

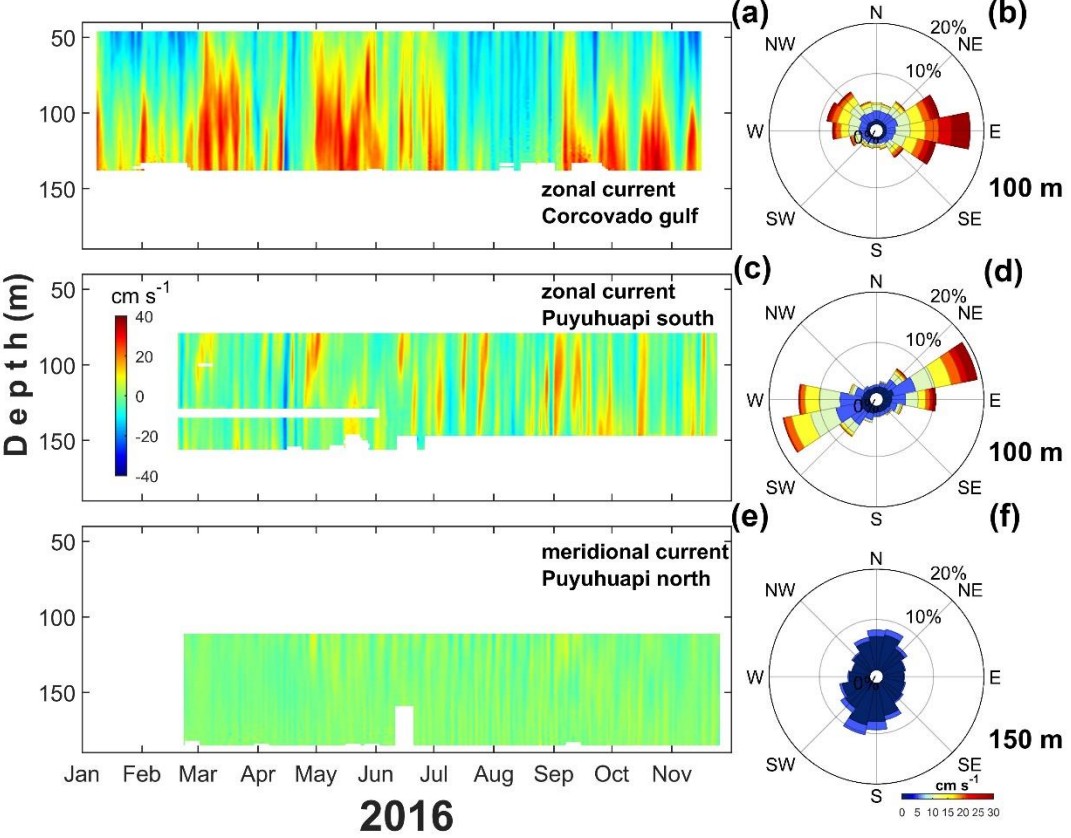

**Figure 12**. Subtidal deep circulation regimen obtained with ADCP performed in the (a) Corcovado Gulf and in (c, e) southern-northern zone of the Puyuhuapi Fjord from January to November 2016. (b, d, and f) Marine current rose obtained at 100 m (Corcovado Gulf), 100 m (Puyuhuapi south), and 150 m (Puyuhuapi north) depths.

The Corcovado currents showed a marked pattern in the 40–70 m layer, which was dominated by westward currents, and short-duration events of approximately 20 cm s$^{-1}$ were observed throughout the entire time series. This strong westward flow seems to be associated with the austral spring-summer seasons (January–March and November–December 2016); however, a weak (~5 cm s$^{-1}$) and longer (July–October 2016) westward flow was observed during the austral winter-autumn period. In the layer between 70 and 130 m, eastward (and strong) currents dominated, and there were periods of one to two months with eastward flow during March–April, May–July, and September–December 2016 with currents > 20 cm s$^{-1}$. The layer between 100 and 130 m showed maximum currents of approximately 40 cm s$^{-1}$ at the end of October 2016 (Figure 12a).

Puyuhuapi Fjord currents were obtained at the southern mouth (South Puyuhuapi, Figure 12c-d) and middle fjord (North Puyuhuapi, Figure 12e-f. At the mouth of the fjord, low-frequency currents showed short pulses that covered most of the 70–160 m layer; typically, those pulses were eastward without an evident seasonal pattern. A unique low-frequency westward event was observed in mid-April 2016, which was also present in the Corcovado Gulf. In general, the currents in southern Puyuhuapi were weak (< 10 cm s$^{-1}$); however, during September and October 2016, eastward events of 20 cm s$^{-1}$ were observed. In northern Puyuhuapi, most currents were weak (< 10

cm s$^{-1}$) without a temporally or spatially evident pattern, and the flow tended to be northward and intensified between May and September 2016.

3.6 Circulation regimen oceanographic model

Deep currents, defined as subtidal or residual currents (average of 50–300 m) over the three years of simulation, have a relatively large spatial variation (Figure 13a, 13b). Consequently, the model could correlate the most energetic zones in the outer channels with those of lower energy inside the fjords. In general, the exchanges between the fjords and ocean registered high average velocities, that is, the Guafo mouth, Moraleda Channel,

Corcovado Gulf, and Desertores Pass, with net current velocities ranging from 10 to 20 cm s$^{-1}$. In contrast, in the fjords, the average velocity of the deep flow was less than 5 cm s$^{-1}$. The predominant flow was generally towards the interior of the fjords (Figure 13a, 13b).

The flushing/residence time calculated for each fjord showed that the sector towards the head of Puyuhuapi Fjord registered the longest retention time (>200 days). In comparison, the shortest times were found in the

Reloncaví Sound−Fjord (60−90 days), except for the Reloncaví Fjord head, which reached values close to 100 days (Figure 13c and 13d). Comau Fjord showed relatively high values, which increased from north to south. In the central part of Comau Fjord, the residence time ranged from 90 to 120 days, whereas at the fjord head, it ranged from 150 to 190 days. Meanwhile, the Aysén Fjord showed residence time values lower than 120 days, which were higher towards the fjord head, whereas the Quitralco Fjord reported 150 days at the fjord head and 100 days at its

central part. Finally, the Cupquelán Fjord exhibited greater homogeneity throughout the fjord, with relatively high values ranging from 120 to 150 days. Most studies reported a spatial gradient of water retention, with the highest values found near the fjord's head.

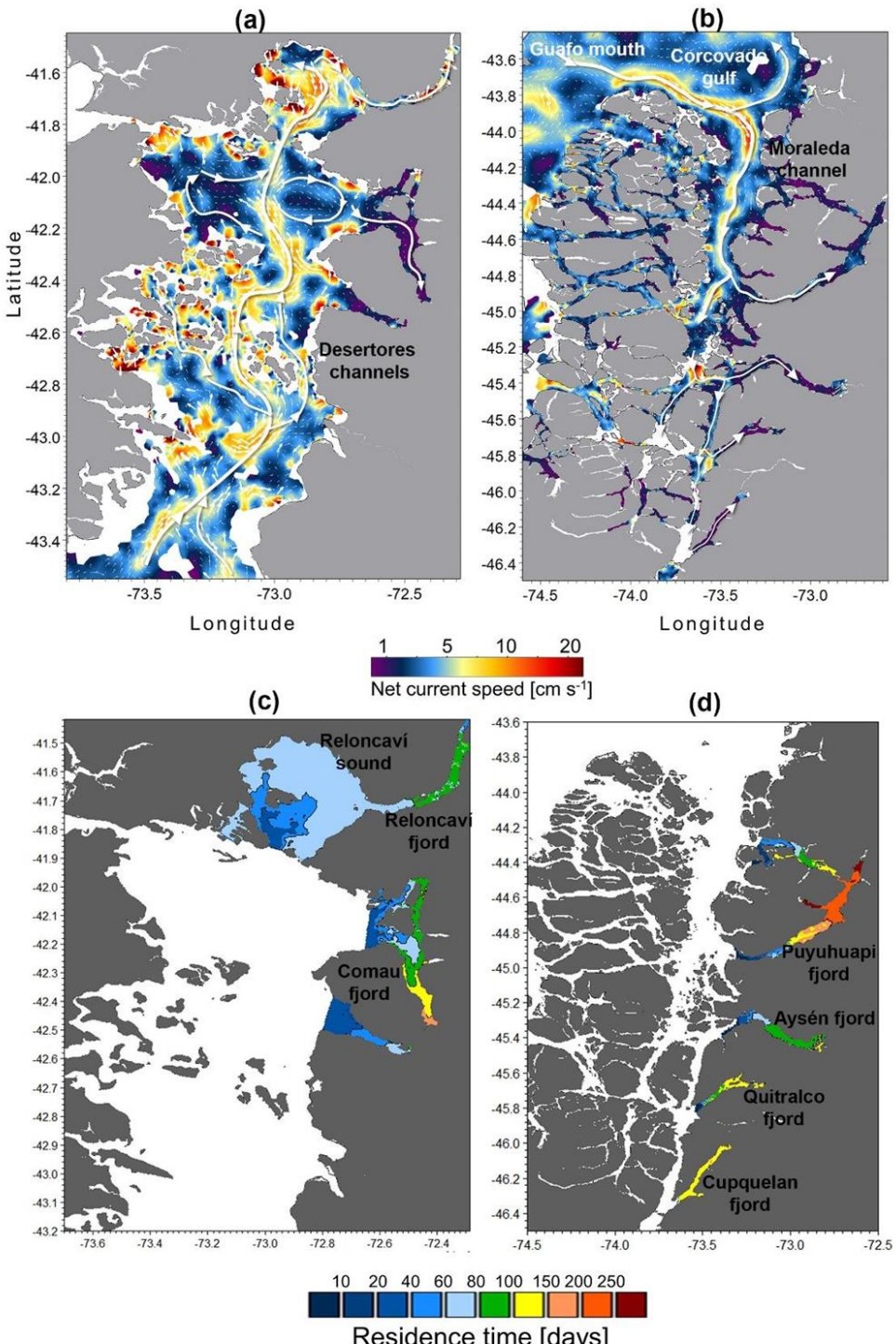

**Figure 13**. (a, b) Map showing the deep currents regime in northern Patagonian fjords. The deep current was defined as subtidal currents calculated as the average of currents from 50−300 m depth. (c, d) The residence time was calculated for each fjord of the northern Patagonian. (a-d) Data were obtained from the hydrodynamic model MIKE 3 FM developed by IFOP.

## 4. Discussion

Oceanographic research in the Patagonian fjords started in 1995 with CIMAR-fjords project, supported by the Chilean Navy. As an oceanographic sampling strategy, Patagonia was divided into three zones, e.g., northern (41°–47° S), central (47°–53° S), and southern (53°–55.8° S) Patagonia, and the frequency of the cruises were primarily seasonal (Sievers, 2008). Moreover, some oceanographic cruises, such as the Hudson expedition in 1970, occurred in this zone (Table 1). The data collected through the CIMAR projects allowed us to understand the main physical (water mass quantification) and chemical (e.g., dissolved oxygen and inorganic nutrients) processes impacting this fjord system, such as the hypoxic conditions due to the influence of the ESSW (Sievers, 2008; Sievers and Silva, 2008; Silva, 2008; Silva and Vargas, 2014; Schneider et al., 2014; Pérez-Santos et a., 2014). Furthermore, some studies about biological processes that include the consumption of DO, such as respiration and the remineralization of organic matter, have also contributed to understanding the processes that favor hypoxia and DO depletion (Montero et al., 2011; 2017; Iriarte et al., 2018). Additionally, the development of hydrodynamic models helped to determine the areas where the water age was significantly high, for example, the Puyuhuapi Fjord (Pinilla et al., 2020), and where stronger mixing stratification occurred (Ruiz et al., 2021). This study scrutinized all processes contributing to the presence of water under hypoxic conditions and LDOW in the northern Patagonian fjords, as discussed in the next section.

4.1 Water masses proprieties and biogeochemical processes in northern Patagonian Fjords

Recent and historical hydrographic data collected from the Patagonian fjords, especially in the northern area, confirmed the presence of analogous water masses detected previously by the Hudson oceanographic cruise in 1970. EW, MSAAW, SAAW, and ESSW were recorded from the surface layer to the bottom (to a depth of approximately 500 m) (Figure 2 and Figure 3). Similar patterns were recognized in terms of the vertical-horizontal distributions, highlighting the location of the salty-hypoxic-LDOW and nutrient-rich waters associated with the ESSW in the Jacaf Channel and Puyuhuapi Fjord (Figure 2 and 4). This water mass enters Patagonia by the Guafo mouth and moves south throughout the Moraleda Channel to end at the Puyuhuapi Fjord and Jacaf Channel (Linford et al., 2023). On the other hand, the EW dominated in the southern area due to freshwater input from the San Rafael Lagoon and rivers discharge, which is characterized by oxygenated waters due to its high solubility. Finally, SAAW and MSAAW covered the entire study area (Figure 2), as reported by Siever and Silva (2008), Schneider et al., (2014), Pérez-Santos et al., (2014), and Pinilla et al., (2020).

The absence of stratification was observed throughout the year in the Chiloé Inner Sea (Figure 5). In this area, both physical and biogeochemical variables showed homogeneous values from the surface layer to the bottom (Figure 2 and Figure 4), indicating the presence of MSAAW (originating from mixing the SAAW with the EW). Furthermore, the calculation of the buoyancy frequency and potential energy anomalies using the output from the oceanographic model demonstrated that more robust mixing occurred in the Chiloé Inner Sea (Ruiz et al., 2021). This explains the importance of this area for DO ventilation and redistribution in the northern Patagonian fjords.

Water masses in northern Patagonia have unique physicochemical properties (Sievers and Silva, 2008; Silva, 2008; Silva and Vargas, 2014; Schneider et al., 2014; Pérez-Santos et al., 2014). In the case of northern

Patagonian fjords, EW is rich in silicic acid because of the riverine supply (Silva and Vargas, 2014), with characteristic pulses at the surface layer that changes throughout the year due to the organisms' consumption (Montero et al., 2011), as observed in northern Patagonia, including the Chiloé Inner Sea (Figure 4).

The SAAW is strongly connected with the estuarine water masses (MSAAW and EW), and the ESSW interacts with it, as shown in this study (Figures 2 and 3). Organic matter degradation processes occurred inside the fjords, mainly in terms of nitrogenous compounds being remineralized. These processes increase the nutrient pool that is originally transported by oceanic water and are responsible for the high accumulation of nitrate and phosphate registered in the subsurface layer of the Puyuhuapi Fjord and Jacaf Channel (Figure 4).

Inside the fjords, where low salinity from high river discharge dominated, a persistent halocline/pycnocline/stratified water column developed; therefore, deep-water ventilation by vertical mixing and diffusion with the oxic surface layer was limited (Figures 2–5). These conditions favored the occurrence of LDOW zones in the fjords, where stratification and particulate organic matter input were higher than at the mouths (Figure 10). Thus, DO consumption was favored owing to organic matter degradation in the Reloncaví, Comau, Puyuhuapi,

Aysén, and Quitralco fjords (Figure 2), as was reported for the Milford Sound in New Zealand (Stanton, 1984), Framvaren Fjord in Norway (Yao and Millero, 1995), Lower St. Lawrence Estuary in Canada (Lefort, 2012), and Rivers Inlet in British Columbia (Jackson et al., 2022).

A latitudinal transect from north to south in the study area revealed distinct signals of SPM within the fjords (Figure 11b). Although SPM concentrations generally fell within expected ranges, there was significant

variability, with values ranging from 0.5 to 500 mg L$^{-1}$ (Muñoz et al., 2014; Pryer et al., 2020). A noteworthy pattern of increased SPM levels below the surface was observed at the fjord mouths, particularly in the southern part of the study area. This finding suggests that fjord outlets play a crucial role in particulate organic matter accumulation, potentially that originating from terrestrial sources. Furthermore, the high SPM values observed in the southern region indicate substantial contributions of allochthonous material, which may be influenced by factors such as river

discharge, sediment resuspension, and glacial activity (Gonzalez et al., 2010). A more detailed analysis of SPM distribution revealed irregular patterns in the percentage of allochthonous organic matter (%POC$_{allo}$). At the mouths of the Quitralco, Aysén, and Puyuhuapi fjords, a subsurface minimum indicated relatively low terrestrial organic matter inputs (Gonzalez et al., 2010). The low-level input of organic matter could be attributed to limited river contributions or the autochthonous organic material production rate coinciding with nutrient depletion and increased

DO (Holding et al., 2017). In contrast, areas such as the Jacaf Channel and the Chiloé Inner Sea exhibited higher contributions of %POC$_{allo}$, indicating a more significant influence of allochthonous sources. Further, the increase in %POC$_{allo}$ with depth highlights the importance of vertical transport and sedimentation processes in determining the distribution of organic matter throughout the water column (Figure 11c). Considering that carbon fluxes and the functioning of the fjord system are highly variable depending on the season of the year (Gonzalez et al., 2010), these

could potentially affect both the trophic webs and carbon export, which are more independent of local primary production than coastal areas.

The C:N ratio exhibited spatial variability, with more homogeneous values in subsurface layers shallower than 50 m (Figure 11d). On average, the bulk particulate organic carbon and nitrogen concentrations maintained a

C:N ratio of 8.5:1, exceeding the Redfield ratio of 6.6:1 typically observed in phytoplankton biomass within the marine realm (Redfield, 1958). Notably, the southern region exhibited higher C:N ratios, coinciding with areas with elevated SPM levels and increased %POC$_{allo}$. While these higher C:N ratios may be intuitively attributed to terrestrial sources with a higher carbon content relative to nitrogen, a more plausible explanation may be the increasing trend of the C:N ratio with depth. The observed C:N ratios suggest the influence of microbial remineralization processes and the preferential removal of nitrogen-rich organic matter (Taucher et al., 2020). The increase of the C:N ratio with depth likely reflects the consumption and degradation of sinking particles by zooplankton and heterotrophic bacteria, resulting in the preferential remineralization of nitrogen (Figure 3) over carbon and the subsequent elevation of the C:N ratio in sinking particles (Tomas et al., 1999; Schneider et al., 2003). The spatial gradients of $\delta^{13}C$ have been previously documented in estuarine and continental shelf waters in Patagonian fjords (Silva and Vargas, 2014 Gonzalez et al., 2010; Iriarte et al., 2018), but there is no evidence of gradient of nitrogen isotopic signals prior to this study. Nevertheless, the pronounced phytoplankton bloom resulted in an enrichment of $\delta^{13}C$ in SPM (Figure 11), similar to observations made in other fjords around the world (Remeikaite-Nikiene et al., 2017). The seasonal distribution of Chl-a evidenced the phytoplankton activity along the northern Patagonian fjords, especially during spring, 2020 (Figure 7). However, in winter of 2021, a phytoplankton bloom was registered in the Reloncaví Fjord and Ancud Gulf. Pérez-Santos et al. (2021) reported the occurrence of high Chl-a and an increase of phytoplankton in the Reloncaví Sound during late winters (August and September, 2017−2020) due changes in the water column conditions, e.g., stratification started when salinity decreased owing to the increase of river discharge mainly caused by ice melting. The stabilization of the water column is one of the main factors triggering primary production in the Patagonian fjords (Pérez-Santos et al., 2021). The isotopic carbon signals revealed an increase in $\delta^{13}C$ values where %POC$_{allo}$ decreased, indicating a shift towards a more depleted isotopic signature. This suggests the predominance of marine-derived organic carbon in areas with lower inputs of allochthonous material (Kumar et al., 2016). Moreover, the lighter carbon isotopic signals observed with depth suggest isotopic fractionation processes associated with remineralization and the preferential utilization of more labile organic carbon by heterotrophic bacteria (Muñoz et al., 2014; Kuliński et al., 2014).

Elevated levels of nitrogen enrichment were primarily observed in the surface layer of the region extending from the southern Chiloé Inner Sea to the Aysén Fjord, indicating the influence of nitrogen inputs from sources such as river discharge, atmospheric deposition, and biological activity (Velinsky and Fogel, 1999). The significant increase in nitrogen values, particularly in the Chiloé Inner Sea, suggests the presence of enhanced nitrogen cycling and retention processes within this fjord system. This increase could result from the degradation of organic nitrogen with depth, where isotopically light $\delta^{15}N$ is preferentially released from the organic matrix, leaving behind isotopically heavier residual $\delta^{15}N$ (Saino and Hattori, 1980, 1987; Altabet, 1988). Similarly, the enrichment of $\delta^{15}N$ in SPM suggests that primary production mainly originates from autochthonous sources in some regions of the fjords (Figure 11). The horizontal–vertical distribution of the isotopic signal of $\delta^{13}C$ and $\delta^{15}N$ showed substantial variability in the northern Patagonian fjords. Our results strongly suggest that this variability was driven by biological processes, as was demonstrated by Savoye et al. (2003). In this work, the authors found that the co-

760 variation of $\delta^{13}C$ and $\delta^{15}N$ reflect the seasonal succession of phytoplankton species due to seasonal changes in abiotic factors, such as temperature and nutrient concentration.

Silva and Vargas (2014) identified an LDOW zone near the fjord heads (<178 µmoL L$^{-1}$), attributed to a significant influx of allochthonous particulate organic matter from local rivers (Figure 10). Nevertheless, the observed organic material mixture resulted in relatively lighter isotopic values of particulate organic matter than 765 those expected for an autochthonous organic matter pool (Figure 7). Regardless, the input of organic matter (autochthonous or allochthonous) to the deep layer increased DO consumption through respiration, overwhelming the oxygen supply from horizontal advection (Figures 8 and 9).

The distribution of CDOM displayed distinct patterns within the Patagonian fjords (Figure 11). Higher CDOM concentrations were observed in the northern region and were attributed to factors such as greater terrestrial 770 inputs, increased microbial activity, and longer residence times (Figure 13) of CDOM in these fjords (Mannino et al., 2008). In contrast, the southern region exhibited lower CDOM concentrations, indicating a decrease in dissolved terrestrial inputs and potentially more efficient removal processes. Localized increases in CDOM concentrations within specific fjords suggest the influence of local processes, such as freshwater inputs, glacial melting, or biological production, contributing to the heterogeneity of CDOM spatial distribution (Mannino et al., 2008).

4.2. Mechanisms driving hypoxia and LDOW in northern Patagonian fjords

As discussed in the previous section, one of the principal mechanisms involved in the origin of hypoxic conditions in some areas of the northern Patagonian fjords was attributed to the advection of the ESSW from the Pacific Ocean to Patagonia. Recently, Linford et al. (2023) demonstrated the poleward advection of water with 780 lower DO values by the ESSW that was transported by the Perú Chile Undercurrent (PCUC). The passage of this water mass into the northern Patagonian fjord system was detected in a DO time series from a mooring system installed at the Guafo mouth. Moreover, as shown in Figure 2, extreme hypoxic conditions were registered in the Quitralco Fjord, where ESSW was not delivered (Figure 6d-f).

In the Quitralco Fjord, the DO concentration was mainly governed by the biological processes of oxygen 785 production and consumption during photosynthesis and respiration, respectively. The primary production experiments conducted in this fjord (Figure 8 and Figure 9) showed higher values of community respiration than the primary production of DO (Table 2). In addition, the oceanographic model showed a less intense deep net current, allowing a residence time of water between 100−150 days (Figure 13). Furthermore, satellite images provide evidence of highly suspended sediments along the Quitralco Fjord due to river discharge (Figure 10). Regarding 790 Magdalena Sound, the shallower hypoxia observed there (70 m depth) could be explained by the occurrence of local upwelling produced when westerly winds blow across the mountains, as was reported for the Comau Fjord by Crosswell et al., (2022) and Díaz et al., (2023). A shallower hypoxia (10−20 m) was also registered in Clayoquot Sound, British Columbia, mainly attributed to the local processes, such as, DO consumption during the organic matter degradation, enhanced by upwelling conditions (Rosen et al., 2022).

DO produced by photosynthesis is primarily depleted through respiration; therefore, in many oceanic environments, the GPP and CR rates tend to be tightly coupled. However, in coastal waters or fjord ecosystems,

owing to the high amount of organic matter (autochthonous and allochthonous) available in the water column, a greater amount of oxygen is consumed (Jackson et al., 2022) during microbial respiration (Robinson, 2019), thus dominating heterotrophic processes over autotrophic processes. This study recorded high CR rates (greater than GPP), indicating heterotrophic metabolism, where more oxygen was consumed than produced (Figure 8). Nevertheless, Puyuhuapi Fjord, Reloncaví Fjord, and Compu estuary showed high oxygen production values (GPP >20 mmol $O_2$ $m^{-3}$ $d^{-1}$) within the 2 to 4 m layer at least in one of the two sampling (Figure 8), suggesting a greater uptake than release of $CO_2$ due to the high surface primary production rates during the spring-summer period at surface waters of some Patagonian fjords (Torres et al., 2011). A recent global assessment of $CO_2$ uptake from estuaries, including tidal systems and deltas, lagoons, and fjord reported that fjords ecosystems can reduce 37% of $CO_2$ emissions (Rosentreter et al., 2023). DO and organic matter are produced only in the well-lit upper ocean but can be consumed throughout the water column. Therefore, in areas of high surface production and slow resupply of oxygen at depth (compared to the rate at which oxygen is respired), DO concentrations may be reduced to hypoxic levels at depth. Nevertheless, deepwater renewal was observed in Puyuhuapi Fjord owing to the inflow of salty water above the sill favoring the ventilation of the intermediate layer and inhibiting the occurrence of deep anoxia conditions (Pinilla et al., 2020), as was also reported by Jackson et al., (2022) in Rivers Inlet, British Columbia.

However, the first map of residence time in the northern Patagonian fjord showed a coincidence of the area where a high residence time was reported (100−250 days, Figure 13) with the depleted oxygen region, for example, hypoxic and LDOW areas (Figure 2). The hydrodynamic model and derived calculations are helpful tools for explaining the oxygen distribution and patterns in the northern Patagonian fjord system.

## 5. Conclusions

Since the beginning of oceanographic research in the Patagonian fjords (1970), hypoxia and LDOW have been reported along the water column. A dataset of hydrographic-biogeochemical parameters, *in-situ* experiments of primary production/community respiration, satellite-suspended sediments, and the output of a hydrodynamic model allowed for the first time the scrutinization of the oceanographic and biogeochemical processes contributing to hypoxia and LDOW in Patagonian subsurface waters. The influence of the ESSW was recognized as one of the first drivers of this phenomenon. This water mass transported poleward poorly oxygenated waters that entered the northern Patagonian fjords through the Guafo mouth and moved southward through the Moraleda and Jacaf channels and the Puyuhuapi Fjord. The distribution of the ESSW inside the northern Patagonian fjords coincides with the most extensive zone of the LDOW records. However, the hypoxic conditions registered in this zone were also attributed to the high rate of DO consumption during community respiration processes, owing to the entrance of allochthonous organic matter (natural and/or anthropogenic) to the fjords. Additionally, the results from marine currents derived from ADCP measurements and the hydrodynamic model demonstrated a weak water circulation and a long residence time contributing to the reduction of deep-water ventilation, resulting in a drop in DO level, and finally, the occurrence of hypoxic conditions. The Quitralco Fjord also reported hypoxic conditions showing the absolute minimum of DO with 9.36 µmol $L^{-1}$ and 1.6% oxygen saturation. ESSW was not observed in this area. Therefore, the presence of hypoxic water in the deep layer of Quitralco was attributed to weak deep-water

circulation and a long residence time, together with a high rate of DO consumption by community respiration and a

higher supply of allochthonous organic matter due to river discharge. Similar processes favoring LDOW were registered at the Aysén, Comau, and Reloncaví fjords, but in the Reloncaví systems, marine currents and residence time were significantly higher than in other fjords. The presence of LDOW in the Reloncaví systems responded mainly to biological processes, e.g., DO consumption during community respiration by phytoplankton and remineralization by bacteria. The analyses of the processes contributing to hypoxic conditions and LDOW in the

water column of northern Patagonian fjords allowed for the identification of the main causes favoring oxygen loss in each fjord investigated. The results provide the environmental information needed to contribute to the sustainable management of the northern Patagonian fjord under anthropogenic activities and climate change scenarios. In addition, it provides significant information for understanding the life cycle of marine species that inhabit Patagonian fjords. Finally, our study provides valuable insights into the relationship between hypoxia, low oxygen

processes, the origin of organic carbon, isotopic signals of carbon and nitrogen, the C:N ratio, and the presence of CDOM in the Patagonian fjords. These findings highlight the complex interactions between terrestrial and marine processes and emphasize the role of fjords as potential sites for hypoxia and the accumulation of organic matter. Understanding these relationships is crucial for accurately assessing the impact of low oxygen processes and carbon cycle dynamics in fjord ecosystems and their contribution to the global carbon cycle. Further research is needed to

elucidate the mechanisms driving these patterns and their implications for the functioning and resilience of fjord ecosystems against continuous environmental changes.

*Data availability*. All data sets used in this manuscript can be requested from the corresponding author.

*Author contributions*. PL: study design, data analysis, and manuscript leader. IPS: study design, collection, and

analysis of physical oceanographic data, and manuscript leader. PM: study design, collection, and analysis of primary production data. PD: study design, collection, and analysis of physical-biological oceanographic data. CA: collection, and analysis of primary production data. EP: developed of oceanographic model and analysis of physical oceanographic data. FB: collection, and analysis of biogeochemical data. MC: analysis of marine current data. AA: study design, collection, and analysis of satellite data. MA: collection and analysis of marine current data. GS: study

design, collection, and analysis of physical oceanographic data. CP: edition and analysis of satellite data. CS: collection, and analysis of physical oceanographic data. SA: collection, and analysis of biogeochemical data. PN: collection, and analysis of biogeochemical data. GM: collection of physical, biological, and biogeochemical data. RA: collection of physical, biological, and biogeochemical data. JSM: processing and analysis of model data for resident time. CSR: validation hydrodynamics model. All authors contributed to the writing of the manuscript.


*Competing interests*. The authors declare that they have no conflict of interest.

*Acknowledgments*.

This research was funded by COPAS Sur-Austral ANID AFB170006, the Office of Naval Research grant N00014-

17-1-2606, COPAS COASTAL FB210021, and CIEP R20F002. Iván Pérez-Santos was also funded by

FONDECYT 1211037. Facundo Barrera was founded by FONDECYT 31308. Cécile Pujol was financially supported by the F.R.S-FNRS (Fonds de la Recherche Scientifique de Belgique, Communauté Française de Belgique) through a FRIA grant. Patricio A. Díaz was funded by Centre for Biotechnology and Bioengineering (CeBiB) (PIA project FB0001, ANID, Chile). M.I. Castillo was funded by CS2018-7929 and ATE220033. We also
thank the Chonos initiative of the Instituto de Fomento Pesquero (IFOP) for facilitating the use of its computer systems to access the MIKE 3, WRF, and FLOW models and the Ministry of Economy, Development and Tourism of the Chilean Government for funding through the Undersecretary of Fisheries and Aquaculture. Elias Pinilla would like to acknowledge funding from the National Science Foundation (NSF Grant Number 2045866).

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
