# Peer review of "Oceanographic Processes Driving Low Oxygen Conditions Inside Patagonian Fjords"

_EGUsphere, 2023_

## Author Response (AR1)

**Review #1**

This manuscript has a wealth of information relative to fjords in the northern Patagonia region. It explores, using many diverse datasets, the physical and biogeochemical properties of these fjords and evaluates the drivers of low oxygen/hypoxia. However, I think that some of the data were not described appropriately or used to their full extent. In other words, even if the manuscript shows a lot of data/plots/numbers, some of the main conclusions remain qualitative (or at least, it feels that way). Furthermore, while the text is easy enough to follow for the most part, many parts are unclear. I do believe that these issues can be fixed by a careful review/improvement of the writing (and maybe in the structure of the text). Below, I share the rest of my concerns/comments, divided by major, secondary, and minor. I believe that this manuscript could be accepted for final publication after these major revisions are undertaken.

**Major comments and concerns:**

* The use of the term "deoxygenation" is misleading/confusing in this manuscript (even in the title). In my mind, and I assume that in many others working in the field of ocean deoxygenation, this word implies a trend; i.e. the loss of oxygen with time in the ocean. The first few sentences of the abstract do use the term in this sense. However, in most of the manuscript "deoxygenation" is used to refer to "loss of oxygen leading to hypoxia/LDOW". I strongly suggest to revise the use of this term. For instance, I believe a more accurate title would be "Oceanographic processes driving low oxygen conditions inside Patagonian fjords" or something along those lines.

We agree with this comment and change the title of the manuscript "Oceanographic Processes Favoring Deoxygenation Inside Patagonian fjords" by "*Oceanographic Processes Driving Low Oxygen Conditions Inside Patagonian Fjords*". Additionally, we eliminated from the text the reference term of Deoxygenation and change to loss of oxygen, etc.

* I commend the authors for adding models to their toolkit. However, the model evaluation is lacking. Only salinity is evaluated by qualitatively comparing model vs observed transect plots – and these plots do not even show the key isohaline (33 g/kg) mentioned in the text! I understand that this is not a modeling paper, but the model does add an important component to the analysis (it is key for the discussion related residence times). Therefore, more effort is needed to show that the models represent the circulation well. The evaluation of other variables must be added (temperature as a bare minimum, but also sea surface height, and ideally, currents). If possible, include quantitative metrics (in addition to qualitative plots).

Thank you for your constructive feedback. We agree with your suggestions regarding the model evaluation.

We'll include temperature, sea surface height, and currents in our revised model evaluation, which will certainly strengthen its representation of the circulation. Additionally, we'll

incorporate quantitative metrics along with qualitative plots to offer a more robust model assessment.

Thank you for your keen observation regarding the key isohaline (33 g/kg). While it's true that this isohaline doesn't visually appear within the model domains, we believe the model is still successful in reproducing the spatial structure, indicating that the processes controlling the salinity transport within the fjords are incorporated.

While the model may not replicate these processes with absolute accuracy, it enables us to understand the fundamental physical transport mechanisms at work. We will make sure to provide a clearer explanation of this in our revised manuscript.

* The analyses and discussion many times jump from describing long-term seasonal means to snapshots (ie, data from specific expeditions). These changes are not done/explained with care in the text, which is misleading or confusing at times. I think the flow of the text needs improvement to make sure it is clear when a conclusion is taken from a long-term mean or a snapshot. For example, in the Results section, Figure 4 (snapshot) and its description are found between a larger discussion of Figs 2, 3 and 5 (long term means); I feel that if Fig 4 and associated description were removed (which do not add a lot anyways), the flow of the text would be much improved. Other improvements are likely needed in the Results and Discussion sections.

We organize the Results section to present first the descriptions of the long-term and next the snapshot. We believe that the examples of areas where extreme low DO waters were detected (Fig. 4 and new Fig 6) is important to the context of the manuscript.

 * The authors calculate the differences in T, S, DO, NO3 and PO4 between a cruise in March 1970 and another in February 2021. They calculate trends with just these two time points and discuss deoxygenation (in the sense of long term decline of oxygen concentrations). Calculating trends and discussing long term changes with only two data points is a stretch. There is no way to prove that these differences are indeed trends and not due to interannual variability. While I do agree that it would be incredibly valuable to show trends and long-term DO decline, it has to be done properly by using a longer time series. I truly hope the authors can gather enough data from their dataset to calculate trends with more data points. But if for any reason they cannot, I strongly recommend to remove this trend analysis. If the authors want to keep Fig 6 in the manuscript, then the text should clarify that changes shown are only informative and that trends cannot be computed with only 2 points in time should be made explicit.

We eliminated the Figure and the description from the text.

* The figures using interpolation/extrapolation need careful revision. In particular, Figure 2 shows ESSW waters (as defined in the salinity panel) with really cold temperatures in spring, which I believe is just a result of incorrect extrapolation (no observations on those values under 8 degreesC).

We better the interpolation of Figure 2.

* The POC partition into autochthonous and allochthonous is not clearly explained. There are typos and different subscripts in the text and in the equation (lines 207-212), making the explanation hard to follow. Furthermore, in the equation the term chosen is POCterr, while later the text refers to POMalloc and the figure 10c shows POCorg (at least that's what I can read in panel 10c with 300% zoom – the caption reads POCallo). Furthermore, the use of carbon and nitrogen isotopes has to be properly described in the methods section (beyond the description of how they were measured). This change, along with improvements in the descriptions and discussions related to the isotopic distributions, will make it easier to understand the conclusions drawn in the Discussion section.

We edited the methodology section and improved the Results and Discussion sections of the Biogeochemical variables and analysis.

* The fontsize in most figures is too small. I needed 200% zoom to read the tiny text in some figures. For instance, Figure 1 might need to be reorganized to 2 columns/3 rows, such that it can take more of the page and doesn't need to be shrunk to fit the width of the page.

We edited the fontsize of many figures and changed the Figure 1 by a new figure fallowing the recommendations.

* I found several places where the > and < symbols were incorrectly used (e.g., > used for "less than").

We corrected the symbols in the manuscript.

**Secondary comments:**

* The last sentence of the abstract is very vague. I suggest to replace it by a more meaningful and concise conclusion.

We deleted the last sentence of the abstract and added a new sentence.

* The word "parameter" is used in this manuscript as "variable", as in temperature being a parameter. While it is not an uncommon use, I would suggest to favor the use of "variable" instead. In ocean modelling and more generally in math, a parameter is a component of the equation that one uses to try to represent the variable.

We changed the word "parameter" to "variable" across the manuscript.

* L100: "Finally, most published manuscripts hypothesize and discuss the processes favoring deoxygenation inside Patagonian fjords but never show any quantification": Besides the already mentioned issue with the term "deoxygenation" (which here could be replaced by "leading to low oxygen conditions"), the text "but never show any quantification" sounds unnecessarily harsh and dismissive of past publications. I'm sure this was not the intention, but I do suggest to find other wording. Furthermore, in the Discussion Section the reader finds a long list of references given regarding the study of "processes impacting this fjord

system, such as the hypoxic conditions due to the influence of the ESSW". None of these studies ever showed any quantification?

We agree with you and change the idea of quantification by words as: evaluate, estimates, analyses, scrutinizes, etc.

* I would appreciate some clarity on how the transects through all fjords were plotted as a continuous section (eg, Fig 2, 3, 5). It seems to me there should be discontinuities, unless data are repeated (e.g., from mouth to head and then back) or arbitrarily interpolated. For instance, starting in San Rafael Lagoon we can plot a continuous transect to the head of Cupquelan fjord; however, to go then we need to go back to the mouth of Cupquelan to start plotting the way to Quitralco fjord. My interpretation is that the authors lined up the stations of each fjord and then interpolated the whole dataset. But, if that's the case, I question whether the interpolation between the head of a fjord to the main channel (or mouth of the next fjord) is meaningful. Particularly, this interpolation may confound the interpretation of horizontal gradients seen in these transect plots. I'd suggest to blank out the locations where the transect is not continuous.

The data presented in the new Figures 2, 4, and 5 is no continue and included the passed over land market in white rectangles.

* I suggest to move section 2.3 "Satellite images" after section 2.4, so the biogeochemistry section comes right after the biology one.

We moved the section 2.3 to 2.4.

* Section 2.6: A brief description of the model would be beneficial for those not familiar with MIKE 3 FM. Is it hydrostatic? Is it finite elements or volume? What vertical coordinate system does it use? Also, please note that the reference list is missing "DHI (2019)", which is the only citation given for the reader to find out more about the model. Lastly, it would be interesting to know why you have two domains (that seem to overlap at their N and S open boundaries), instead of just one (note: by no means I imply that you should have one big domain; but a modeler wonders if the reason is computing power/efficiency, legacy, etc).

Thank you for your suggestions. We will provide a brief description of MIKE 3 FM, including its hydrostatic nature, use of finite volume elements, and the vertical coordinate system it employs.

We apologize for the missing "DHI (2019)" reference and will ensure to include it in our revised manuscript. The use of two overlapping domains is due to budget constraints and computational capacity that led to staggered development over time. This detail will be included in our revised manuscript.

* Water mass analysis in section 3: 1) The description (L276-280) gives the impression that the Estuarine Water (EW) is a water type associated with the freshwater from the San Rafael Lagoon, which then moves north. However, I think that the surface EW identified at the different fjords is likely originated from the freshwater inputs in each fjord. 2) Fixing the

previous point will also improve the description of MSAAW. The text describes MSAAW as a mixture of EW and SAAW, and figure 2c-d shows it as one water mass; however, the northern and southern 'branches' of MSAAW show quite the different temperatures. The latter might be explained by EW in the north being associated to the freshwater in the northern fjords (rather than the ice melt from the Lagoon).

*We edited the results presented in this section and added: "Moreover, the EW was also observed in the northern domain of the study area, especially at the surface layer in the Reloncaví system and the Comau and Reñihue fjords, contributing to the formation of the MSAAW. In this region, EW's origin was mainly due to the freshwater supply by river discharge. We identified two different sources favoring the EW's origin and the MSAAW's formation. 1) The combination of ice melting from the San Rafael with river discharge in the southern region and 2) the freshwater supply by river discharge in the northern region, both sources contributed to the difference in conservative temperature shown by the MSAAW".*

\* Further on water mass analysis: The authors do not discuss temperature when defining water masses (which is OK for the most part). However, I was surprised by seeing the ESSW colder than SAAW in Figure 2a,b. I think it would be valuable to add a brief description of the temperature ranges for these water masses and/or why an equatorial water mass is colder than a subantarctic one in this region. Note: not being an expert in this region, I did a quick search and found ESSW ranges of 13-14C (Silva et al 2009) and SAAW between 7-9C (Palma and Silva 2004), so now I'm even more curious about the temperatures in your figures.

*We added a new table and figure (Table 4 and Figure 3) to present and quantify the water masses in the study area. The results showed that ESSW was in general colder than SAAW as was obtained by Linford et al., (2023, Table 3). The difference between our results with Silva et al., (2009) and Palma and Silva (2004) could be attributed to the depth range used in the calculation of water masses.*

\* GPP:CR analysis: I am wondering if there is any chance that the spring bloom is in September/October and that your sampling missed it. Could you comment on this?

*Yes, the spring bloom could have occurred in September/October or even earlier... during August (winter late bloom, Montero et al. 2017) and we might not have recorded it. However, sampling campaigns were carried out within the productive season of the annual productivity cycle, and GPP and CR rates showed values that agreed well with measurements reported in previous studies for the Patagonian fjords during the productive period.*

*We have now added a short paragraph where this situation is indicated.*

\* CDOM: There is a clear (and striking) pattern of high-low-high CDOM in the northern side of the region of study (>~800km in Fig 10g). However, the authors only mention the high CDOM at the very north (Reloncavi) and then the low CDOM south of Jacaf Channel (L415-416). This should be fixed. That said, the authors do not discuss CDOM beyond the

description of the fig 10g. I believe that the authors should either add a discussion on what we learn through these CDOM data or remove the panel and associated description.

We edited the Results and Discussion sections in terms of CDOM descriptions.

**Minor comments/edits:**

L36: mechanismS

We change "mechanism" by "*mechanisms*".

L50: While COASTAL hypoxia

In the manuscript of Breitburg et al. (2018) a global map of hypoxia that occurred in the Oceans and coastal waters was presented. We edited the sentence *"Hypoxic conditions and deoxygenation have expanded globally over the last decade along coastal waters and oceans".*

Breitburg, D., Levin, L. A., Oschlies, A., Grégoire, M., Chavez, F. P., Conley, D. J., ... and Zhang, J.: Declining oxygen in the global ocean and coastal waters, Science, 359, 6371, doi:10.1126/science.aam7240, 2018.

L57: replace "y" by "and"

We replaced "y" by "and", e.g., *Díaz and Rosenberg, 1995.*

L65: this sentence is confusing as is. I believe you meant "In a Patagonian fjord used in recreational fishing for rainbow trout (Salmo gairdneri),"

We edited the sentence.

L80: this sentence is unclear. "the Eastern South Pacific (ESP) OMZ extends poleward, diminishing its influence on the adjacent coast of the Patagonian fjord system". Do you mean something like ", decreasing in strength and size to the south near the Patagonian fjord system"? No need to use my words, but please describe better what you mean by "diminishing its influence".

We edited the sentence.

L81 and other places in the manuscript: Linford et al (2023) is under review. Might need to change this reference if the paper is not accepted by the time this one is.

The manuscript of Linford et al. (2023) was recently published. We edited the references in the reference section.

*Linford, P., Pérez-Santos, I., Montes, I., Dewitte, B., Buchan, S., Narváez, D., et al.: Recent deoxygenation of Patagonian fjord subsurface waters connected to the Peru–Chile*

*undercurrent and equatorial subsurface water variability. Global Biogeochemical Cycles, 37, e2022GB007688. https://doi. org/10.1029/2022GB007688, 2023.*

L87: Guafo mouth (FIG. 1A),

We edited the text.

L89-90: "far from the influence of ESSW": "far" is misleading here, since many of these fjords do not seem far from each other. According to Silva & Vargas (2014), they are behind sills that prevent the ESSW flow into the deep waters of the fjords.

We change the sentence "Nevertheless, these conditions are also found in other areas far from the influence of ESSW" to "*Nevertheless, these conditions are also found in other areas where sills block the pass of the ESSW…*"

L102: "with salmon aquaculture occupying the first position". I understand what you mean, but this doesn't sound the best way to put it (in English).

We edited the sentence "with salmon aquaculture occupying the first position, with a national production of ~1,000,000 tons of salmon in 2019" to "*The Patagonian fjord ecosystem is under substantial continued economic pressure due to the salmon aquaculture and other economic activities (Billi et al., 2022)".*

L103: "The northern Patagonian fjord (a region similar to the study area; Figure 1b, 1c)": the paper refers to the area of analysis as northern Patagonian fjords, so I'm confused about "a region similar to the study area". Are the authors referring to the study area or somewhere else?

We edited the sentence as "*The northern Patagonian fjords (Figure 1b, 1c)….*"

L146: Table 1: what does the * represent next to (Temperature, Salinity) in the HUDSON expedition?

We added the following sentence *"*Temperature and salinity were measured with reversing thermometer and inductive salinometer respectively".*

L242: how do you provide current and sea level boundary conditions based on CTD stations?! Please explain better your velocity and sea level open boundaries.

We appreciate your insightful feedback. To elaborate further:

Our methodology employs CTD data solely for setting temperature and salinity conditions at the boundaries. For determining the water levels at the open boundary, we implemented a harmonic analysis (as advocated by Pawlowicz et al., 2002), utilizing data from a regional barotropic model (citing Pinilla et al., 2012).

Importantly, the flow data at the boundaries in our hydrodynamic model is prescribed as zero. This results in the current dynamics within the model domain being balanced internally across space and time. We intend to provide additional clarity on this aspect in our revised manuscript.

To enhance the robustness of our model, we are planning to incorporate validation using water level and current series at various points within the domains. This will serve to verify the model's performance and ensure its accuracy across diverse locations.

We hope this comprehensive response addresses your queries and provides you with a clearer understanding of our methodology.

L250-252: this sentence is too long and grammar is a bit unclear towards the end.

We deleted this sentence and presented a new description.

L254: please add a few words on whether the performance of FLOW-IFOP is good/acceptable (rather than just pointing to the evaluation).

We appreciate your input. In the upcoming revision, we will explicitly address the reliability and efficiency of FLOW-IFOP in our study. Additionally, in response to your suggestions, we will incorporate specific metrics to assess the model's effectiveness in Northern Patagonia's principal rivers.

Section 3: it would be good to reiterate here (with just a few words) that the long-term seasonal mean used all data shown in Table 1.

We added e new sentence: "*The long-term annual mean includes all data sets presented in Table 1. This section scrutinizes the behavior of the conservative temperature, absolute salinity, DO, and inorganic nutrients*".

L274: Desertores Pass in not shown in any of the maps, as far as I can tell. Also, add a call to Fig 2a,b at the end of the sentence.

We added the label Desertores Pass to Figure 2a and Figure 2b and edited the sentence.

L277: I think it should say Figure 2c,b

We edited the sentence.

L285-289: please revise the grammar and clarity of these two sentences.

We edited the sentence as follow: "*In the area contained by the ESSW, low DO (LDOW) and hypoxic waters were observed, but LDOW was registered at Reloncaví Fjord, where ESSW was not observed. In general, the Chiloé Inner Sea (CIS) showed a homogenization of the water column, mainly during the fall-winter seasons, in which high DO values (267−312 µmol L$^{-1}$) and oversaturated waters (< 100% DO Saturation) were registered. Additionally,*

*more extensions of the hypoxic conditions and LDOW were registered during the spring-summer seasons (Figure 2e–f)*".

L290: Figure 2 caption: why not say explicitly temperature, salinity and oxygen?

We edited the Figure 2 caption, "*Figure 2. Long-term seasonal mean of (a-b) conservative temperature, (c-d) absolute salinity, and (e-f) dissolved oxygen collected along a vertical section in the northern Patagonian fjord during the fall-winter and spring-summer seasons*".

L308: Magdalena Sound: this location identifier is not clear, as it is not shown the x axis of fig 3. I suggest that you add within the parenthesis already showing lat/lon (which doesn't help much when looking at fig 3), that this sound is found between the Puyuhuapi Fjord and Jacaf Channel.

We edited the sentence and added the label "Magdalena Sound" to Figures 1b, 1c, and 1e: "*The analysis of hypoxia and LDOW conditions in the northern Patagonian fjord system highlighted the presence of two areas with water bodies with these characteristics, e.g., the Puyuhuapi-Jacaf and the Reloncaví regions. Moreover, Magdalena Sound (44.6° S / 72.9° W, located between the Puyuhuapi Fjord and the Jacaf Channel, Figure 1) showed shallower hypoxia over the entire Patagonian region*".

L308: also, in this line it says "over the entire Patagonia region". Do you mean over the entire region of analysis (ie, northern Patagonia fjords) or really over the entire Patagonia (for which you are not showing results, so you would have to add a citation)?

We edited the sentences: "*Moreover, Magdalena Sound (44.6° S / 72.9° W, located between the Puyuhuapi Fjord and the Jacaf Channel, Figure 1) showed shallower hypoxia over the entire northern Patagonia fjords*".

L315: caption missing C and F in the first two sets of parenthesis

We edited the sentences: "*Figure 4. Vertical section of dissolved oxygen carried out along (a-c) Magdalena Sound (-44.65° S / 72.87° W) and (d-f) Quitralco Fjord during November 2016 and November 2020, respectively*".

L319: BruNt

We changed "Brut" to "Brunt".

L320: I believe you mean <. Also note that the text refers to the mixing in the whole water column but the figure only shows the top 50m. I suggest to explicitly mention that the figure zooms into the top 50m.

We edited the text and changed the figures to show the water column.

L324: I believe you mean >

We edited the text.

L380: "the observed high concentration of sediments was due to the presence of a diatom bloom mainly formed by *Skeletonema costatum*": do you mean SPM instead of sediments? Is the high SPM outside of the fjord mouth in Fig 9a representing the bloom occurring at the time (then, SPM would not be necessarily sediments) or did the bloom occur earlier and the high SPM represents resuspension of the previously sedimented diatoms? A clearer explanation is needed

The diatom Skeletonema costatum has a silica shell which has spectral characteristics that are complex to determine from satellite measurements (which typically only measure a few spectral bands). Complex waters, with a mix of sediments, chlorophyll, and other particulates, are particularly challenging for remote sensing. In Figure 9a we can appreciate whitish watercolors getting out of the fjord, and, more to the south, greenish waters which correspond to the Skeletonema costatum bloom region. So these are rather two separate water masses, and it would be difficult to assess, based on a cloudy and 5-day time step satellite time series, which one would occur before.

(the text in the manuscript, starting in line 380, should be changed to the following, for more clarity: "In the southeast part of the Reloncaví Sound, the observed high concentration of sediments was due to the presence of adiatom bloom **(seen as greenish waters in the southeast part of the Reloncaví Sound),** mainly formed by Skeletonema costatum, which attained concentrations of more than 5 million cells L -1within the fjord (Figure not shown)"

L410: >

We edited the text.

L412: showed light what? Overall, I think the paper needs a clearer description (in the methods section perhaps?) of what the isotopic distributions mean/what information we extract from them.

We changed the sentence "At depth, the organic matter showed light relative to the surface (Figure 10e)", to "*At the deep layer, the organic matter showed relatively light values compared to the surface layer*".

L453: need to mention Fig 12 at the end of the first sentence. Then, you can remove the sentence in Line 454-455 ("This result…. current speed").

We edited the sentence.

L458: DEEP flow

We edited the sentence.

L486: "processes favoring hypoxia and the reduction of DO, such as primary production". Please replace "reduction" by "decrease" or "decline" (I do not think you mean the reduction of DO to water or hydrogen peroxide). Also, the second part of that line is shocking as written, because primary production does not consume DO. Maybe you mean eutrophication? If not, please improve the sentence and explain the connection between PP and hypoxia/DO decrease.

We changed the sentence for: "*Furthermore, other studies have contributed to the understanding of processes favoring hypoxia and the decrease of DO, such as organic matter degradation and community respiration, associated with the abundance of different phytoplankton species*".

L510: "production": mixing does not produce oxygen. Replace by "ventilation", "reaeration" or a similar concept.

We changed the sentence for: "*This explains the importance of this area for DO ventilation and redistribution in the northern Patagonian fjords*".

L513-514: "even though the biotic and abiotic processes that occur in every body of water respond to the biogeochemical cycles to some extent": I do not understand this phrase. BGC cycles ARE a combination of biotic and abiotic processes...

We deleted the sentence.

L526-527: "but there is no evidence of gradient of nitrogen isotopic signals": is this a conclusion from the data presented here? Looking at fig 10f, Chiloe Inner Sea seems to have higher values than regions to the south.

We edited the sentence: "*but there is no evidence of gradient of nitrogen isotopic signals before this study. Nevertheless, the pronounced phytoplankton bloom resulted in an enrichment of $\delta^{13}C$ in SPM (Figure 10), similar to observations made in other fjords around the world (Remeikaite-Nikiene et al., 2017). In the same way, the enrichment of $\delta^{15}N$ of SPM results in primary production evidence that, in at least some regions of the fjords (e.g., Chiloé Inner Sea), the isotopic signals it was of mainly autochthonous origin (Figure 10)*".

L529-535: please improve writing, since sentences are hard to follow. Also, I don't think that is was clearly explained how the results suggest that "the cause of the variability in C and N isotope fractionation was biological processes"

We clarified the sentence in the Discussion section.

L538: "maintaining"? I'd think it would be "adding", since "maintaining" implies that there is nutrient consumption at these depths, which is not likely.

We edited the sentence.

L540-542: "This process seems…": this sentence is confusing. "This process" refers to the accumulation of nutrients through remineralization, but then the sentence ends on how organic matter in the southern region was retained and sedimented, leading to LDOW and hypoxia. Also, while I agree with the overall suggestion of how the hypoxia was generated, I think the explanation is speculative, rather than well justified by the data (eg, figures 2 and 3, which are used as reference, have no information on the organic matter; there is no information on sediment oxygen consumption, beyond the fact that hypoxia was observed close to the seafloor in one profile).

We edited the sentence: "*This feature seems to be even more evident from the Jacaf Channel to the south (Figure 3), where it seems that probably organic matter was mostly retained and sedimented, favoring zones more prone to LDOW and hypoxia (Figure 2 and Figure 3)*".

 L543: fjord heads: note that Fig 9 shows that the rivers do not necessarily flow into the fjord heads.

We edited the sentence: "Inside  the fjords, where low salinity….."

L570: (greater THAN GPP)

We edited the sentence.

L572-573: "Nevertheless, Puyuhuapi, Reloncaví, and Compu fjords always showed high oxygen production values (GPP >20 mmol O2 m-3 d-1) at 2 m depth (Figure 8), confirming that Patagonian fjords act as a net sink of atmospheric CO2 due to the high surface primary production rates (Torres et al., 2011)": The first part of this sentence is not always true, since Fig 7 shows GPP <= CR at 2m for Compu (2020, 2022) and Puyuahuapi (2020). Furthermore, it is not appropriate to confirm a net sink of atmospheric CO2 only through GPP observations (eg, strong GPP could be fueled by upwelled nutrients; but those upwelled waters could also be high in inorganic carbon and lead to an overall source of CO2 for the atmosphere).

There was an error in graph 7 with the values of GPP for Puyuhuapi. We have now fixed it.

Also, we have now rewritten this paragraph, following the reviewer's suggestion about the atmospheric CO2 sink.

L595: I think this sentence needs to add BIOGEOCHEMICAL before 'processes' to be accurate, since Linford et al (under review) seems to have looked into the advection of LDOW from the ESSW.

We edited the sentence by *"….first time the quantification of the oceanographic and biogeochemical processes contributing to the hypoxia and deoxygenation of the Patagonian subsurface water.*

L601: ", owing to the entrance of allochthonous organic matter (natural and/or anthropogenic) to the fjords". I'm not convinced this was shown quantitatively. But I trust that after improving the POCalloc methods, results and discussion, this issue will be solved.

We clarified the information in the manuscript as was mentioned before.

L611: why are some residence times significantly higher in some fjords? Is it because of the presence of a sill? I think the authors should explain the "Why" of the higher residence times.

Thank you for your insightful question about the variance in residence times among fjords. You're correct to suggest that certain geographical features like sills may affect the residence times.

The dynamics of each fjord are complex and not completely understood yet. However, we've observed that fjords with less river discharge, in comparison to others, tend to be more dynamic with shorter residence times. Moreover, fjords with steeper or shallower sills dissipate more energy from tides and can be slower towards their interiors.

We acknowledge that this information would provide valuable context to our study and will ensure to include it in the revised version of the paper.

L613: "DO consumption during primary production by phytoplankton": This is wrong. Please fix.

We edited the sentence by "…*DO consumption during the community respiration by phytoplankton, and secondary production by bacteria*".

**Review #2**

**Review of** *'Oceanographic Processes Favoring Deoxygenation Inside Patagonian Fjords'* **by Linford et al**

The manuscript by Linford et al. uses physical and biogeochemical data from multiple sources to identify regions with high and low oxygen in the Patagonia coastal system. I was impressed by the scope of data used to address this issue. Deoxygenation in coastal waters is a very important issue and one that deserves attention from the scientific community. While the topic is important and the amount of data used was impressive, there were some gaps in this manuscript. I detail these below in the major and minor comments.

**Major comments**

My first major concern is the term **deoxygenation**. I think that the authors are basing this on Figure 6, which compares 2 physical and chemical profiles – one collected in March 1970 and one collected in February 2021. While it is likely that deoxygenation is happening in the Patagonia coastal system, using 2 profiles to quantify deoxygenation is insignificant. If quantifying deoxygenation is one of the central foci of the manuscript, I suggest the authors examine the entire time series after the seasonal cycle has been removed. See Aksnes et al., 2019 (https://doi.org/10.1016/j.ecss.2019.106392) and Jackson et al., 2021 (https://doi.org/10.1029/2020GL091094) for suggestions on how to best show deoxygenation in fjord systems.

We eliminated the focus on deoxygenation of this manuscript due to the lack of data to quantify this process. We concentrate the attention in the physical and biogeochemical processes contributing to hypoxia conditions and low dissolved oxygen (LDOW).

My second major concern is the figures. Most of them were illegible due to small font size. Many figures were of maps and were difficult to interpret for those not familiar with the region. I think most of the figures need to be overhauled to make them more legible.

We edited most of the manuscript figures to better understand the numbers and font sizes.

My third major concern was lack of quantification of the water masses. While the major water masses were named, how they were defined was not stated. Of particular concern is that the authors discuss mixing and modification of the water masses and how the modification could impact oxygen yet neither mixing nor modification was explicitly examined or quantified.

We added a new table (Table 4) and figure (Figure 3 new) to identify and quantify the water masses presented in the study area using a TS diagram with the addition of DO data. In section 2.1, we incorporated the methodology applied, and new results and discussion were also added to the text.

My fourth major concern is the lack of chlorophyll data presented in the manuscript. The authors report many times about phytoplankton production and respiration. While really useful, these data were only collected sporadically compared to the physical and chemical data. I think adding chlorophyll data to Figure 2 would help the reader see how oxygen concentrations compare to chlorophyll concentrations.

We added a new figure to show the chlorophyll-a patterns during two oceanographic campaigns.

**Minor Comments**

-Line 30 – should 'decade' be 'decades'?

We changed "decade" by "decades".

- Line 31 – I think you mean both surface and subsurface waters her?

We edited the sentence.

- Lines 37 to 39 – I found this confusing because Puyuhuapi Fjord was mentioned twice for low oxygen water. Please clarify

I agree with you, but in the case of Puyuhuapi Fjord, Hypoxia conditions (<30% oxygen saturation) and Low DO water (30%–60% oxygen saturation) was registered.

- Lines 50 to 52 – I find the sentence confusing and contradictory. Please clarify.

We edited the sentence.

- Line 74 – I don't think this is true – not all OMZs are related to upwelling.

We deleted this sentence, even though most of the OMZ zone coincided with upwelling systems (C. Aguirre et al., 2019), excluding the Indian Ocean OMZ.

- Figure 1 – Due to the small font size, it is very difficult to read where d) and e) are located in the map. Also, in a) the authors don't show where the study area is in relation to the South American continent.

We edited and presented a new Figure 1.

- Line 128 – 'Profilers' should be 'profiles'

We changed "Profilers" by "Profiles".

- Line 138 – How was the AML data processed? There are no standard CTD processing steps for AML instruments so showing the reader that these data were properly processed is important.

We added the following sentences, *"In the cases of SEB CTDs, data were processed according to the manufacturer's protocol and software (SBE Data Processing). To the AML CTD, the data raw passed a quality control, eliminating records out of range to finally average data every one meter."*

- Line 139 – What oxygen sensors were used? What is the error estimates for the oxygen measurements both when they were compared with Winkler titrations and when they weren't compared with Winkler?

We edited the sentence: "*Membrane, optical sensors (e.g., SBE 43 and Optode 4831) and the Winkler method have been used to obtain DO data (Strickland and Parson, 1968). Some experiments were conducted to validate CTD DO data with the Winkler method showing satisfactory results and high statistical correlation (e.g., $R^2$ range from 0.92-0.99)*",

- Table 1 – I suggest making this table into a figure that shows the density of data collected by year and by season. It is really difficult to visualize how much data were collected in a table.

The Figure 1b and Figure 1c showed a map of the space distribution of all the data collected and included in this manuscript. The dataset was divided into seasonal stations (e.g., fall-winter and spring-summer). On another side, the Table 1 presented the information divided by years.

- Line 156 – I think some justification is needed here if the incubation time was not consistent. In other words, how can the data be comparable if the incubation time was different for each sample?

The incubation period was the same for all samples, approximately 8 to 9 hours. Time-zero bottles should be immediately set to measure the amount of oxygen when incubation begins. Now we have added the incubation period in the text.

- Line 166 – It took me a few minutes to understand how these 5 BP experiments fit into the 9 in situ experiments mentioned on line 149 – I suggest referring the reader to Table 2.

We added the reference of Table 2 to the end of the sentence.

- Section 2.5 – Information is missing here – what is the time period of the moorings? What were the sensor depths? Were other sensors on the moorings in addition to the ADCPs? In addition, I don't see the mooring locations in Figure 9.

We added information to the section. *"The moorings covered the period of January to December 2016, and the ADCP sensors were moored between 40 m and 100 m"*.

- Line 237 – what is the resolution of the SHOA nautical data?

The resolution of the nautical data is 500 meters.

- Section 3.1 – I'm not sure how the mean was calculated. Specifically, were all stations consistently sampled? Were there consistent measurements in the two seasons? A figure, as mentioned above, that shows the number of profiles as a time series would be helpful.

We added a time series of all CTDO profiles used in Figure 2. As the time series shows, the spring-summer seasons presented a major number of CTDO profiles in comparison to the fall-winters seasons, but at least 10 cruises were used to obtain the long-term seasonal mean. The mean was calculated using the Data Interpolation Variational Analysis (DIVA) gridding software developed by the University of Liege (http://modb.oce.ulg.ac.be/mediawiki/index.php/DIVA). DIVA analyze and interpolate data set in an optimal way as the optimal interpolation method, taking into the account the coastline and bathymetry of the study area. The calculation is executed on a finite element mesh adapted to the gridding domains (Troupin et al., 2010).

*Troupin, C., F. Machín, M. Ouberdous, D. Sirjacobs, A. Barth, and J.-M. Beckers (2010), High-resolution climatology of the northeast Atlantic using Data-Interpolating Variational Analysis (Diva), J. Geophys. Res., 115, C08005, doi:10.1029/2009JC005512.*

We added this information to the section 3.1.

- Line 277 – What proof do the authors have that ice was melting in San Rafael Lagoon?

We edited the sentence: "*This location also had the lowest salinity (15–21 gkg$^{-1}$), indicating the presence of estuarine water (EW) owing to the contribution of the ice melting from the San Rafael Lagoon and rivers discharges (Figure 2e-f)*".

- Lines 277 to 279 – What proof do the authors have of mixing? Perhaps showing a TS diagram would help to show the authors that water mass modification is occurring.

We added a TS diagram to present and quantify the water masses identified in this study.

- Lines 281 to 282 – A reference is needed to support this statement

We added the reference in the Discussion section, *"This water mass enters Patagonia by the Guafo mouth and moves south throughout the Moraleda Channel to finish inside the Puyuhuapi Fjord and Jacaf Channel (Linford et al., 2023)".*

- Figure 2 – A note is needed as to why so much less oxygen data were available. Again, adding a time series figure with the number of oxygen profiles would help. Also, I don't know what direction the x-axes is facing (i.e. north to south or south to north). The illegible font size makes this especially challenging. Where are the station locations in relation to the maps in Figure 1? Also, it would be helpful if a hypoxia contour line was added to oxygen.

We edited the Figure 2 to better the understanding. A subplots showing time series of profiles was added to the new figure. In case of oxygen data, less records were analysis in comparison to the CTD, because most of the CIMAR cruise obtained the DO by Winkler method at standard horizontal levels (0, 10, 25, 50, 75, 100, 150, 200, 250, 300, 350, and 400 meters) and not with CTDO at 1m.

- Figure 3e – I was struck by the low silicate near the surface. Is silicate limiting for phytoplankton here? Again, it would be useful to compare this figure with phytoplankton concentrations.

The low silicate near the surface layer means the use of this inorganic nutrients by the phytoplankton community as we mentioned in the discussion section, "*In the case of northern Patagonian fjords, EW is rich in silicic acid because of the riverine supply (Silva and Vargas, 2014), with characteristic pulses at the surface layer that changes throughout the year due to the organisms' consumption (Montero et al., 2011), as observed in northern Patagonia, including the Chiloé Inner Sea (Figure 4)*".

The Chl-a data presented in the new Figure 7 shows high concentrations of Chl-a in the similar area where the silicate was lower.

- Line 308 – I don't think that Magdalena Sound is labeled on the map in Figure 1.

We added the position of Magdalena Sound to Figure 1.

- Figure 4 – The map insets are illegible.

We eliminated the map from the figures and added to the Figure 1.

- Figure 6 – The subsurface oxygen minimum shown in e) has been well described in the literature. Please see Jackson et al. (2021, https://doi.org/10.1007/s12237-021-00999-y) and Rosen et al. (2022, https://doi.org/10.3389/fmars.2022.1000041) for examples of this feature in Canadian waters.

- Figure 7 – I can't read this figure at all. So I am unable to assess it.

We divided the figure to better represent the results.

- Section 3.3 – I find this section very descriptive and qualitative. I understand that the satellite data were used to estimate where terrigenous material could be entering the ocean. But why weren't the SPM and POC data also used to prove that the areas identified as high SPM by satellite were verified with the SPM/POC data?

The images described in section 3.3 correspond to periods with cloud-free conditions, necessary to obtain the SPM estimates with satellite data. There is not always a match between in situ data and available Sentinel-2 data, which has a revisit time of ~5 days. The satellite data provide a different view of the situation, i.e., the spatial extent of these episodes.

- Lines 393 to 394 – I don't think that the direction of river outflow can be observed from Sentinel satellite data?

If the water rich in sediments coming from the Rio de los Huemules is seen to penetrate into the fjord, we can assume the current flow is directed towards the fjord, and viceversa. Sediments (and chlorophyll, temperature and other variables) can serve as tracers of water dynamics.

- Figure 9 – I have no idea where these regions are in relation to the Patagonia coastal system
We added a subplot to show inside the study area the region selected to present the SPM results.

- Figure 10 – This is confusing – why are SPM and POC inversely correlated? I would expect that regions of high SOM would also have high POC.

We edited the Results and Discussion sections to clarify the description of Biogeochemical variables.

- Figure 11 – In c), why were the meridional and not the zonal currents shown? Also, I'm struggling to understand how these data contribute to the overall storyline.

We added the marine currents data to demonstrate the decrease in the intensity of currents from the outside zone (Corcovado Gulf) to the inside area (Puyuhuapi Fjord). This regime contributes to the residence time and is one of the reasons favoring hypoxia conditions in the Puyuhuapi Fjord. The zonal and meridional currents presented in the figure were selected according to the current roses' behavior and were added to the new figure.

- Line 466 – In most coastal systems, the residence time of the water varies with depth (e.g. Pawlowicz et al., 2007, https://doi.org/10.3137/ao.450401). What depth is the residence time calculated for?

We appreciate your query about the depth at which the residence time is calculated. We apologize for the lack of information in the model methodology section. In this study, the residence time was calculated using the average from 50m to the bottom, typically the zone where the lowest oxygen values are observed.

Your feedback is valuable, and we will add a paragraph to better describe this aspect in our revised manuscript.

- Lines 490 to 491 – Based on my comments above, I don't think that the authors can prove that the high SPM in satellite data is allochtonous organic matter

We deleted the sentence.

- Line 505 – Mixing was never investigated in this manuscript. Rather, the authors calculated buoyancy frequency, which shows the absence of stratification.

We edited the sentence: *"An absence of stratification was observed throughout the year in the Chiloé Inner Sea (Figure 5)"*.

- Lines 522 to 523 – No chlorophyll data were shown in Figure 10..adicionar la referencia a Figura 2 nueva con Chla

We added the reference of the new Chla figure to this senstence.

- Lines 536 to 537 – What evidence to the authors have to support that ESSW and SAAW are strongly connected to the estuarine water masses?

We edited the sentence: *"The SAAW is strongly connected with the estuarine water masses (MSAAW and EW), and the ESSW interacts with the SAAW, as observed in this study (Figure 2 and Figure 3)"*.

- Line 573 – I don't think that high $O_2$ always means that a region is a sink of atmospheric $CO_2$. This relationship can be complicated in some coastal region (e.g. Johannessen et al., 2014 **https://doi.org/10.4319/lo.2014.59.1.0211**

We have now rewritten this paragraph, following the reviewer's suggestion about the atmospheric $CO_2$ sink. (Line 587)

---

## Referee Report (RR1)

**Second review of *Oceanographic Processes Driving Low Oxygen Conditions Inside Patagonian Fjords* by Linford et al.**

The authors have done a nice job of tightening up and clarifying the manuscript. It is an important manuscript that brings together many different types of data. I sometimes struggled with the storyline because it attempts to bring together many data types that have been collected in different space and time to tell the story of oxygen in the Patagonian fjord system. Because of this, the manuscript sometimes seems too speculative and I have documented my concerns below. I recommend this manuscript for publication after moderate revisions.

- Line 54 – Is this correct? There are many natural mechanisms that can lead to hypoxia (e.g. remineralization, weak circulation, age of water, etc.)
- Lines 82 to 83 – I find this sentence confusing; where exactly does the OMZ decrease in size and strength?
- Lines 85 to 87 – Here the ESSW and OMZ and discussed separately. But isn't ESSW one of the reasons that the OMZ is there?
- Lines 102 to 104 – I think there are some typos here.
- Figure 1 – Much better. Though I notice that the moorings aren't on this map.
- Line 131 – Is 2000 profiles correct? Line 37 says that 1507 stations were sampled. Does this mean that there were 1507 separate stations sampled and only some of these stations were sampled more than once?
- Line 158 – add 'salinity' after EW,
- Section 2.2 – I don't see a mention of the 2009 data in this section
- Section 2.4 – Figure 10 shows the concentration of suspended particulate matter from satellites. Yet the authors interpret these figures in section 3.3 as the colour of the water. I'm confused here – how can the authors tell the colour of the water from SPM data?
- Section 2.5 – I don't see the location of the moorings on any map. Also, there is no description of how the currents were rotated.
- Lines 367 to 379 – This is a repeated paragraph
- Line 387 – This is the first mention of ice melt. What kind of ice are the authors suggesting – sea ice or glacial melt? What proof do they have of the ice melt?
- Lines 392 to 395 – How was this quantified?
- Figures 2 – Please make the fonts on the place names bigger. Also, please remind the readers in the caption what positive and negative AOU values mean.
- Figure 3 – I can't see the symbols. Please make them bigger.
- Figure 4 – Please make the fonts of the place names bigger
- Figure 4 – Please make the fonts of the place names bigger. Also, I suggest adding isohalines on this figure so the authors can see whether the stratification is linked to salinity.
- Figure 6 – Why were these specific dates chosen to display?
- Figure 7 – I can't see the labels on the map. Also, why were these specific dates chosen to display?
- Lines 528 to 532 – Are there in situ data to back this up?
- Line 530 – I don't think that SPM will tell you whether a diatom bloom was present

- Line 538 – I don't think that SPM will tell you whether the sediments are carbonate-rich
- Line 539 – I don't think that SPM will tell you about the concentration of organic matter
- Lines 580 to 589 – How did CDOM and SPM relate to the satellite images?
- Section 3.5 – I struggled with this section because I wasn't sure what the key points are that the authors are trying to make. I was left wondering whether this section is necessary?
- Figure 12 – What do positive and negative current values mean? How were the currents rotated? Where are these moorings – there are no maps that show this
- Line 610 – I'm not sure that this current is strong?
- Lines 663 to 665 – I don't think that this study scrutinized all processes contributing to hypoxic water. For example, the anthropogenic impact wasn't discussed.
- Lines 729 to 731 – I don't see any proof to back this statement up.
- Lines 805 to 812 – these manuscripts might be of interest in this section:
  - Jackson et al., 2023
    https://agupubs.onlinelibrary.wiley.com/doi/10.1029/2023GL104549
  - Thomson et al., 2017
    https://agupubs.onlinelibrary.wiley.com/doi/full/10.1002/2016JC012512
  -

---

## Author Response (AR2)

**Suggestions for revision or reasons for rejection**

(visible to the public if the article is accepted and published)

Review of "Oceanographic Processes Favoring Deoxygenation Inside Patagonian Fjords" by Linford et al
We submitted a new title: "Oceanographic Processes Driving Low Oxygen Conditions Inside Patagonian Fjords"

I thank the authors for their efforts to address the comments by another reviewer and myself. While I find that the paper has really improved, I believe that it still needs some work to be publication ready. The manuscript presents an amazing amount of data, both historical and brand new; however, the connections and links among the different datasets are sometimes missing or superficial. Furthermore, the writing would benefit from a careful revision of the use of English; some sentences do not read properly and some statements are either unclear or incorrect. I describe my major comments below, followed by all my detailed comments. Overall, I recommend a moderate/major revision before publication in Biogeosciences. I think that this manuscript has the potential to be a wonderful contribution to coastal oceanography and to the understanding of hypoxia in nearshore environments – but some more effort is needed to take the manuscript to its full potential.

**General main comments:**

**G-1) The manuscript feels like a collection of data description.** There is so much in there that is understandably hard to put it all together. Some variables are presented/discussed very briefly, such that the connections with other variables (i.e., the "big picture") are lacking. Furthermore, the text sometimes is not easy to follow, especially for readers not used to the geography and names in this region. A suggestion to improve the readability of the paper would be to structure the Discussion Section per BGC-region (ie., gather all the fjords/channels/etc that respond to similar mechanisms under a subtitle and explain carefully those mechanisms). This is just a suggestion, I'm sure there are many ways to improve the discussion.
IP: revisar

**G-2)** Please note that in many places in the text, but particularly in the **discussion**, some sentences read as if the statement was based on some previous published work (rather than a product of this work).
Iván
We agree with you, and we edited the discussion section.

**G-3)** While I sincerely appreciate the author's efforts to extend the model evaluation (which was indeed needed), now the model section in Methods is too long and includes validation results, which do not belong in a Methods section. I recommend all of the validation is moved to the appendix (not just the figures, which are there already, but also the text related to model evaluation); only provide a brief summary in the main text (in Methods or maybe in Results), pointing to the appendix for more details.

We will relocate the detailed validation results from the Methods section to the supplementary material. This will ensure that the Methods section remains concise and focused solely on the methodology. A succinct summary of the model evaluation will be retained in the main text. As recommended, this summary will either be placed in the Methods.

**G-4)** Font size in some figures remains too small; geographical locations are missing in some plots.
We better the Font size of all figures.

**G-5)** In the context of SPM (suspended particulate matter), sometimes it is assumed that SPM=sediments and SPM=organic matter. Not all SPM is sediments (e.g., the authors identify a phytoplankton bloom with satellite SPM) and not all SPM is necessarily organic matter (could be inorganic too).
We changed SPM from satellite to SSPM to clarify the description of the SPM obtained from biogeochemical sampling.

**G-6)** The introduction should describe (albeit briefly) the hydrological cycle of the rivers in this region. For instance, do they have spring or summer freshets? Are they rain-fed or glacier/snow-fed? This information is relevant for some of the observations presented in the manuscript.

We added a new paragraph to the introduction section "The main contributor to the increase of allochthonous organic matter is the river supplies, mainly during late winter and early spring, owing to the dominance of the ice melting instead of the precipitation during the winter season. As one of the highest river discharges highlighted, the Puelo, Petrohue, and Cochamo Rivers in the Reloncaví Fjord, the Cisne River in the Puyuhuapi Fjord, and other freshwater contributions from small rivers in the northern Patagonian fjords (Castillo et al., 2016; Schneider et al., 2014)".

**G-7)** Abstract needs to be improved. I suggest avoiding oversimplifications and avoid focusing on specific examples unless they help explain the big picture.

We edited the abstract following the recommendation of the reviewer.

**G-8)** There is little in the results regarding the horizontal advection of DO, beyond the water mass analysis; however, the discussion mentioned how respiration overwhelmed oxygen supply from horizontal advection. This kind of unsupported statements hurt the manuscript (even if the concept might be true); there are several instances of unsupported statements in the text.

We deleted the sentence in the discussion section: "Regardless, the input of organic matter (autochthonous or allochthonous) to the deep layer increased DO consumption through respiration, overwhelming the oxygen supply from horizontal advection (Figures 8 and 9)".

**All detailed comments:**

**1) L32-33:** The first sentence of the abstract reads: "The dissolved oxygen (DO) levels of oceanic-coastal waters has decreased over the last decades owing to the increase in surface and subsurface water temperature caused by climate change." This statement opens the paper with an oversimplified explanation of ocean deoxygenation; it misses other physical issues like reduced ventilation due to increased stratification (which is in part due to warmer surface waters, but also due to ice-melt). It later mentions biological and human effects as contributors to oxygen loss – but in some coastal regions, eutrophication is the main culprit. I strongly recommend the authors to start the paper with a statement that is unrefutably true (a simple solution would be to add "in part due to the increase in surface…" to the sentence, although I'd recommend modifying the sentence to include stratification/ventilation concepts). As a secondary comment, I am not sure what is meant by "oceanic-coastal" waters (open ocean and coastal waters? Just coastal ocean?).

We edited the sentence as follows: "The dissolved oxygen (DO) levels of coastal ocean waters have decreased over the last decades in part due to the increase in surface and subsurface water temperature caused by climate change, the reduction of ocean ventilation, and the increase in stratification, and eutrophication."

**2) L40:** 'Simultaneously' reads as if the records of hypoxia occurred at the same time, which may not be the case. Suggest replacing by 'Furthermore'
We replaced "Simultaneously" by "Furthermore" from the text.

**3) L45-47:** This example is very specific (only one fjord and the values from just one of the two observations in that fjord) and as such, it does not do a good job at supporting the final sentence of the abstract (which is a general statement). Please replace with a sentence that provides a summary of the role of biology, so the last statement is fully supported.

We eliminated the specific example and added a new sentence: "The role of the biology activity in oxygen reduction was evidenced in the dominance of community respiration over gross primary production".

**4) L49:** 'applying of numerical model' does not read well (grammatical issues). Furthermore, I'd suggest

writing instead about the significance of combining observational and modelling work (rather than just mentioning the 'application of a numerical model').
We replaced "applying of numerical model" by "combining observational and modeling work".
We added the following sentence: "This approach underscores the importance of a holistic understanding of the subject, encompassing both real-world observations and the insights provided by modeling techniques" at the end of the paragraph to highlight the idea.

**5) L60:** 'y' --> 'and'
We replaced "y" by "and" from the text.

**6) L86:** please provide the DO range (2 – 3 mL/L) also in the units you use in the paper (uM)
We
We replaced "(2 – 3 mL/L)" by "( 89 - 134 uM)" using the conversion  1 ml/L=44.661  µM

**7) L95:** remove "those" and the last "the"
We removed "those" and "the" from the text.

**8) L102-104:** Firstly, this sentence feels disconnected from the rest of the paragraph, which goes on to talk about the pressures on the Patagonian fjords. I recommend moving it to the end of the previous paragraph (line 101). Secondly, but not less important, the sentence still reads too harsh. Given the authors' reply, I think that they misinterpreted my previous comment, focusing on the (previous) word 'quantification' instead of the 'but never show any' (quantification). The sentence now reads "Finally, most published manuscripts hypothesize and discuss the processes favoring hypoxia and LDOW inside Patagonian fjords without showing any evidence, particularly that based on in-situ data and fieldwork experiments". Just looking at the previous paragraph, they cite many references regarding DO-consuming processes in Patagonian fjords --> don't they show any evidence? I strongly suggest that the authors move and rephrase this sentence, highlighting their amazing dataset, instead of saying what previous work did wrong/couldn't do.

We prefer to delete the sentence.

**9) Fig 1:** much improved
We edited again Figure 1.

**10) L116:** 'carried out' --> 'used'
We changed "carried out" to "used".

**11) L128:** analysE
We modified "analys" by "analyze".

**12) L131:** ~2000 profiles --> the number of profiles changed from the last version and it didn't get updated here. It should read ~1500 here. That said, I am curious to know why so many data got removed (7 CIMAR cruises were removed from Table 1).
We detected a number of oceanographic cruises out of the study area, then we eliminated and therefore the new number of the files (1507 stations) is less than the previously reported (2017 stations).

**13) L132-133:** I think that the in situ dataset, the primary production experiments and the modelling results *combined together* help demonstrate the processes leading to low DO conditions in the northern Patagonian fjords. The profiles alone don't really prove the relationship between water masses and biogeochemical processes, and the PP experiments + model contribute more than just find "other processes" leading to low DO. I strongly recommend these sentences are modified to make emphasis on the uniqueness of this paper (ie, the combination of a wide range of datasets and tools).

We edited the sentence: "The main goal of this study was to analyze the processes contributing to hypoxia and LDOW in northern Patagonia, such as ESSW advection, DO consumption during the use of organic matter (community respiration), biogeochemical processes, deep-water ventilation, and residence times of

water inside fjords. A combination of *in-situ* dataset, the primary production experiments, and the modeling results were used to demonstrate the occurrence of processes promoting low oxygen conditions in the northern Patagonian fjords".

**14) L141:** SEB --> SBE
We modified "SEB" by "SBE".

**15) L143:** remove "a" before "quality control"
We removed "a".

**16) L157:** Could you add a few words to describe why only S is used to define water masses?

We added a new sentence to explain the salinity criteria: "). In Patagonian fjords, the density features of the water column are dominated by the salinity vertical-horizontal distribution instead of the water temperature, justifying the use of salinity in the identification of water masses (Aiken 2012; Pérez-Santos et al.,                                                                                                              2014)".

**17) L158-163:** The way these sentences are written, it feels as results. Here, you should describe the different types of water masses in a general way (salinity characteristics, origin, etc). Leave specific descriptions/wording as 'was detected', 'was found', 'was identified' for the Results section, when you describe your own plots.
We edited the sentence.

**18) L168:** "gridding domains" does not sound right. But I don't really understand what you are trying to say, so I cannot propose a solution.

We changed "gridding domains" to "study area domains".

**19) L178:** add "(DO)" or just use DO if already introduced
We added "(DO)" at the description.

**20) L191-200:** not my area of expertise, so I cannot assess the BP methods

The BP experiment was conducted using the standard method described in this section.

**21) L208:** Suggestion for the caption: Gross primary production (GPP), community respiration (CR), and bacterial secondary production (BSP) from in situ experiments in northern Patagonian fjords and channels.

We edited the Table 2 caption, "Table 2. In situ experiments were carried out in Patagonian fjords and channels. Gross primary production (GPP), community respiration (CR), and bacterial secondary production (BSP) from in situ experiments in northern Patagonian fjords and channels".

**22) L215:** remove 'variable' and change 'datas' into 'data'
We removed the word "variable" and modified "datas" by "data".

**23) L265:** The ADCP provides "flow" magnitude and direction [could also use "current"]
We edited de sentence: "The ADCP provides current magnitude and direction".

**24) L274:** some times the model is referred to as 'MIKE 3 FM' and some times just as 'MIKE 3'. What is the correct one?
The correct form is "MIKE 3 FM" refers to "Modelling for Integrated Environmental Management", the third version. The "FM" refers to "Flexible Mesh".
We have corrected "MIKE 3 FM" throughout the text.

**25) L276:** 1) add a comma before 'named'; 2) use the same format to describe the latitudinal extension of both domains (eg, use the same parenthesis form used for D1_Chiloe also for D2_Aysen).

We modified the text in section 2.6. This sentence is not present in the new text.

**26) L277:** I find this sentence confusing, particularly the 'maintaining the same configuration' part. Do you mean that the D1_Chiloe and D2_Aysen have been ran and validated in the four papers listed in lines 278-280? Or do you mean that the parameters/parameterizations used for the two applications of MIKE 3 (D1_Chiloe & D2_Aysen) are the same as those in the listed papers? Maybe it's a combination of the two? Please clarify (in particular, the reader needs to know if these two applications are brand new or if they have been developed for other purposes before).
We have taken steps to provide more clarity in the methodology section of our manuscript. To clarify:
The D1_Chiloe and D2_Aysen models are core components of the "CHONOS Initiative" accessible at chonos.ifop.cl. This open-access system provides comprehensive oceanographic data, designed primarily for environmental applications (Reche et al. 2020).

The framework was initiated by the Instituto de Fomento Pesquero (www.IFOP.cl) with support from the Undersecretary of Fishing and Aquaculture of the Government of Chile. This framework aids decision-making in salmon aquaculture in southern Chile. It has informed studies on algal blooms (Mardones et al. 2021, Diaz et al., 2021), larval fish dispersion (Landaeta, 2023), and marine circulation (Soto Riquelme et al. 2023, Perez Santos et al. 2019).

**27) L282:** suggestion: "…domains (D1_Chiloe and D2_Aysen) was …"
We modified the sentence as suggested "…domains (D1_Chiloe and D2_Aysen) was …". We moved the sentence to the supplementary material.

**28) L283:** suggestion: "The D1_Chiloe domain was developed first, followed by …"
We modified the sentence as suggested "The D1_Chiloe domain was developed first, followed by …". We moved the sentence to the appendix.

**29) L291:** What is SHOA? Spell out.
SHOA (an acronym for its Spanish name, Servicio Hidrográfico y Oceanográfico de la Armada de Chile).
We explained the acronym in the caption of Figure 1 (where it was first mentioned in the text).

**30) L294:** what is meant by 'element size'? Is it the (min/max/mean?) side of the triangle? Is the square root of the triangle area? Please clarify in the text

We edited the section on the circulation model.

**31) L298:** replace 'superior' by 'higher'
We modified the text in section 2.6. This sentence is not present in the new text.

**32) L307:** Using THESE data
We replaced "this" by "these".

**33) L310-314:** Switch order of sentences: "The correlations for the three main rivers in these regions – Puelo, Palena and Aysen (Table S1, Fig S1) were 0.88, 0.76, and 0.87 respectively (see Fig S2 in supplementary material). The full performance of the …"
We switched the orden of sentences in the text. We moved the sentence to the appendix.

**34) L316:** either remove 'data' or say 'on data from a regional barotropic model'
We modified the sentence: "..., based on data from a regional barotropic model".

**35) L316-320:** These sentences are results from the model evaluation and do not belong in the methods section. Please see my general comment regarding this topic.
We will move this section to the supplementary material.

**36) L323-329:** I'll comment on these sentences below, but my general comment still is valid (describe model setup first; then briefly describe the model evaluation and point to the appendix for details)

We restructured the section to first describe the model setup and subsequently offer a brief description of the model evaluation, directing readers to the supplementary material for a more comprehensive overview.

**37) L323:** "The performance of modelled currents compared against ADCP data from different moorings (Fig S1, table S1) is shown in the supplementary material (Fig S4 and S5)." The sentence that follows is repetitive and can be deleted.
We modified the text in section 2.6. This sentence is not present in the new text.

**38) L330-338:** This is results, does not belong to methods. Needs to go to the Appendix (see my general comment). As a more generic comment, be cautious with the use of the word "identical".
This information was relocated in the supplementary material, as you rightly pointed out that it presents results rather than methods.

**39) L341-355:** same as above (see my general comment).
Similar to above answer.

**40) L356:** suggest to make this paragraph a separate subsection called "2.7 Flushing time calculation", "Modelled flushing time" or something like that
We agree with your suggestion and we added a new subsection titled "2.7 Flushing time calculation.

**41) L367-378:** repeated text. Delete
We deleted repeated paragraphs.

**42) L381:** also refer to Figs 1c,d and Table 1?
We added "(location and detailed information are presented in Figures 1c,d and Table 1)" at the end of the sentence.

**43) L385:** 'but colder' --> 'and the coldest'
We replaced "but colder" by "and the coldest".

**44) L392:** Suggestion: "Therefore, we identified two different sources of EW that led to the formation of the MSAAW…"
We replaced the sentence "We identified two different sources favoring the presence of EW and formation of the MSAAW..." by the suggestion "Therefore, we identified two different sources of EW that led to the formation of the MSAAW..."

**45) L395:** Just a comment to ponder: MSAAW gets described as a unique water mass due to its salinity, with freshwater provided by different sources in the north and south of the study area; however, if temperature were considered as well, we would be talking about two distinct water masses. This is why I think a bit of justification for using S only would be useful (comment around line 157)

Similar answer to comment 16.
**46) L399:** add citation to Fig 2g,h at the end

We added the citation to the text.

**47) L400-401:** remove 'mainly' and fix '<' (it should be '>'). That said, this sentence feels disconnected. There is a sudden focus on Chiloe & high DO (when there are also other areas with high DO that go unmentioned). Then, the topic of high DO is dropped and the following sentence in the paragraph goes back to describing hypoxia and LDOW. Please improve the flow of the text.
IP: revisar
We removed the word "mainly" and changed "(DO Saturation > 100%)"
We added a new sentence to the text: "In addition, high DO records were measured in the San Rafael Lagoon, between the Aysén Fjord and Puyuhuapi Fjord, and in the Corcovado Gulf (Figure 2g–2h)".

**48) L402:** "more extensions" --> I don't understand.

We modified the sentence "Additionally, **more extensions** of the hypoxic conditions and LDOW were registered during the spring-summer seasons" by "Furthermore, a larger area with hypoxic conditions and LDOW was recorded during the spring-summer seasons (Figure 2g-2h) compared to the autumn-winter period" for your better understanding.

**49) L403-404:** This is the only mention to AOU in the whole paper. I recommend the AOU panels are removed from Fig 2. If the authors want to keep them, they need to properly define AOU in the methods section and make better use of the panels in the results and discussion.

We removed the AOU data from the text and Figure.

**50) L405:** suggestion: remove the whole line and start the paragraph with "Figure 3 ….". Even better, re-write the sentence so it doesn't start with "Figure 3…". For instance: The quantification of water masses through a TS diagram (Figure 3) highlights the dominance of…. You can cite Table 4 at the end of the sentence.

We modified the entire paragraph based on suggestions: "The quantification of water masses through a TS diagram (Figure 3) highlights the dominance of the MSAAW, with proportions of 60.96% and 54.67% during the spring-summer and fall-winter seasons, respectively. Following closely, the SAAW was the second dominant water mass, with proportions of 15.25% and 22.64%. The EW came next, and finally, the ESSW displayed the smallest proportion, with values of 10.77% and 11.15%. It's worth noting that ESSW was generally characterized by cold, salty, and poor dissolved oxygen (DO) content in comparison to the EW, MSAAW, and SAAW, as detailed in Table 4."

**51) Fig 2:** 1) the font sizes are still too small. Panels c to j need x and y labels with a similar font size as in panels a and b. Furthermore, the geographical legends on the top of panes c and d are barely legible, the require bigger font size; use acronyms if needed, and spell them out in the caption. 2) the colour scale of oxygen has two sets of units, which is confusing. DO in uM is quite different to DO saturation in percent. Maybe you want to have 2 colour bars, one for each unit? But 100 uM is definitely not 100%.

We edited Figure 2 according to all the comments proposed by the reviewer.

**52) L415:** Fig 2 caption: It needs to spell out the acronyms used in the figure (so far CIS and DP, but maybe new ones). Also: "(a,b) Time series of the number of CTD profiles used to compute seasonal averages."
We edited the caption of Figure 2.

**53) Table 4:** add units to CT columns
We added the units: CT (°C)

**54) L427:** 'As shown in Fig 2,'
We replaced "As previously shown" by "As shown in Fig. 2" from the text.

**55) L429:** I think you mean 'below depths of 50 m' instead of 'at a depth of 50 m' ?

We replaced "at a depth of 50 m" by "at depths greater than 50 m".

**56) Fig 4:** same font size issues as fig 2.
We edited the font size of Figure 4.

**57) L443:** 'On another side' --> 'In contrast'
We replaced "On another side" by "In contrast'" from the text.

**58) L443-445:** It would be useful to put these observations of stratification in light of some information on the seasonality of river discharge in these regions. For instance, higher BFV values seen in spring-summer are likely due to the peak discharge being in one of these seasons – do these rivers have spring or summer

freshets? Also, how is river flow in fall-winter? Given the still high stratification, the discharge is still considerably large; are they rain-fed? Is there a rainy season?

We added a new sentence to enhance the description of the stratification parameter.
The main drivers of the stratification in northern Patagonian fjords depended on the freshwater supply by rivers and precipitations. The highest river flow relies on the melting of ice that occurs during the spring, but during the fall and winter, rivers also increase the flow due to the precipitation regime. Additionally, stratification enhanced in summer owing to the increase in solar radiation, making the spring-summer season the period where the absolute maximum of stratification was registered (e.g., ~100 cycl h$^{-1}$) (Figure 5a).

**59) L450-451:** BruNt, fjordS, and remove comma after summer
We modified the caption of figure 5:
"The long-term seasonal mean of Bru**n**t-Väisälä frequency along a vertical section in the northern Patagonian fjord**s** during the (a) spring-summer and (b) fall-winter seasons".

**60) L453:** section 3.2 (not 3.3)
We modified 3.3 to 3.2

**61) Fig6:** the O2 color scale is confusing. What are the colors in the plot, % or uM?. X-label should read "section distance from the head of the fjord [km]" (or whatever is appropriate, if it's not from the head of the fjord)
We edited the Figure.

**62) L470:** "This observation is scrutinized.." Maybe remove this sentence or replace by " , which will be further discussed in Section 4"
We edited the sentence.

**63) L471:** deficiency --> lack?
We replaced "deficiency" by "lack" from the text.

**64) L472:** before --> above?
In the sentence "before" it refers to what was indicated in the previous sentences. We changed to "earlier".

**65) Fig 7:** 1) font size for the text in panels d and e is too small. 2) Caption: start with '(a)'; Remove 'obtained'; Make '(d) and (e)'; mention what is the red line in those panels

We edited the label of Figure 7.

**66) L481:** 3.3
We modified 3.2 to 3.3.

**67) L482:** remove 'campaigns' and the last 'the'
We removed the words "campaigns" and "the".

**68) L510:** "Gross", not "Global"
My editing error, of course it is "Gross"

**69) Fig 8:** It might be worth dividing this figure into two separate figures: panels a & b on one figure and c to j in another. Also, please note that Autotrophic and Heterotrophic labels are switched in Figure 8c-j. Lastly, I noticed that the manuscript only refers to figures 8c and 8j (except for a mention in the discussion to Fig 8 -assuming these panels- in the context of air-sea CO2 exchange). Maybe lines 488 & 490 should read '(Figure 8 c to j)' instead of '(Figure 8c, 8j)'?

We divided Figure 8 into two figures according to the comments of reviewers during the first round. We

edited the text (Figure 8c to 8j)

**70) Fig 9:** hard to read the text in italics. Remove italics and enlarge. Caption: 'Gross'
We changed the italic fonts to regular fonts and we edited the sentence.

**71) L526:** 'sediments': it should read 'particles' or 'matter'. It's not necessarily sediments, as discussed below (skeletonema bloom).
We replaced "sediments" by "particles" from the text.

**72) L529:** 'sediments' --> 'suspended particles'. Note that this is part of the same comment I made before, but I feel that the authors did not fully understand my previous comment (old L380 comment).
We replaced the word "sediments" by "suspended particles".

**73) L532:** Remove 'While'
We removed the word "While".

**74) L534-536:** Change 'sediments' for 'suspended particles' or 'suspended matter'. Furthermore, a suggestion: replace "Some examples of this phenomenon, such as the one on … " by "However, heterogeneous distribution can also be found (e.g. on April 6 2018 on Comau Fjord, figure 10d), suggesting the influence….".
We replaced the word "sediments" by "suspended particles" and we modify the sentence as suggested: "However, heterogeneous distribution can also be found (e.g. on April 6 2018 on Comau Fjord, figure 10d), suggesting the influence  of suspended particles within the fjord,"

**75) Fig 10:** This a small comment, but it'd be good to keep consistency and use log mg/L in this figure, instead of log g/m3. The two units are the same, but fig 11 uses mg/L, so it would be good to keep them consistent.
We edited the units of Figure 10.

**76) L555:** differential --> I think you mean different?
Yes, we modified the word by "different".

**77) L558-559:** here it says that high SPM values were high at 50-150m from the lagoon to Jacaf Channel except for Quitralco. However, I don't think this description is accurate or maybe it's just confusingFirst of all, it would be good to define first what is considered 'high'. Most of the plot shows >2 in the subsurface, even north of Jacaf and also in Quitralco. Really high values are only found in Cupquelan; smaller areas with subsurface values >3 mg/l are found in Quitralco and Aysen. Please clarify and improve the description; right now it's unclear why high values are highlighted everywhere in the southern region except in Quitralco (which seems higher in the cited depth range than say, in Puyuhuapi)
We edited the sentence: "We observed different signals of SPM (Figure 11b) along a latitudinal transect conducted from north to south in the northern Patagonia fjords (Figure 11a). Specifically, significant increases in SPM were registered in Cupquelán Fjord (~ 7 mg L$^{-1}$) and at the surface layer of the mouths of the Comau Fjord (~ 6 mg L$^{-1}$). Furthermore, smaller areas with values of SPM> 3 mg L$^{-1}$ were found at the subsurface layer of the Quitralco and Aysén fjords. In the northern region, within the Chiloé Inner Sea and the subsurface layer of the Comau and Reloncaví fjords, SPM values were minimal".

**78) L561:** Please show Chiloe Inner Sea (CIS) in Fig 11a,b,c, as done in previous transect plots. Just looking at where Chiloe Inner Sea is in Fig 1, makes me think that this region has both low and high (as in >3 mg/L). But hard to know because CIS is not shown in Fig 11b

We edited Figures.

**79) L561-563:** core? Characteristic cores? This description is unclear. Also, the paragraph first described conditions at the southern region, then went to the north, and then went south again, ending in north. The text is not flowing well and the geographical "jumps" gets confusing.

We edited the paragraph as was presented in the comment 77.

**80) L570-572:** Firstly, please note that "the C:N ratio decreased along the water column (from surface to the seabed)." is a description applicable for most of the transect, not just the Chiloe Inner Sea region. Secondly, Reloncavi and Chiloe show a similar patterns for C:N, but Reloncavi is not mentioned at all. Rather than a 'decrease with depth', the most noticeable characteristic of Chiloe (and Reloncavi) is that the vertical gradients of C:N are overall small, with ranges mostly within 6-9. Lastly, it's quite noticeable that while Chiloe and Reloncavi have similar low C:N, but they differ in their CDOM concentrations and POCallo & isotopic composition. These differences and similarities merit a mention and analysis.

We edited the sentence: "The C:N ratio exhibited a discernible reduction with increasing depth along the water column in the broader study area, encompassing the Chiloé Inner Sea region and extending to the Reloncaví Fjord. Although a similar C:N ratio trend was observed in both Chiloé and Reloncaví, it is noteworthy that the predominant feature is not a uniform 'depth-dependent decrease' but rather an overall modest vertical gradient of C:N, typically ranging within 6-9. Furthermore, it is evident that, despite their comparable low C:N ratios, Chiloé and Reloncaví diverge regarding CDOM concentrations and the isotopic composition and POC allocation. Specifically, the Chiloé Inner Sea and the Reloncaví Fjord exhibit analogous patterns of minimum C:N ratios (6-9). At the same time, the subsurface regions of the Puyuhuapi and Aysén fjords register absolute maximum values (C:N>14), as depicted in Figure 11d".

**81) L580:** "Notable observations" --> what does it mean? Is the high CDOM notable? Or are the observation notable because it's the first time CDOM is measured? Please improve wording.

We edited the sentence: "The first CDOM observations in the Patagonian fjords region denoted high concentrations throughout the water columns in the north area, spanning from the Chiloé Inner Sea to the Reloncaví system (Figure 11g)."

**82) L581:** Really low CDOM in Comau is not mentioned
We added the information of the absolute minimum detected in the Comau Fjord.

**83) Fig 11:** bigger font size still needed in x,y axes, colorbar and region indicators. 11c: Colobar should read POCallo. Please indicate Chiloe Inner sea both in 11a and 11b,c. Caption: suggestion: "(a) Study area showing transect (stations in blue; transect in red) carried out in November …."

We edited the caption.

**84) L597:** "inner sea of the region": do you mean Chiloe Inner Sea? This is the way you have been referring to the region
We replaced "inner sea of the region" by "Chiloe Inner Sea" from the text.

**85) L597-598:** showed a marked east-west "direction", rather than "gradient". Also, should the parethensis point to figure 12b, rather than just 12?
We edited the sentence.

**86) Fig 12 caption:** add at the end "Location of moorings shown in Fig S1"
We edited the sentence.

**87) L626:** Note that 'deep currents' are not by definition subtidal or residual. Suggested re-writting: Deep subtidal (or residual) currents (average of 50-300 m over the three years of simulation), ….

We edited the sentence.

**88) L627:** remove Consequently.
We removed the word "Consequently".

**89) L636:** replace 'north to south' by 'mouth to head', since it's easier for the reader unfamiliar with the region.

We replaced "north to south" by "mouth to head" from the text.

**90) L641:** What is meant by "Most studies"? Cite other studies and/or base this observation from your calculations

We deleted the sentence.

**91) L654:** replace 'some' by 'other'

We replaced "some" by "other".

**92) L663:** is this sentence missing an AND/OR between mixing and stratification? Later in the same line but next sentence: suggest to replace 'This' by 'Our', since otherwise it reads as if you were pointing to Ruiz et al.

We added "and" between the words and we replaced "This" by "Our" from the text.

**93) L667:** typo in "properties"

We corrected to "properties".

**94) L672:** suggest: "salty-hypoxic/LDOW"

We modified the sentence according to suggestion: "salty-hypoxic/LDOW".

**95) L673:** "This water mass enters Patagonia AT DEPTH by the Guafo mouth…"

We modified the sentence by "This water mass enters Patagonia within the subsurface layers via the Guafo Mouth..." for better compression.

**96) L675:** replace "On the other hand" by "In contrast". Also suggest: "the EW dominated in the UPPER LAYERS OF THE southern area, …"

We replaced "On the other hand" by "In contrast" from the text.

**97) L676:** due to its high solubility AND RECENT CONTACT WITH THE ATMOSPHERE

We added "...and recent contact with the atmosphere" to the text.

**98) L688:** cite Fig 4f after "silicic acid", and consider whether "silicate" is more appropriate than "silicic acid"

We modified the sentence: "...rich in silicate (Figure 4f) because..."

**99) L688:** remove S in changeS

We removed "S".

**100) L688-689:** "Pulses" indicate changes with time, while this figure shows a long-term mean. I think that with wording such as 'characteristic pulses" the authors are trying to refer to the high spatial variability in silicate. If so, please fix; otherwise, please improve description.

We edited the sentence.

**101) L690-691:** The first sentence of this paragraph is confusing. By definition, MSAAW is SAAW + EW. Does that make SAAW strongly connected with EW? And does that make MSAAW an estuarine water mass?

We deleted the sentence.

**102) L699:** Suggestion to improve the wording: Thus, organic matter degradation led to DO consumption

We replaced "Thus, DO consumption was favored owing to organic matter degradation..." by "Thus, organic matter degradation led to DO consumption...", as suggested.

**103) L707:** what are fjord outlets?

We changed "outlets" to "mouths".

**104) L708:** remove 'that'
We removed "that" from the text.

**105) L712:** why "subsurface"? the minimum values in those fjords seem to be at the surface in Fig 11c.
We edited the sentence.

**106) L719-721:** I do not follow/understand the last sentence of this paragraph.

We edited the sentence: "Given the notable seasonality in carbon fluxes and fjord system dynamics, as highlighted by Gonzalez et al. (2010), the potential implications extend to the variability of trophic webs and carbon export. Recognizing that these effects are more autonomous from local primary production than open ocean water systems is crucial. The seasonally driven variations in carbon fluxes can significantly influence the intricate interactions within trophic webs and contribute to fluctuations in carbon export processes within the fjord system".

**107) L721:** same issue regarding the word "subsurface" as in my L712 comment. Maybe replace by "upper layers"?
We edited the sentence as in comment 105.

**108) L727:** why is the C:N vertical gradient "more plausible" than the terrestrial vs marine source? The combination of high POCallo + high C:N makes the 'terrestrial source' a convincing explanation too. I would argue that it's likely a combination of the two explanations, rather than one being more plausible than the other
We agree with this comment, and re-edited the sentence:
We edited the sentence: "While the intuitively higher C:N ratio might be attributed to terrestrial sources with a higher carbon-to-nitrogen content, a more plausible explanation could be the increasing trend of the C:N ratio with depth and elevated particulate organic carbon (POCallo), or a combination of both. The observed C:N ratios suggest the influence of microbial remineralization processes and the preferential removal of nitrogen-rich organic matter (Taucher et al., 2020), potentially resulting in a lower POCallo signal".

**109) L735:** "but there is no evidence of gradient of nitrogen isotopic signals prior to this study": Do you mean that this study is the first one showing delta15N or that previous study with delta15N exist but showed no gradients? Also, it would be good to further explain the knowledge gained by this new information. Maybe this statement fits better in the following paragraph, which focuses on delta15N.

We agree with this comment, and added to the next paragraph: "The horizontal–vertical distribution of the isotopic signal of δ13C and δ15N showed substantial variability in the northern Patagonian fjords and, for the first time, reported for the d15N at least".

**110) L735-736:** remove "Nevertheless". Should likely cite Fig 7b after 'phytoplankton bloom' and add "in the upper 50 m" before citing Fig 11e (add "e")
We modified the sentence: "The pronounced phytoplankton bloom (Figure 7b) resulted in an enrichment of δ 13C in SPM in the upper 50 m (Figure 11e), "

**111) L737:** Suggestion: start a new paragraph with "The seasonal distribution of Chl-a…"
We modify the paragraph.

**112) L742:** Does "ice melting" start in early August in this region? The winter bloom happened sometime between 28-Jul and 10-Aug (dates taken from Table 1), which sounds more like the middle of the winter and too early for ice melting. Did 2021 had a particularly short or warm winter? Could the re-stratification that led to the phytoplankton bloom be due to some other driver, like precipitation or rain-fed river discharge? Unless better supported, ice melting does not seem a reasonable explanation for re-stratification in the middle of the winter.
We edited the sentence. Pérez-Santos et al. (2021) demonstrated that the primary driver process that

contributes to water column stratification is the early start of the ice melting during the middle (August) and late winter season (September). On the other hand, when the ice melting finishes, stratification of the water column depends on solar radiation during the summer seasons.

**113) L743-748:** from "The isotopic carbon..." onwards: these sentences go back to the topic of isotopes, which was discussed earlier in this paragraph. You had already moved to chlorophyll/blooms. Please improve the flow of this paragraph and avoid changing topics back and forth
We moved the paragraph.

**114) L763:** this sentence refers to the role of allochthonous particulate organic matter from local rivers and Figure 10 is referenced. Please make an explicit mention that Fig 10 shows suspended particulate matter and not necessarily POM (i.e., it could be a combination of organic and inorganic components). Otherwise, you need to prove that SPM is all or mostly POM (and just quickly looking at Fig 11, I don't see easily a correlation between high SPM and high POCallo)
We agree with the comment and consider deleting this paragraph to avoid speculation.

**115) L765:** should cite figure 11 instead of 7
Thanks, it was our mistake in the numbering of the figure. We changed "7" to "11".

**116) L766-767:** this sentence refers to the increased DO consumption in the "deep layer" and cites figure 8 & 9. However, the two figures show GPP & CR measured in the top 20 m of the water column – so not "deep". A better writeup is needed to properly link the LDOW to local DO consumption. Furthermore, the sentence mentions how respiration overwhelmed oxygen supply from horizontal advection. There has been no report on DO horizontal advection beyond the advection of low DO by the ESSW. So the final statement of this sentence does not seem to be supported so far.
Similar answer to comment 114.

**117) L769-770:** The way this sentence is written, it sounds as if Mannino et al studied these fjords. This kind of issue is found in many sentences in the discussion section.

We edited the sentence: "The distribution of CDOM in the Patagonian fjords reveals distinctive patterns, as depicted in Figure 11. Notably, elevated CDOM concentrations were observed in the northern region, likely stemming from increased terrestrial inputs, heightened microbial activity, and prolonged CDOM residence times (Figure 13). This phenomenon aligns with observations in northern hemisphere fjords (Mannino et al., 2008). Conversely, the southern region displays lower CDOM concentrations, indicative of diminished dissolved terrestrial inputs and potentially more efficient removal processes. The proposal that localized CDOM concentration peaks within specific fjords signify the influence of local processes, including freshwater inputs, glacier melting, or biological production (Mannino et al., 2008), contributes significantly to the spatial heterogeneity of CDOM distribution in Patagonian fjords".

**118) L783:** I don't understand 'delivered' in this context. 'Present'?
We replaced "was not delivered" by "was absent".

**119) L789:** highly suspended sediments --> high concentration of suspended sediments
We edited the sentence.

**120) L790-791:** Are there any wind data to back up this hypothesis? Also, the local upwelling would be leading to shallow hypoxia because of the low-DO waters being advected upwards, because of enhanced primary production or both or something else?

We have wind data from a meteorological station installed in Puyuhuapi Fjord. The wind data was collected every 15 minutes in the period 2014-2019 (Figure XXX). A wind rose computed for this period showed a predominance of the westerly winds, helping to sustain the hypothesis presented in the manuscript.

[Figure]

Figure XXX. Wind data collected in Puyuhuapi Fjord.

**121) L793:** remove 'the' before 'local'
We removed "the".

**122) L795:** "DO produced by photosynthesis is primarily depleted through respiration": in your work, this is only true in heterotrophic fjords - you showed fjords that were autotrophic. As a general statement, this is not correct.
We have removed this sentence and modified the paragraph.

**123) L796:** "; therefore, in many oceanic environments, the GPP and CR rates tend to be tightly coupled.": I don't think this is accurate as written. Usually, they are vertically uncoupled, ie GPP is larger in the euphotic zone and respiration/remineralization dominate below that; they are coupled in a full-water-column sense.
We have removed this sentence and modified the paragraph.

**124) L795-798:** Overall, I think these sentences should be improved – concepts are not properly explained and text is poorly written. The key is that the fjords/nearshore tend to have more organic matter than the open ocean (autoc. and alloc.), therefore leading to higher oxygen consumption. Even if the top 20m are autotrophic (GPP>CR), the deeper waters can be hypoxic due to the decomposition of sinking organic matter. Note that in many parts of the text CR is treated as the only biological process consuming oxygen (ignoring remineralization, i.e. the breakdown of organic matter by aerobic bacteria, which also consumes oxygen)
We have improved the paragraph.

**125) L801-802:** Note that Reloncavi fjord is not shown in Fig 8. It is shown in Fig 9, without information on where the GPP is high (top 4m? top 20m?) and with units of gC/m2/d. Overall, this sentence does not read very well.

We eliminated the reference to the Reloncaví Fjord in this sentence.

**126) L806:** could you clarify if the 37% is of total CO2 emissions or anthropogenic CO2 emissions? I think it will be a useful thing to add to this sentence.
We edited the sentence "A recent global assessment of $CO_2$ uptake from estuaries, including tidal systems and deltas, lagoons, and fjord reported that fjords ecosystems can reduce 37% of $CO_2$ anthropogenic emissions (Rosentreter et al., 2023)".

**127) L809-810:** I find this sentence confusing, because I had understood that the deep water renewal in Puyuhuapi fjord was from ESSW waters, which are hypoxic. So the renewal would actually contribute to hypoxia/LDOW…
We edited the sentence: "Nevertheless, sporadic deepwater renewal was observed close to the bottom (160−180 meters) in Puyuhuapi Fjord owing to the inflow of dense water above the sill favoring the ventilation and inhibiting the occurrence of deep anoxia conditions (Pinilla et al., 2020), as was also

reported by Jackson et al., (2022) in Rivers Inlet, British Columbia".

We postulate along the manuscript that LDOW and hypoxia conditions in Puyuhuapi Fjord are owing to the advection of ESSW. In the manuscript of Pinilla et al., (2020), we detected deep water ventilation produced by dense-oxygenated water that was located down the ESSW, helping to decrease the hypoxia conditions. We continue studying the mechanisms involved in this process.

**128) L812-813:** 1) Remove "However,". 2) Suggestion: have a figure showing a map of lowest deep DO in the same area/volume where you calculate the residence time. It will be easier for people not familiar with the region to connect the long residence times and the low DO conditions if both are presented as a map (instead of a map and a transect).

We added new subplots to Figure 13 to show the distribution of dissolved oxygen at level 150 meters. The new results showed the coincided area with longer residence time with the LDOW and hypoxia.

We edited the sentence: "The first map of residence time in the northern Patagonian fjord showed a coincidence of the area where a high residence time was reported (100–250 days) with the depleted oxygen region, for example, hypoxic and LDOW areas (Figure 13). The hydrodynamic model and derived calculations are helpful tools for explaining the oxygen distribution and patterns in the northern Patagonian fjord system".

**129) L820:** "sediments" --> "particulate matter"
We edited the sentence.

**130) L821:** "allowed for the first time the scrutinization of…": this sentence does not sound right. Please improve. Suggestion: "allowed for a holistic evaluation of…"
We edited the sentence.

**131) L830:** Suggestion: "contributing to the reduction of" --> "leading to slow"
We edited the sentence.

**132) L840:** 'favoring' --> 'leading to'
We edited the sentence.

**133) L852:** please check if this statement is in agreement with the journal's open data policy
We checked the journal policy, and the section was well-written.

**134) References:** many references are given such that they follow a previous reference, instead of starting in a new line
We re-entered the "references" section with the correct format.

**Second review of *Oceanographic Processes Driving Low Oxygen Conditions Inside PatagonianFjords* by Linford et al.**

-        The authors have done a nice job of tightening up and clarifying the manuscript. It is an important manuscript that brings together many different types of data. I sometimes struggledwith the storyline because it attempts to bring together many data types that have been collected in different space and time to tell the story of oxygen in the Patagonian fjord system. Because of this, the manuscript sometimes seems too speculative and I have documented my concerns below. I recommend this manuscript for publication after moderate revisions.

**1) Line 54** – Is this correct? There are many natural mechanisms that can lead to hypoxia(e.g. remineralization, weak circulation, age of water, etc.)
We edited the sentence: "The origin of the hypoxia and LDOW are attributed to natural and anthropogenic processes, e.g., remineralization of organic matter, weak circulation, extended residence time, and stratification was reported as a natural process (Rabalais et al., 2010; Bianchi et al., 2010). In contrast, water eutrophication was documented as one of the main anthropogenic processes.  (Díaz et al., 2001; Conley et al., 2009; Meire et al., 2013)"

**2) Lines 82 to 83** – I find this sentence confusing; where exactly does the OMZ decrease insize and strength?
We modified the sentence "Along the Perú-Chile coastline, the Eastern South Pacific (ESP) OMZ extends poleward, decreasing in strength and size to the south near the Patagonian fjord system (Silva et al., 2009)" by "Along the Peru-Chile coastline, the Eastern South Pacific (ESP) Oxygen Minimum Zone (OMZ) extends poleward. As one moves southward, it gradually diminishes in both size and strength until it reaches near the Patagonian fjord system(Silva et al., 2009)".

**3) Lines 85 to 87** – Here the ESSW and OMZ and discussed separately. But isn't ESSW one ofthe reasons that the OMZ is there?
We have considered that the sentence is correct, we made no modifications.
"While it is true that the ESSW carries oxygen-poor water as it originates in the equatorial region and travels southward, it's important to note that the OMZ is a complex phenomenon influenced by physical and biological interactions related to the upwelling system. ESSW is a contributing factor, as it introduces oxygen-poor water to the OMZ, but the formation and maintenance of the OMZ involve a broader range of processes. In essence, they are interconnected, but they are not the same."

**4) Lines 102 to 104** – I think there are some typos here.
We corrected the typos.

**5) Figure 1** – Much better. Though I notice that the moorings aren't on this map.
We added the positions of moorings to Figure 1b.

**6) Line 131** – Is 2000 profiles correct? Line 37 says that 1507 stations were sampled. Doesthis mean that there were 1507 separate stations sampled and only some of these stations were sampled more than once?
We detected a number of oceanographic cruises out of the study area, then we eliminated and therefore the new number of the files (1507 stations) is less than the previously reported (2017 stations).

**7) Line 158** – add 'salinity' after EW,
We indicated in the previous sentence (155-157) that the water masses were classified using the salinity criterion. We believe that it is not necessary to write the additional word "salinity" after the EW.

**8) Section 2.2** – I don't see a mention of the 2009 data in this section
We added a new sentence to the section 2.2. "Additionally, we presented values of GPP and CR from February 2009 in Figure 9, published before by Montero et al. (2011)".

**9) Section 2.4** – Figure 10 shows the concentration of suspended particulate matter from satellites. Yet the authors interpret these figures in section 3.3 as the color of the water.I'm confused here – how can the authors tell the colour of the water from SPM data?

We added a sentence and the reference used to the calculation of the satellite suspended particulate matter: "We calculated SSPM following Nechad et al., (2016) a semi-empirical algorithm adapted to complex coastal and riverine waters.
Additionally, we changed SPM from satellite to SSPM to clarify the description of the SPM obtained from biogeochemical sampling.

**10) Section 2.5** – I don't see the location of the moorings on any map. Also, there is no description of how the currents were rotated.

We added the positions of moorings to Figure 1b.

**11) Lines 367 to 379** – This is a repeated paragraph
We deleted the repeated paragraph.

**12) Line 387** – This is the first mention of ice melt. What kind of ice are the authors suggesting – sea ice or glacial melt? What proof do they have of the ice melt?
We added a new paragraph to the introduction section as was recommended by reviewer #1.
"The main contributor to the increase of allochthonous organic matter is the river supplies, mainly during late winter and early spring, owing to the dominance of the ice melting instead of the precipitation during the winter season. As one of the highest river discharges highlighted, the Puelo, Petrohue, and Cochamo Rivers in the Reloncaví Fjord, the Cisne River in the Puyuhuapi Fjord, and other freshwater contributions from small rivers in the northern Patagonian fjords (Castillo et al., 2016; Schneider et al., 2014)".

**13) Lines 392 to 395** – How was this quantified?
We don't quantify the different sources of freshwater supply influencing the salinity. For the description of this sentence, we used the geographical map presented in Figure 1b, and with the location of glacial and rivers, we infer the relationship of this source with the Estuarine water formation. We understand and agree with you about the significance of the quantification processes and will be a new challenge for future research using Isotopic analysis.

**14) Figures 2** – Please make the fonts on the place names bigger. Also, please remind thereaders in the caption what positive and negative AOU values mean.

We deleted the information of the AOU from the text and Figure 2 as was recommended by reviewer #1.
**15) Figure 3** – I can't see the symbols. Please make them bigger.
We edited the symbols in Figure 3.

**16) Figure 4** – Please make the fonts of the place names bigger
We edited the font size of the places in Figure 4.

**17) Figure 5** – Please make the fonts of the place names bigger. Also, I suggest adding isohalines on this figure so the authors can see whether the stratification is linked tosalinity.
We edited Figure 5.

**18) Figure 6** – Why were these specific dates chosen to display?
We selected these examples to show the places with the shallower hypoxia conditions (Magdalena Sound) and lower oxygen values (Quitralco Fjord) in Patagonian fjords. The selection of the dates depended on the data available.

**19) Figure 7** – I can't see the labels on the map. Also, why were these specific dates chosento display?

We edited the labels on the map. We selected a cruise during the spring (November 2020) and winter (August 2021) seasons to show the Chl-a patterns. The selection of the dates depended on the data available.

**20) Lines 528 to 532** – Are there in situ data to back this up?
We added a supplementary figure (Figure S9) to the manuscript to show the in-situ data of phytoplankton sampling.

[Figure]

Figure S9. *In-situ* phytoplankton sampling in the Reloncaví Fjord and Reloncaví Sound showing the abundance of *Skeletonema costatum* during May 2017.

**21) Line 530** – I don't think that SPM will tell you whether a diatom bloom was present
We agree with you, but we don't mention the use of SPM to identify the bloom, instead the true color images.
We edited the sentence: "In the southeast part of the Reloncaví Sound, the observed high concentration of suspended particles was accompanied by the presence of a diatom bloom (evidenced by greenish waters at the true color images in the southeast part of the Reloncaví Sound), mainly formed by *Skeletonema costatum*,…….."

**22) Line 538** – I don't think that SPM will tell you whether the sediments are carbonate-rich
We eliminated the speculative sentence.

**23) Line 539** – I don't think that SPM will tell you about the concentration of organic matter
We eliminated the speculative sentence.

**24) Lines 580 to 589** – How did CDOM and SPM relate to the satellite images?

Although numerous studies explore the relationship between Chromophoric Dissolved Organic Matter (CDOM) and Suspended Particulate Matter (SPM) using satellite imagery, we find it inappropriate to compare these variables obtained in situ and satellite images directly. The distinct sampling dates for each dataset introduce temporal misalignments that may compromise the validity of such comparisons. Furthermore, considering the vast expanse and climatology of the sampled area, it is highly improbable that our field sampling coincided with the exact dates of satellite data acquisition. Lastly, it's crucial to acknowledge that the region is characterized by high cloud cover, further complicating the acquisition of high-quality satellite images.

**25) Section 3.5** – I struggled with this section because I wasn't sure what the key points are that the authors are trying to make. I was left wondering whether this section is necessary?

We added this section to demonstrate with in-situ data the areas where intense and weaker velocities were recorded. We used the in-situ ADCP data and the circulation model to highlight the significance of marine currents in deep water ventilation and, therefore, discuss the mechanisms causing hypoxia in Patagonian fjords.

**26) Figure 12** – What do positive and negative current values mean? How were the currentsrotated? Where are these moorings – there are no maps that show this

We edited the sentence: "In (a and c) the red-blue color represents the eastward-westward currents, and in (e) red-blue color represents northward-southward currents".
The location of the moorings is presented in Figure 1b.

We added a new paragraph to explain how currents were rotated: "The currents were rotated following the main axis of the Guafo channel and the Puyuhuapi Fjord (Figure 1), thus implying that in Guafo, currents were rotated 14º south of the east whereas in Puyuhuapi south currents were rotated 36º north of the east and in Puyuhuapi north the axis was rotated 22º east to the north. Following those rotations, the contours of the time series of the along-channel/fjord components are shown in Figure 11".

**27) Line 610** – I'm not sure that this current is strong?
We changed "strong" to "intense".

**28) Lines 663 to 665** – I don't think that this study scrutinized all processes contributing tohypoxic water. For example, the anthropogenic impact wasn't discussed.
We edited the sentence: "Our study scrutinized the natural processes contributing to the presence of water under hypoxic conditions and LDOW in the northern Patagonian fjords, as discussed in the next section".

**29) Lines 729 to 731** – I don't see any proof to back this statement up.

We deleted the sentence.

**30) Lines 805 to 812** – these manuscripts might be of interest in this section:
o   Jackson et al., 2023
      https://agupubs.onlinelibrary.wiley.com/doi/10.1029/2023GL104549
o   Thomson et al., 2017
      https://agupubs.onlinelibrary.wiley.com/doi/full/10.1002/2016JC012512
-   o